# bacLIFE: a user-friendly computational workflow for genome analysis and prediction of lifestyle-associated genes in bacteria

Guillermo Guerrero-Egido[1,2,3,4,8], Adrian Pintado [1,3,4,8], Kevin M. Bretscher [1,2,3,4,8], Luisa-Maria Arias-Giraldo[2], Joseph N. Paulson[5], Herman P. Spaink [1], Dennis Claessen [1], Cayo Ramos [4,6], Francisco M. Cazorla [3,4], Marnix H. Medema [1,7], Jos M. Raaijmakers [1,2] & Víctor J. Carrión [1,2,3,4] ✉

Bacteria have an extensive adaptive ability to live in close association with eukaryotic hosts, exhibiting detrimental, neutral or beneficial effects on host growth and health. However, the genes involved in niche adaptation are mostly unknown and their functions poorly characterized. Here, we present bacLIFE (https://github.com/Carrion-lab/bacLIFE) a streamlined computational workflow for genome annotation, large-scale comparative genomics, and prediction of lifestyle-associated genes (LAGs). As a proof of concept, we analyzed 16,846 genomes from the *Burkholderia/Paraburkholderia* and *Pseudomonas* genera, which led to the identification of hundreds of genes potentially associated with a plant pathogenic lifestyle. Site-directed mutagenesis of 14 of these predicted LAGs of unknown function, followed by plant bioassays, showed that 6 predicted LAGs are indeed involved in the phytopathogenic lifestyle of *Burkholderia plantarii* and *Pseudomonas syringae* pv. phaseolicola. These 6 LAGs encompassed a glycosyltransferase, extracellular binding proteins, homoserine dehydrogenases and hypothetical proteins. Collectively, our results highlight bacLIFE as an effective computational tool for prediction of LAGs and the generation of hypotheses for a better understanding of bacteria-host interactions.

Microorganisms exhibit an immense taxonomic and niche diversity which defines their lifestyles and critical roles in numerous functions in natural and man-made ecosystems[1–4]. They are universally found in diverse environments, playing varied roles that span from acting as vital catalysts for biogeochemical processes, such as (C, N) cycling and greenhouse gas emissions, to significantly influencing the growth, development, and well-being of their eukaryotic hosts[5–7]. The distinctive functions of each bacterium within its specialized niche

[1]Institute of Biology, Leiden University, Sylviusweg 72, 2333 BE Leiden, The Netherlands. [2]Department of Microbial Ecology, Netherlands Institute of Ecology (NIOO-KNAW), Droevendaalsesteeg 10, 6708 PB, Wageningen, The Netherlands. [3]Departamento de Microbiología, Facultad de Ciencias, Campus Universitario de Teatinos s/n, Universidad de Málaga, 29010 Málaga, Spain. [4]Departamento de Protección de Cultivos, Instituto de Hortofruticultura Subtropical y Mediterránea "La Mayora", Campus Universitario de Teatinos, Universidad de Málaga-Consejo Superior de Investigaciones Científicas (IHSM-UMA-CSIC), 29010 Málaga, Spain. [5]Department of Data Sciences, N-Power Medicine, Redwood City, CA 94063, USA. [6]Área de Genética, Facultad de Ciencias, Campus Universitario de Teatinos s/n, Universidad de Málaga, 29010 Málaga, Spain. [7]Bioinformatics Group, Wageningen University, Droevendaalsesteeg 1, 6708 PB Wageningen, The Netherlands. [8]These authors contributed equally: Guillermo Guerrero-Egido, Adrian Pintado, Kevin M. Bretscher. ✉e-mail: vcarrion@uma.es

contribute to its unique phenotype or lifestyle. The presence of genes associated with a particular lifestyle within a niche is crucial in defining the function or lifestyle of each bacterium and determining its interactions with other organisms. For plants, microbes can promote growth via enhanced nutrient acquisition, production of phytohormones or remediation of contaminated soils[8,9]. They can confer protection against pathogens[10], insect pests[11] and abiotic stresses such as salinity and drought[12,13]. In return, plants provide a 'home' and 'shelter' for microbes, food for growth, and a means of dispersion through pollinators interacting with the plant[14]. However, microorganisms can also have deleterious effects on their hosts via infections or the production of cell wall-degrading enzymes and toxins that adversely affect growth and development[15].

For many bacterial and fungal genera, the genetic basis of these two contrasting lifestyles is not well understood. Previous studies have shown the existence of genetic features in plant-associated bacteria such as the repertoire of genes involved in carbohydrate metabolism[6,16]. One of the best examples of how thin the line can be between being pathogenic or beneficial to their animal or plant hosts can be found in members of the Burkholderiaceae family. For pathogenic *Burkholderia* species, three main groups can be distinguished: (i) members of the *B. cepacia* complex cause severe infections in cystic fibrosis or immunocompromised patients but can also exhibit plant-beneficial traits[17,18], (ii) members of the *B. pseudomallei* complex have an opportunistic or obligate pathogenic lifestyle in humans and cause melioidosis[19]; and (iii) members of the *B. plantarii*, *gladioli* and *glumae* group cause disease in various plant species, particularly in rice[20]. The Burkholderiaceae family also encompasses the genus *Paraburkholderia* (previously taxonomically delineated as *Burkholderia*) harboring primarily environmental species with beneficial effects on plants[21]. For example, *Paraburkholderia phytofirmans* is a well-studied endophytic bacterial species that promotes plant growth and enhances stress tolerance[22–24]. Similarly, *P. dipogonis* and *P. diazotrophica* contribute to nitrogen acquisition via nodulation[25,26]. A second, well-known bacterial genus with diverse lifestyles is *Pseudomonas*, encompassing more than 316 characterized species to date[27]. *Pseudomonas* species have been shown to (i) promote plant growth and protect against diseases[28], (ii) remediate hydrocarbon-contaminated soils[29], and (iii) infect plants and animals, causing various diseases[30–32].

For both bacterial genera, various traits and genes have been identified for their involvement in the beneficial effects as well pathogenicity[20,32]. To date, however, it has been difficult to discriminate between the different lifestyles within these two genera as well as of many other bacterial species associated with eukaryotic hosts. For *Burkholderia* and *Pseudomonas*, several genetic markers have been used in Multi Locus Sequence Typing analyses to discriminate between pathogens and environmental species or strains[33,34]. Although this approach enables classification, it does not facilitate the identification of genomic signatures linked to each bacterial lifestyle. Prior computational methodologies have demonstrated the feasibility of identifying genetic markers associated with different lifestyles by examining the presence or absence of gene families or orthologous genes in bacterial populations[6]. Databases of genes with a described plant-growth-promoting potential (PGPP) or virulence factors have been established, allowing users to quickly map genes of interest and reveal the pathogenic or PGPP potential[35,36]. Additionally, machine learning models have been explored in predicting bacterial lifestyles using such data, as evidenced by recent studies[16]. However, despite the existing computational methodologies and databases, there remains a critical need for a comprehensive and user-friendly tool that enables researchers to accurately predict bacterial lifestyles, identify candidate genes, and explore the genomic signatures associated with diverse bacterial genera and their interactions with eukaryotic hosts.

Here, we provide a new streamlined computational workflow that annotates bacterial genomes and performs large-scale comparative genomics, to predict the bacterial lifestyle and to pinpoint candidate genes and biosynthetic gene clusters associated with that lifestyle. To test the potential of this bioinformatic tool, designated bacLIFE, we extracted 16,846 genomes of species and strains belonging to the Burkholderiaceae family and *Pseudomonas* genus. We then developed a computational approach for comparative genomics, where genes are grouped into similar function gene clusters. By machine learning of the gene cluster distributions, we first tested the predictability of the different lifestyles and pinpointed the candidate genes that were significantly associated with these lifestyles. This study primarily focused on general lifestyles categorized by pathogenic interactions with other organisms, particularly distinguishing between animal, plant pathogens and environmental bacteria with a non-pathogenic lifestyle. It is essential to note that users should determine the appropriate lifestyle classes based on their specific research questions.

For *Burkholderia/Paraburkholderia* and *Pseudomonas*, we obtained a total of 786 and 377 predicted lifestyle-associated genes (pLAGs) for a phytopathogenic lifestyle, respectively. We also predicted genomic regions enriched in virulence factors by looking at the pLAGs' positions. Using site-directed mutagenesis and phenotypic characterization, we experimentally validated the functional role of 5 yet unknown true LAGs in pathogenicity of *Burkholderia* on rice and of a novel true LAG, encoding a Non-Ribosomal Peptide Synthetase (NRPS), in pathogenicity of *Pseudomonas* on bean. Collectively, the results of this study showed that bacLIFE is a user-friendly, interactive, and effective framework for genome-wide diagnostics and prediction of LAGs in bacterial genera living in diverse environments and on/in different eukaryotic hosts. Paired with experimental validation, bacLIFE's pLAGs can be confirmed as true LAGs thereby enhancing confidence in the identification process.

## Results and discussion

### bacLIFE: a user-friendly framework for large-scale comparative genomics, lifestyle prediction, and identification of lifestyle-associated genes

bacLIFE was built and written in Python and R, is organized using a Snakemake workflow manager[37], and is freely available as open-source software (https://github.com/Carrion-lab/bacLIFE). bacLIFE takes full and draft genome sequences in fasta format as input to automatically generate clusters of genes and absence/presence matrices of these clusters in the genomes, which constitute the basis for its predictions. The pipeline consists of three modules (Fig. 1). The clustering module (i) uses Markov clustering (MCL)[38] in combination with linclust from the MMseqs2 tools[39] (Supplementary Fig. 1), enabling the generation of a database of functional gene families (see Methods). The MCL method, based on *all vs all* BLAST alignment statistics, is known as an accurate algorithm to cluster proteins by function[40]. In addition, bacLIFE introduces a novel approach by integrating antiSMASH[41] and BiG-SCAPE[42] to also generate absence/presence matrices at the BGC level. The lifestyle prediction module (ii) employs a random forest machine learning model to forecast bacterial lifestyle or other specified metadata from these matrices (Supplementary Fig. 2). In the analytical module (iii) the results from the above-described modules are embedded in a Shiny user-friendly interface for comprehensive and interactive comparative genomics (Supplementary Fig. 3). In this interface, bacLIFE enables the exploration, visualization, and downstream analysis by Principal Coordinates Analysis (PCoA), dendrograms, pan-core-genome analyses and prediction of lifestyle-associated genes (pLAGs). A pLAG is referred to as a gene or gene cluster that exhibits a distinct pattern of presence for a specific (annotated) lifestyle while being largely absent in other lifestyles. The novelty and strength of bacLIFE lies in the combination and integration of diverse analytical tools in a logical, automated, and interactive framework for comparative genomics and prediction of new LAGs. For

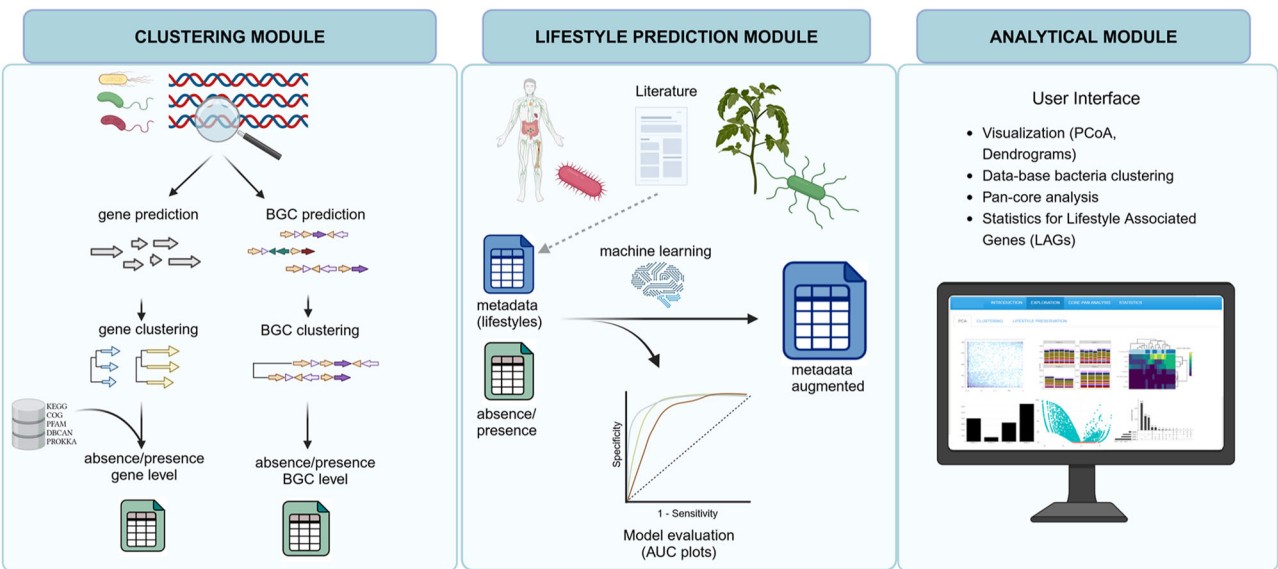

**Fig. 1 | bacLIFE flowchart of the three-step pipeline.** The first step of the bacLIFE framework is to download the genomes. The second step is to reduce the redundancy by clustering genomes at the 0.99 Average Nucleotide Identity (ANI) using Mash[85] and representative genome selection. The bacLIFE framework is divided into three modules (from left to right panel): (i) Prediction and clustering is performed at the gene and Biosynthetic Gene Cluster (BGC) level in the clustering module. Genes are clustered by function into gene clusters using the Markov cluster algorithm (MCL) and BGCs are clustered into Gene Cluster Families (GCFs) using BiG-SCAPE[42]. Gene cluster and GCF absence/presence matrices are generated as output of the clustering module and used as input for the next modules. (ii) In the lifestyle prediction module, metadata about bacterial lifestyle is collected from literature. The predictability of the metadata with the absence/presence matrices is tested and evaluated with machine learning models as random forest. The accuracy of the predictions is tested and the metadata is augmented by assigning lifestyles to unknown-lifestyle bacteria. (iii) The augmented metadata and absence/presence matrices are used as input in the analytical module for further statistical analysis to identify Lifestyle Associated Genes (LAGs). Additionally, different visualization options are included in this module. Figure made in BioRender.com.

more detailed information, please see the documentation at https://github.com/Carrion-lab/bacLIFE.

## Lifestyle prediction of *Burkholderia*/*Paraburkholderia* and *Pseudomonas* species

*Burkholderia*/*Paraburkholderia* and *Pseudomonas* genera (4611 and 12,235 genomes, respectively) were selected as case studies to evaluate the accuracy and predicted LAGs of bacLIFE due to the ample knowledge regarding their habitats and lifestyles (Supplementary Datasets 1). The computational time and speed of the bacLIFE framework was reduced by clustering all genomes at 99% Average Nucleotide Identity (ANI) similarity, resulting in 644, 200, and 2050 genomes of *Burkholderia, Paraburkholderia* and *Pseudomonas*, respectively (Supplementary Datasets 1). Thus, we mitigate the impact of genome redundancy present in multiclonal genomes, potentially influencing statistical analyses. Based on literature, three main lifestyles were defined in both datasets: environmental (e.g., *Paraburkholderia* spp., *P. fluorescens)*, opportunistic animal pathogens (e.g., *B. cepacia* complex, *B. pseudomallei* complex, *P. aeruginosa* and *stutzeri*) and plant pathogens (e.g., *B. plantarii, glumae, gladioli, P. syringae* and *viridiflava*) (Supplementary Datasets 2). A significant proportion of genomes in both datasets remained unassigned to any lifestyles due to the absence of species-level designations in NCBI, which may compromise statistical power in the further steps of the bacLIFE pipeline.

To predict the lifestyle of non-annotated genomes, we trained and tested a random forest machine learning model with the gene cluster absence/presence matrices output from the bacLIFE clustering module. The *Pseudomonas* lifestyle prediction accuracy, represented by the sensitivity and specificity in ROC plots using five-fold cross-validation, showed that the Area Under the Curve (AUC)[43] (Supplementary Fig. 4) was in the range of 0.96–0.99 for each lifestyle prediction (opportunistic animal pathogen, plant pathogen, environmental). These results are in line with the AUC values (0.985) reported for plant-associated *Pseudomonas*[16]. Regarding the *Burkholderia*/*Paraburkholderia* dataset,

the random forest algorithm demonstrated exceptional performance, achieving perfect classification with an Area Under the Curve (AUC) value of 1.00 for each lifestyle (Supplementary Fig. 4).

To assess the random forest algorithm's performance in predicting the lifestyle of potential novel species with no representative in databases, we employed a leave one species out validation. In the *Burkholderia*/*Paraburkholderia* dataset, we trained the random forest model without including any *B. vietnamiensis* genome, which is a known opportunistic human pathogen. Using this model, we predicted the lifestyle of *B. vietnamiensis* genomes and found that all 40 genomes were classified as opportunistic human pathogens with 90% of the trees voting for the same class. Using the same approach, the random forest algorithm correctly classified 92/124 *P. fluorescens* genomes as environmental strains and 30/35 *P. cichorii* genomes as plant pathogens, with a minimum number of trees voting for the class of 70% for both predictions. The lower performance of *P. fluorescens* in this analysis can be explained due to the reclassification of some genomes previously labeled as 'fluorescens', now designated as other species. This can be observed in genomes such as *P. fluorescens* W-6 (Supplementary Datasets 2), where the classifier value for an environmental lifestyle is 0.65 while most of fluorescens genomes exhibit values of more than 0.9.

The application of machine learning enabled us to assign lifestyles to these genomes by leveraging metadata from well-studied species/strains. Therefore we predicted the formerly unknown lifestyles of 137 out of 145 (94.5%) and 740 out of 976 (75.9%) *Burkholderia/Paraburkholderia* and *Pseudomonas* strains, respectively (Fig. 2 and Supplementary Datasets 2). Using bacLIFE's clustering output from the *Burkholderia/Paraburkholderia* and *Pseudomonas* datasets and the augmented metadata file containing the predicted lifestyles (Supplementary Datasets 2), Principal Coordinate Analysis (PCoA) was used to visualize the lifestyles distribution (Fig. 2). For *Burkholderia/Paraburkholderia*, members of the *Paraburkholderia* genus, renowned for their beneficial effects on plants, formed a cohesive cluster

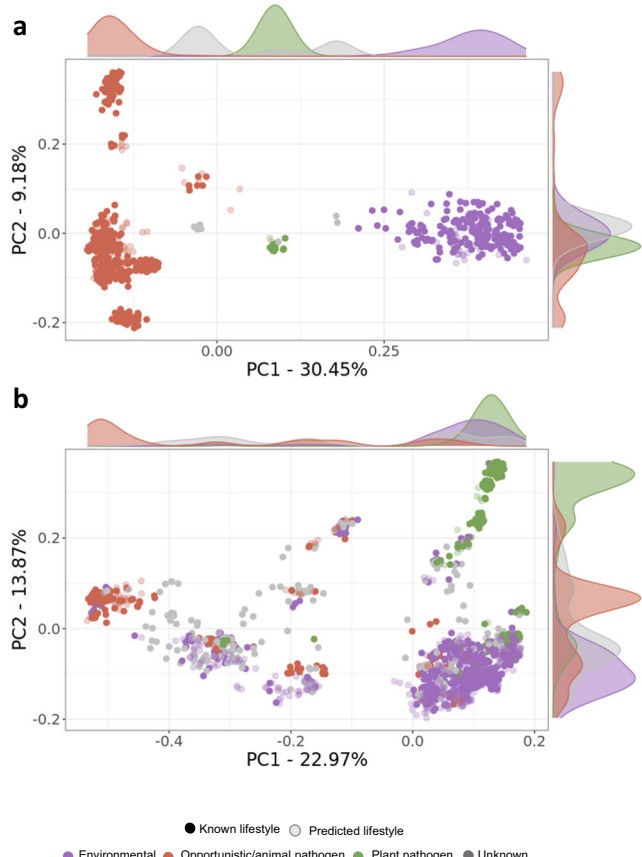

**Fig. 2 | A systematic analysis of the *Burkholderia/Paraburkholderia* and *Pseudomonas spp.* lifestyles. a**, **b** Principal Coordinate Analysis (PCoA) plots based on the dice dissimilarity calculated using the absence/presence matrix output by the bacLIFE clustering module for *Burkholderia/Paraburkholderia* and *Pseudomonas spp.*, respectively. Each point in the plot represents a genome, and the color indicates the annotated (dark color) or predicted (light color) lifestyle classification. Density plots in both axes are displayed showing the distribution of points for each lifestyle along each axis.

(Supplementary Fig. 5a). On the other hand, plant pathogens like *B. gladioli*, *glumae*, and *plantarii* formed a smaller cluster situated between the *Paraburkholderia* cluster and the opportunistic animal pathogen group. Within the opportunistic animal pathogen group, we observed more variability attributed to various species and bacterial complexes such as the *B. cepacia* and *B. pseudomallei* complex. For *Pseudomonas*, well-known plant pathogens such as *P. syringae, savastanoi* and *viridiflava* among others grouped together and apart from the animal pathogenic *P. stutzeri, mendocina,* and *aeruginosa* (Supplementary Fig. 5b). Environmental *Pseudomonas* species and strains were found distributed widely, with the majority forming a larger cluster, containing plant-beneficial *P. fluorescens* and *protegens*, the plant pathogens *P. gingeri* and *P. palleroniana* and the animal pathogens *P. monteilii* and *juntendi*. These findings provide additional evidence of the adaptive ability of *Pseudomonas* species to shift between lifestyles depending on the environmental conditions[44]. In summary, the bacLIFE pipeline substantially improved the lifestyle prediction of poorly classified *Burkholderia/Paraburkholderia* and *Pseudomonas* species and strains using machine learning algorithms.

When examining the distribution of lifestyles across the phylogeny derived from both datasets, we observed a strong correlation between lifestyle and phylogeny (Fig. 3 and Supplementary Fig. 6). In the *Burkholderia/Paraburkholderia* dataset (Fig. 3), all plant pathogenic species cluster within the same phylogenetic clade. Similarly, in

the *Pseudomonas* dataset (Supplementary Fig. 6), 80% of plant pathogens belong to the same clade, corresponding to species from the syringae complex. However, approximately 20% of plant pathogens, including species like *P. gingeri* and *P. corrugata*, exhibited distinct phylogenetic delineation in the tree. This high interconnection between lifestyle and phylogeny in both datasets poses a challenge in distinguishing between genes associated with a genuine lifestyle impact and those associated primarily with phylogeny.

### Identification of general and lifestyle-associated functions

The relative abundance of general functions among the different lifestyles was calculated for *Burkholderia/Paraburkholderia* and *Pseudomonas* using the absence/presence gene cluster matrices from the bacLIFE clustering module with the COG-aggregated[45] functional categories (Fig. 3, Supplementary Figs. 6 and 7). COG categories RNA processing and modification (A) and Cytoskeleton (Z) were removed from the analysis because they are mainly based in Eukaryotes. Statistical analysis (Kruskal–Wallis $p < 0.05$, Supplementary Datasets 3) showed that most COG categories were different among groups, suggesting the presence of specific LAGs (Supplementary Fig. 7). Clade-specific differences within a lifestyle were also observed, for example in the *B. ubonensis* group, for which a higher relative abundance of genes involved in Defense mechanisms (V), Cell motility (N), and Extracellular structure (W) was observed relative to other species with the same lifestyle (Fig. 3). A higher relative abundance of genes involved in Carbohydrate transport and metabolism (G) was observed in environmental and phytopathogenic *Burkholderia/Paraburkholderia* and *Pseudomonas* strains as compared to strains with an opportunistic pathogen lifestyle (Fig. 3 and Supplementary Figs. 6 and 7). These results confirm and extend earlier results by Levy et al.[6]. Inorganic ion transport and metabolism (P) has been reported as more abundant in *Pseudomonas* associated with humans, according to Saati-Santamaria et al.[46]. This trait is also observed to be more prevalent in opportunistic animal-associated bacteria within both the *Pseudomonas* and *Burkholderia* genera.

We observed a significant increase in the COG category Intracellular trafficking, secretion, and vesicular transport (U). Genes related to secretion systems and vesicles, which play a role in the virulence of the plant pathogen *Pseudomonas*[47], were found within this category (Supplementary Fig. 7b). To further validate the distribution of bacLIFE's gene cluster among the genomes, given the increased prevalence of COG category U in *Pseudomonas* plant pathogens, we examined the established distribution patterns of genes related to the Type III secretion system (T3SS) and its effectors. This comparison aimed to assess whether bacLIFE's results align with the findings in the existing literature. The T3SS structural genes are divided in two groups, *hrc* genes that are highly conserved among Pseudomonads and *hrp* genes that are mainly present in plant pathogenic *Pseudomonas* such as *P. syringae*[48]. Using BLAST[49], we mapped the *hrc* and *hrp* genes (10 and 16 genes, respectively) to the bacLIFE gene clusters, revealing that *hrc* genes were widely spread among all members of the *Pseudomonas* genus, whereas the *hrp* genes were mostly exclusively present in plant pathogenic *P. syringae* (Fig. 4). Interestingly, we observed that several *Pseudomonas* species/strains, such as four *P. gingeri* and three *P. mendocina* strains, not belonging to the *P. syringae* complex contained the *hrc/hrp* genes. These results suggest the importance of horizontal gene transfer in the evolution and dissemination of this well-known virulence trait[50]. Furthermore, we analyzed the distribution of 26 well-studied Type III secretion system (T3SS) effectors[51] (Supplementary Fig. 8). These effectors were divided into two groups: 'prevalent' effectors, including AvrE, HopB, HopM, and HopAA, which are known to be present in most *P. syringae* species, and 'sparse' effectors, which are only found in a few *P. syringae* strains. Our results consistently confirmed the presence of the four prevalent effectors in the majority of *P. syringae* strains. Additionally, some

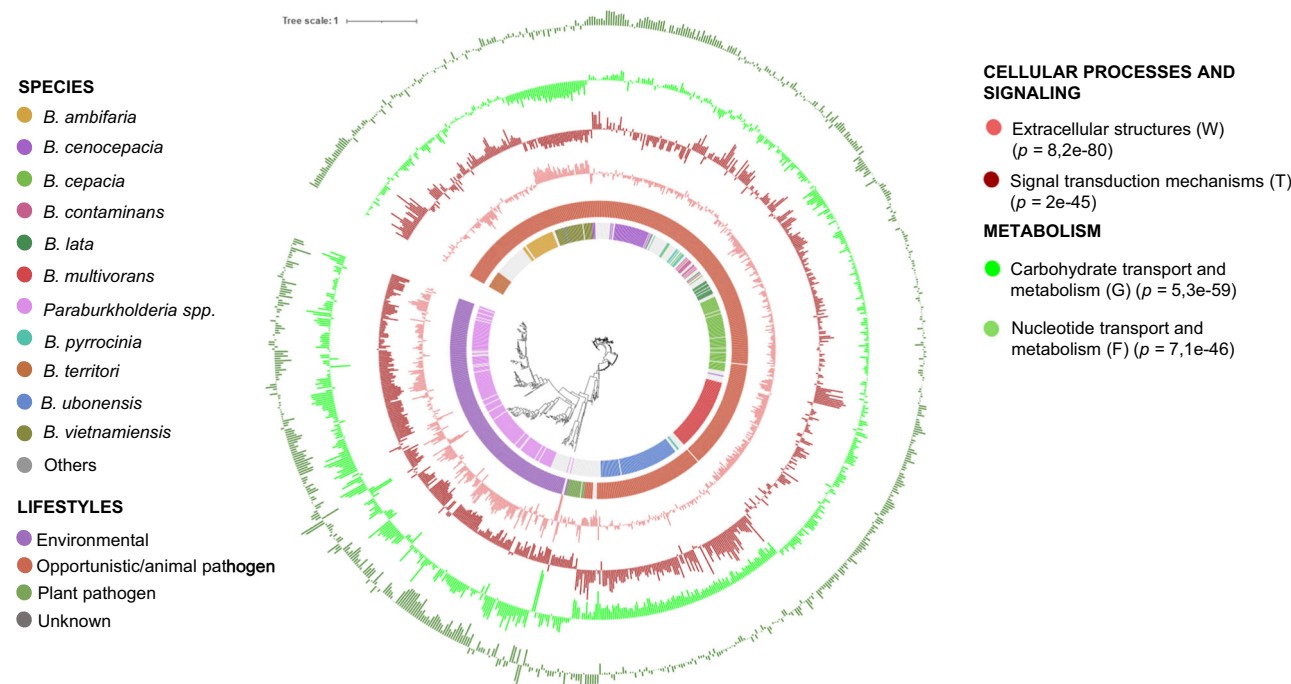

**SPECIES**

- *B. ambifaria*
- *B. cenocepacia*
- *B. cepacia*
- *B. contaminans*
- *B. lata*
- *B. multivorans*
- *Paraburkholderia spp.*
- *B. pyrrocinia*
- *B. territori*
- *B. ubonensis*
- *B. vietnamiensis*
- Others

**LIFESTYLES**

- Environmental
- Opportunistic/animal pathogen
- Plant pathogen
- Unknown

**CELLULAR PROCESSES AND SIGNALING**

- Extracellular structures (W) ($p$ = 8,2e-80)
- Signal transduction mechanisms (T) ($p$ = 2e-45)

**METABOLISM**

- Carbohydrate transport and metabolism (G) ($p$ = 5,3e-59)
- Nucleotide transport and metabolism (F) ($p$ = 7,1e-46)

**Fig. 3 | Functional categories of COGs (Clusters of Orthologous Groups) and their correlation with the predicted lifestyles in the *Burkholderia/Paraburkholderia* genera.** Phylogenetic analysis of the *Burkholderia/Paraburkholderia* genera (n = 845) based on concatenated alignment (2146 amino-acid positions) of 54 ubiquitously conserved proteins identified with PhyloPhlAn 3.0[102] and visualized/annotated using iTOL[106]. The first two inner color-coded rings depict the *Burkholderia/Paraburkholderia* species used in this study and their associated lifestyle. The phylogenetic tree is also decorated with three groups of bar plots representing the abundance of the most significant (Kruskal–Wallis, $P < 0.05$) genes associated with some selected COG[45]. The bar plots can display negative values because the zero point on the plot represents the mean percentage of that COG category in the entire dataset. Negative values indicate a proportion below the genus average, while positive values signify a proportion above the genus average. Mean proportion of the *Burkholderia/Paraburkholderia* COG categories (from inside to outside: V 1.88%, T 4.59%, N 1.42%, W 0.12%, G 6.33%, F 1.76%).

---

prevalent effectors, such as AvrE and HopB families, were also detected in other members of the *P. syringae* complex, including *P. viridiflava* and *P. cichorii*. The remaining effectors showed a sparse distribution exclusively within the *P. syringae* complex, aligning with findings reported earlier in the literature[51].

## Prediction of lifestyle-associated genes in *Burkholderia* and *Pseudomonas*

A Fisher's exact test in combination with a threshold of 70% group prevalence and >2 log2fold change between group presence is implemented in the analytical module and used to predict specific pLAGs. In *Burkholderia/Paraburkholderia*, this yielded 382, 478, and 786 pLAGs for the predicted environmental, opportunistic/animal pathogen and plant pathogen lifestyles respectively (Fig. 5a). In *Pseudomonas*, 48, 74, 377 pLAGs were found for the environmental, opportunistic/animal pathogen and plant pathogen lifestyles, respectively (Fig. 5b). The greater presence of pLAGs in *Burkholderia/Paraburkholderia* might be attributed to a stronger correlation between lifestyle and phylogeny when compared to *Pseudomonas*. Tools like phyloGLM[52] and treeWAS[53] were created to consider phylogeny when exploring gene associations. For plant pathogenic pLAGs (see methods section for more detailed information), both tools exhibited substantial overlap with the threshold used in bacLIFE. PhyloGLM, being notably more restrictive, identified fewer associations, while treeWAS, with a more relaxed approach, detected more associations compared to bacLIFE.

Both datasets demonstrated that the plant pathogen lifestyle exhibited the highest number of pLAGs, which aligns with the close clustering observed among genomes from the species/strains with this lifestyle in the PCoA analysis (Fig. 2a, b). In every lifestyle, a high

proportion (~30%) of these pLAGs code for proteins with unknown functions in the COG database (Fig. 5a, b), underscoring the larger unexplored genomic information contained within the lifestyle-associated genes in comparison with the core-genome[51]. However, pLAGs constitute a portion of the accessory genome that is better understood in terms of functionality when compared to unique genes or genes found in only a single bacterium.

## Known LAGs predicted by bacLIFE

To functionally validate the pLAGs predicted by bacLIFE, we focused on the phytopathogenic lifestyle of both taxa. We found 786 and 377 phytopathogenic pLAGs in *Burkholderia/Paraburkholderia* and *Pseudomonas*, respectively. We characterized pLAG-enriched regions as instances where five or more pLAGs were aligned in the same transcriptional direction in *B. plantarii* ATCC 43733 for *Burkholderia* and in *P. syringae* B728a for *Pseudomonas*. First, we looked into virulence mechanisms among these genomic regions and other pLAGs that are already described in literature for both plant pathogenic genera. In *Burkholderia*, we found four genomic regions enriched in pLAGs associated with type II, III, and VI secretion system proteins[54] (Supplementary Datasets 4). Interestingly, in some of these genomic regions, we also found pLAGs annotated as phage genes next to secretion system proteins. This information sheds light on the hypothetical viral origin of the region and the utilization of phage-like lysozymes in secretion systems for plant pathogenic *Burkholderia*[55]. In addition, bacLIFE detected genes for the two synthases ToxC and ToxD, responsible for the production of the virulence factor toxoflavin[56], which are exclusive to the genomes of *B. glumae* and *B. gladioli* species (Supplementary Datasets 4). Lastly, we found multiple pLAGs for N-acyl-homoserine lactones (AHLs) (Supplementary

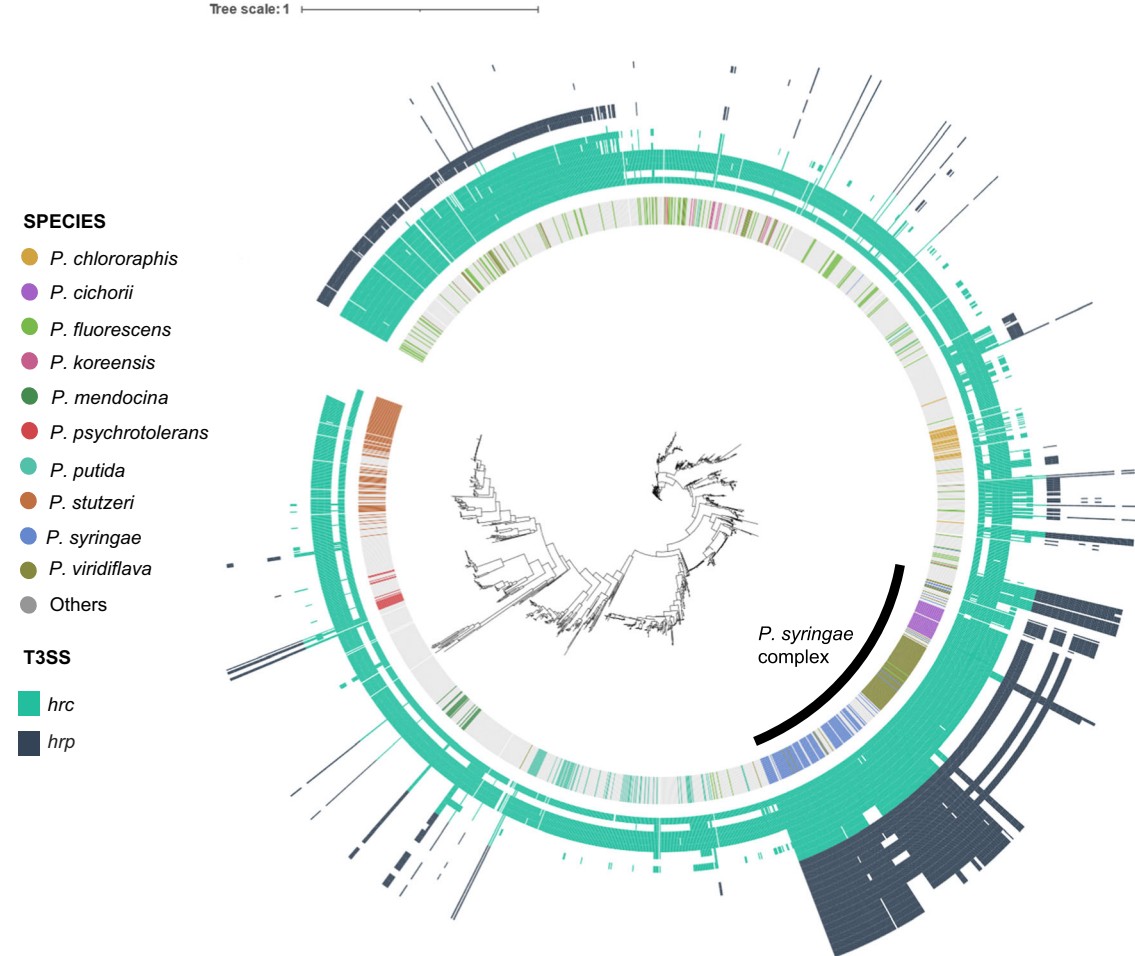

**Fig. 4 | Gene distribution of structural genes from the T3SS *hrc/hrp* in *Pseudomonas*.** Phylogenetic analysis of the *Pseudomonas* genera (*n* = 2050) based on concatenated alignment (11001 positions) of 77 ubiquitously conserved proteins identified with PhyloPhlAn 3.0[102] and visualized/annotated using iTOL[106]. The first inner color-coded ring depicts the most common *Pseudomonas* species used in this study. Bacteria from the *Pseudomonas syringae* complex are highlighted in the tree with a black line. The phylogenetic tree is also decorated with two heatmaps depicting the absence/presence of hrc genes (light blue heatmap, *n* = 10) and hrp genes (dark blue heatmap, *n* = 16).

Datasets 4). Quorum sensing, mediated by AHLs, plays a pivotal role in establishing and maintaining successful symbiotic relationships between legumes and nitrogen-fixing rhizobia[57]. Other functions of AHLs are also in pathogenicity, for example for *B. gladioli* pv. agaricicola infection of the mushroom-forming fungus *Agaricus* spp.[58].

For *Pseudomonas*, bacLIFE predicted a group of consecutive genes associated with the type II secretion system and its effectors[59] (Supplementary Datasets 5). Additionally, bacLIFE detected a group of more than 10 consecutive genes involved in tellurite resistance (Supplementary Datasets 5) in *Pseudomonas* plant pathogens. Tellurite resistance is known to be present in plant pathogenic *P. syringae*[60,61] and confers resistance to copper. Copper-based chemicals have been widely employed to combat bacterial diseases, resulting in the emergence of copper resistance mechanisms[61,62]. Previous studies have demonstrated that host plants utilize copper ions as a defense strategy against bacterial infections, activating immune responses through defense signaling pathways[63,64].

We compared plant pathogenic pLAGs in *Burkholderia* and *Pseudomonas*. Using BLAST, we searched for bidirectional best hits between plant pathogenic pLAGs of *Pseudomonas* and *Burkholderia*, finding 13 shared pLAGs out of 786 in *Burkholderia* and 377 in *Pseudomonas*. This suggests a minimal 1–3% overlap between plant pathogen pLAGs of these two genera. Among these 13 genes, we find several sugar transporters (Supplementary Datasets 6). Sugar

transporters play a crucial role in plant pathogenic bacteria as they are essential for these microbes to obtain sugars from their plant hosts[65]. These transporters, which are common among plant pathogens in two different genera, are likely responsible for the uptake of specialized sugars from plants, enabling these bacteria to thrive in the hostile environment they encounter.

Lastly, we performed a BLAST[49] analysis on the 786 phytopathogenic pLAGs in *Burkholderia/Paraburkholderia* and the 377 phytopathogenic pLAGs in *Pseudomonas*. These were compared against the PHI-base, a database containing genes experimentally validated for their roles in pathogen-host interactions[35]. By selecting the representative sequence from all plant pathogen pLAGs gene clusters in *Burkholderia*, we conducted BLAST searches against the PHI-base, resulting in 253 out of 786 significant hits (e-value < 1e-2), which accounts for 16.4% of the queried pLAGs (Supplementary Datasets 4). In contrast, when we used the pan-genome as a representative sequence for all gene clusters, the BLAST analysis yielded 4788 hits out of a total of 157,696 aligned sequences, representing only 3% of the total. Similarly, in the case of *Pseudomonas*, 142 out of the 377 plant pathogen pLAGs had significant hits (Supplementary Datasets 5), while 4693 hits were obtained out of 191,300 gene clusters (2.45%) comprising the pan-genome. This comparison underscores the enrichment of virulence factors within the plant pathogen LAGs of *Burkholderia* and *Pseudomonas* ($\alpha$ = 1e-5, Fisher's exact test). While we have a higher

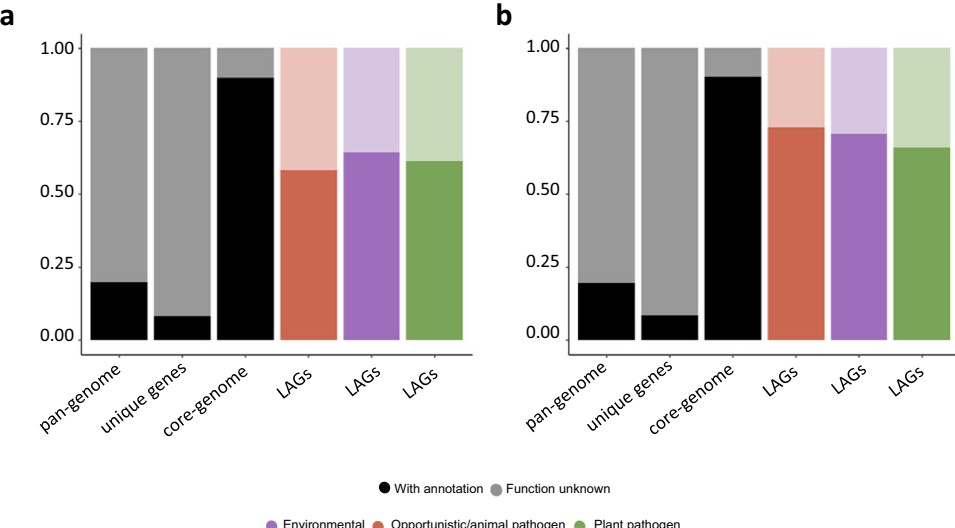

**Fig. 5 | *Burkholderia/Paraburkholderia* and *Pseudomonas spp.* Lifestyle Associated Genes (LAGs).** Stacked bar plots representing the function known (dark colored) and unknown (light colored) in **a** *Burkholderia/Paraburkholderia* spp. and **b** *Pseudomonas* spp. pan-genome, unique genes or genes only present in one genome, core genes or genes present in >90% of all genomes and LAGs. LAGs are defined as gene clusters that are present at least in 70% of the genomes belonging to a lifestyle/group and more than a 2 log2 fold change between the lifestyle/group mean and the remaining bacteria mean presence. Core genes are defined as gene clusters present in >90% of the genomes across the complete dataset. Unknown function gene clusters are defined as clusters containing genes without any COG annotation provided by the eggNOG-mapper tool.

proportion of hits within our pLAGs, the majority of these hits were observed in the pan-genome. This phenomenon can be attributed to the filtering criteria we applied, specifically the requirement of >70% group presence and >2 log2fold change to define a pLAG. It is worth noting that genes excluded by this filter may still play roles in interactions with the host. These genes could include those present in only a few bacteria and are pathovar-specific, as well as genes found in most bacteria but are involved in host interactions only in certain strains. Consequently, these genes may not satisfy the rigorous pLAG criteria, highlighting a limitation of bacLIFE, which primarily operates based on gene distributions.

## Discovery of new LAGs using bacLIFE

The high proportion (~30%) of pLAGs in plant pathogenic bacteria with unknown functions (Fig. 5 and Supplementary Datasets 4, 5) provided a basis for the discovery of novel virulence factors in both genera. To validate the importance of these novel pLAGs in virulence, we selected 13 pLAGs associated with the phytopathogenic lifestyle in *Burkholderia*. The bacLIFE Shiny app allows users to generate a table of the detected pLAGs in a specific genome of the dataset. This table includes the genome position of the pLAGs and allows to find regions enriched in pLAGs. We prioritized these regions to find potential operons but we wanted to also test the role of pLAGs that were in non-pLAG-enriched genomic regions. Therefore, we classified these pLAGs as either pLAGs in genomic regions enriched in pLAGs (consecutive) or in individual pLAGs (solo) based on whether five or more pLAGs were positioned in the same transcriptional direction in the *B. plantarii* ATCC 43733 genome (Table 1 and Supplementary Fig. 9). The pLAG-enriched regions chosen for mutagenesis were verified as pLAG-enriched regions in other *Burkholderia* plant pathogenic species by examining the genomes of *B. gladioli* ATCC 25417 and *B. glumae* AU6208.

The selected pLAGs coded for hypothetical proteins, transcriptional regulators or transmembrane proteins and were widespread and highly conserved in the plant pathogenic *Burkholderia* strains as per the LAGs definition (Fig. 6A and Supplementary Fig. 10). *B. plantarii* DSM 9509 was used as a model for experimental validation of the selected pLAGs (Supplementary Fig. 11). Site-directed mutants were generated for each of the 13 selected pLAGs and in vitro experiments

showed that 5 out of the 13 generated mutants were not affected in growth as compared to the wild-type strain (Supplementary Fig. 12); these were selected for in vivo virulence assays with rice seedlings (Fig. 6b and Supplementary Fig. 13). The results showed that virulence of mutants in all the 5 pLAGs was severely reduced in rice seedlings as compared to the disease symptoms caused by the wild-type strain confirming their status as true LAGs (Fig. 6c and Supplementary Fig. 13).

The genes encoded in the mutants ΩLAG14 and ΩLAG16 were part of a genomic region of 8 genes associated with plant pathogens (Supplementary Fig. 9). LAG14 is a putative extracellular solute-binding protein with a PF00497 sensor domain, which recognizes nicotinate, quidalnate, pyridine-2,5-dicarboxylate, and salicylate[66]. In *Bordetella pertussis*, interaction of nicotinate with the sensor kinase modulates the expression of certain virulence factors[67]. Therefore, we postulate that LAG14 may not be a virulence factor itself but affects regulation of other virulence factors in *Burkholderia*. LAG16 is a glycosyltransferase (GT), which can take part in the synthesis of polysaccharides, crucial components of the cell wall and biofilms[68]. A previous study showed that a knockout of a single glycosyltransferase resulted in a significant reduction in virulence in *Xanthomonas citri*, highlighting the potential role of GTs in virulence[69]. LAG22 is a small protein annotated as alpha/beta hydrolase with an undescribed DUF4180 domain with homologs present in other Gram-positive bacteria and encoded in a genomic region composed of five genes associated with plant pathogens (Supplementary Fig. 9). To date, however, none of these proteins have been functionally characterized. On the other hand, LAG11 encodes a hypothetical protein without any identified domains and is likely co-expressed in an operon with a gene encoding an M14 metallopeptidase. None of the adjacent genes were classified as LAGs (Supplementary Fig. 9).

Last, LAG23 encodes a homoserine dehydrogenase, an enzyme involved in conversion of L-aspartate-4-semialdehyde to L-homoserine. This is a critical step in the aspartate pathway for the biosynthesis of lysine, methionine, threonine, and isoleucine, and is involved in cellular functions such as cell wall biosynthesis, translation, and carbon metabolism[70]. The aspartate pathway is present exclusively in plants and microorganisms, not in mammals, making homoserine

**Table 1 | Selection of LAGs present in plant pathogenic *Burkholderia* for experimental validation**

| LAG number[a] | Selection criteria[b] | bacLIFE cluster | Gene name | NCBI annotation |
|---|---|---|---|---|
| ΩLAG1* | S | cluster_033461 | bpln_1g17870 | Hypothetical protein |
| ΩLAG2* | S | cluster_028676 | bpln_2g03080 | Hypothetical protein |
| ΩLAG3* | S | cluster_020609 | bpln_2g09480 | Transmembrane protein |
| ΩLAG6* | S | cluster_010044 | bpln_2g18880 | Threonyl/alanyl tRNA synthetase SAD |
| ΩLAG7* | S | cluster_012899 | bpln_2g19390 | Alpha/beta hydrolase family protein |
| ΩLAG8* | S | cluster_026410 | bpln_2g19500 | AraC family transcriptional regulator |
| ΩLAG9* | S | cluster_033043 | bpln_2g24500 | Glutathione S-transferase |
| ΩLAG11 | S | cluster_041460 | bpln_2g25710 | Hypothetical protein |
| ΩLAG14 | C | cluster_025698 | bpln_2g29370 | Extracellular solute-binding protein |
| ΩLAG16 | C | cluster_023880 | bpln_2g29390 | Glycosyl transferase |
| ΩLAG18* | S | cluster_017514 | bpln_RS27180 | Hypothetical protein |
| ΩLAG22 | C | cluster_027639 | bpln_2g24780 | Alpha/beta hydrolase |
| ΩLAG23 | S | cluster_026232 | bpln_2g03660 | Homoserine dehydrogenase |

[a]Asterisk indicates LAG mutants with a retarded growth in comparison with wild-type strain.
[b]S single LAGs, C consecutive LAGs.

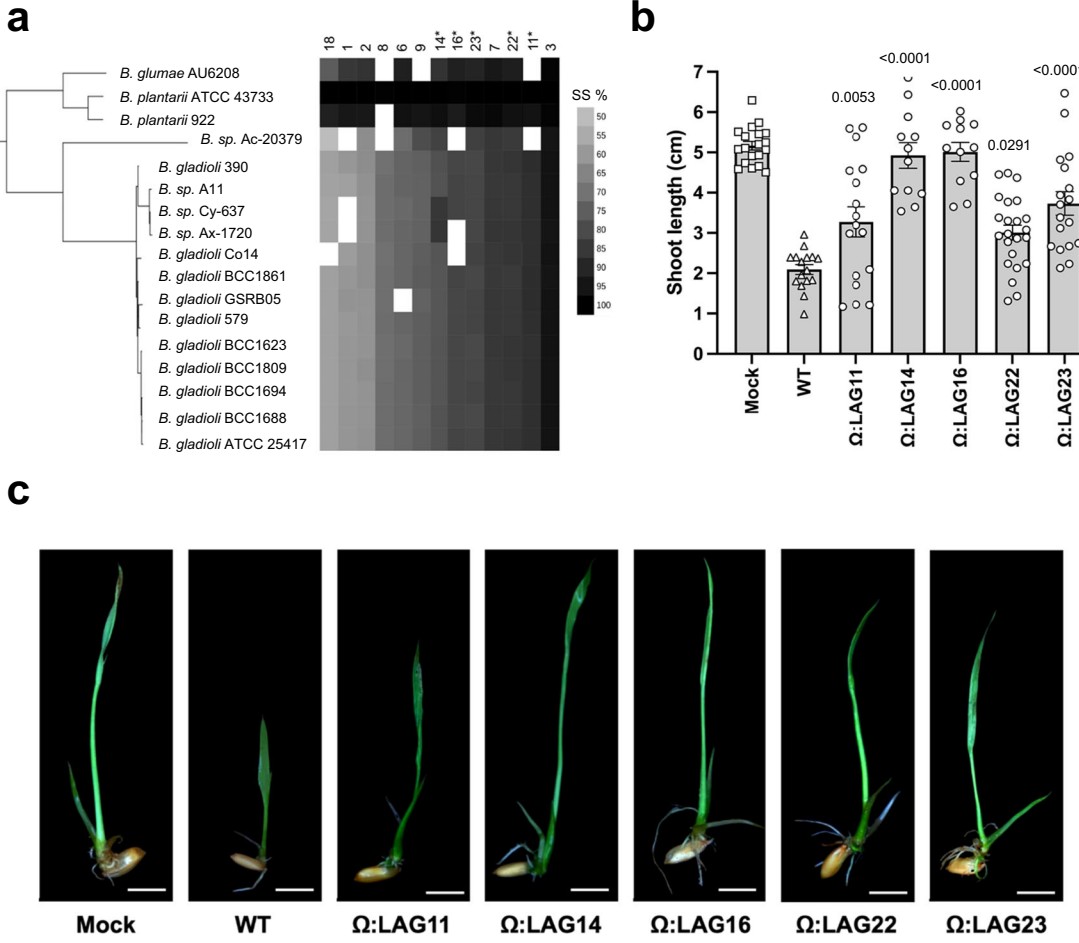

**Fig. 6 | Role in virulence of potential Lifestyle Associated Genes exclusively identified in *Burkholderia* plant pathogens. a** Phylogenetic analysis of the *Burkholderia* spp. ($n = 17$) associated with plant pathogenic lifestyle, based on concatenated alignment (2146 amino-acid positions) of 54 ubiquitously conserved proteins identified with PhyloPhlAn 3.0[102] and visualized/annotated using iTOL[106]. The phylogenetic tree is also coupled with species assignment and a sequence similarity (SS%) heatmap showing the distribution of the 13 selected LAGs. **b** *Oryza sativa* L. cv. Baldo seeds shoot length preinoculated with the wild-type strain (*B. plantarii* DSM 9509) and LAGs generated mutants ($OD_{600} = 0.5$). Error bars correspond to the standard deviation of at least 14 different seeds from two independent experiments. The mean serves as the measure to define the center of the error bars. Statistically significant differences were determined by one-way ANOVA ($P \le 0.05$) followed by post hoc Tukey test and relative to the wild-type strain. **c** Disease symptoms displayed by rice seedlings 10 days after infection with *B. plantarii* DSM 9509 and LAGs mutants.

dehydrogenase an attractive candidate for development of new pesticides or antibiotics with minimal effect in animals[71]. Previous work has shown that a null deletion of homoserine dehydrogenase gene in *Mycobacterium tuberculosis* H37Rv had a bactericidal effect, showing that inhibition of the aspartate pathway leads to elimination of chronic infections[72]. Similar to LAG11, none of the genes present in the genomic context of LAG23 were classified as LAGs (Supplementary Fig. 9).

To address the potential influence of phylogeny on pLAG selection, we examined the distribution and correlation of pLAGs chosen for mutagenesis and experimental validation (Fig. 6a). We found that three out of five LAGs compromised in virulence on rice seedlings (LAGs 14, 22, and 23) were entirely correlated with the phylogeny and present in all plant pathogenic *Burkholderia* genomes in the dataset. This observation raises an important point: excluding genes highly correlated with phylogeny may result in missing putative LAGs. However, it may also increase the number of predicted LAGs linked to phylogenetic noise. We compared our LAG selection method with other methods that take phylogeny into account such as phyloGLM and treeWAS (see methods). Regarding to the 8 pLAGs with reduced growth, we decided to not subject these mutants to plant experiments, as their reduction in virulence may be correlated with reduced bacterial growth given the interconnected relationship between colonization and pathogenicity. While there could be variations in colonization within plant tissues or among different plant species, the practical challenges associated with testing multiple species make such investigations beyond the scope of this study.

In summary, with our experiments we were able to validate the role of 5 new true LAGs identified by bacLIFE for their role in virulence of *Burkholderia*. bacLIFE was more efficient in the identification of genes involved in the virulence process when LAGs were present in LAG-enriched regions, 3 out of the 5 (LAG14, LAG16 and LAG22) were present in two genomic regions enriched in other LAGs, while the 2 other LAGs (LAG11, LAG23) were solos (Table 1 and Supplementary Fig. 9). Exploring genomic regions enriched in LAGs as a potential filtering approach holds promise in further refining LAG selection and identifying genuine lifestyle associations and could help in differentiate real LAGs from phylogenetic noise.

## Integrated secondary metabolite analysis in bacLIFE reveals known and unknown Biosynthetic Gene Clusters linked to a phytopathogenic lifestyle

Through the integration of antiSMASH[41] and BiG-SCAPE[42] in bacLIFE, we were able to study secondary metabolism of *Burkholderia/Paraburkholderia* and *Pseudomonas*. The antiSMASH results showed that clusters of BGCs or BiG-SCAPE Gene Cluster Families (GCFs) were mainly distributed and conserved among bacterial species or clades with conserved lifestyles (Supplementary Figs. 14, 15, and 16). In *Burkholderia/Paraburkholderia*, the secondary metabolite repertoire was composed of a high genomic abundance of genes encoding terpene, bacteriocin, and homoserine lactone synthetases[58,73,74] (Supplementary Fig. 17a). Within *Pseudomonas*, the most abundant class of BGCs were Non-Ribosomal Peptide Synthetase (NRPS) families, consistent with previous studies[75–77] (Supplementary Fig. 17a). NRPSs play a crucial role in the biosynthesis of important compounds in pseudomonads, including siderophores, lipopeptides, antibiotics, and virulence factors, contributing to their survival and pathogenicity[75–77]. When examining the average number of BGCs per genome across different lifestyles (Supplementary Fig. 17b, c), we observed a trend towards a higher number of NRPSs (Student's *t*-test α = 0.01) in plant pathogen genomes of both *Burkholderia/Paraburkholderia* and *Pseudomonas*. While this association is well-documented in *Pseudomonas*, limited information is available on the role of NRPSs in plant pathogenic *Burkholderia*[78].

Using the GCFs absence/presence matrix generated by the bacLIFE clustering module and applying the Fisher's exact test

($P < 0.01$ and >15% group presence), we were able to find GCFs statistically significantly associated with a plant pathogenic lifestyle in *Burkholderia/Paraburkholderia* and *Pseudomonas*, in a similar fashion as described above for the LAGs. In *Burkholderia* (Supplementary Datasets 7), we found multiple phytopathogenic BGC families associated with terpene and homoserine lactone synthesis, well-known virulence factors in *Burkholderia gladioli*[58]. In *Pseudomonas* (Supplementary Datasets 8), we found BGCs likely encoding the production of the well-studied syringofactin and syringomycin, two NRPS-encoded lipopeptides that are critical for pathogenicity[79,80] as well as two BGC families with an NRPS gene showing 41% and 40% similarity to the NRPS gene of the fragin BGC of *Burkholderia cenocepacia* H111 according to antiSMASH. Fragin is known as a metallophore with antifungal and antibacterial activity[81]. In addition, bacLIFE analysis resulted in the discovery of an unidentified and plant pathogen-associated Non-Ribosomal Peptide Synthetase – Polyketide Synthase (NRPS-PKS). NRPS-PKS hybrid compounds are relatively scarce in *Pseudomonas* (Supplementary Fig. 16b). However, a subset of these compounds, including syringolin, pyoluteorin, and coronatine, are recognized for their diverse functionalities, such as inducing chlorosis, acting as fungicides, or suppressing plant defense mechanisms[82].

The identified hybrid gene cluster was found in various plant pathogens, including *P. syringae* and *P. viridiflava*, among other *Pseudomonas* species (Fig. 7a, b, and Supplementary Datasets 8). A phylogenetic tree of *Pseudomonas* strains annotated as plant pathogen, combined with a BLAST search (*e*-value = 0.01) and structure comparison based on antiSMASH outputs, demonstrated that this yet unknown NRPS-PKS gene follows a similar distribution to the BGCs of well-known virulence factors syringofactin and syringomycin BGC families within the *P. syringae* species while being highly conserved also in *P. viridiflava* (Fig. 7b). The genes present in this NRPS-PKS cluster were highly conserved, while the flanking regions were variable (Fig. 7c). Based on antiSMASH prediction, the NRPS-PKS encodes a peptide composed of the amino acids Arginine-Proline-Cysteine-X-X-Proline, although the compound and its structural composition remain yet unknown (Fig. 7d). NRPS-PKS typically produce a diverse group of natural products with complex chemical structures and pharmaceutical potential. Through comparative sequence analysis, we identified a significant match for the longest core biosynthetic gene and a flanking gene with amino-acid sequence identity of 49% and 58%, respectively, to the WP_004571777.1 and WP_004571774.1 proteins encoded in the rimosamide biosynthetic gene cluster of *Streptomyces rimosus subsp. rimosus* (BGC0001760), as retrieved from the MIBiG[83] database. Rimosamides and detoxins are rare secondary metabolites that exhibit protective anti-antibiotic activities. These compounds are synthesized on modular NRPS-PKS enzyme complexes following a conserved thiotemplate mechanism[84]. To study the functional role of this novel BGC in virulence, a mutant in the first core biosynthetic gene was generated in *P. syringae* pv. phaseolicola 1448A (Pph ΩNRPS) and tested in in vivo experiments with bean. Mutant Pph ΩNRPS strain was not affected in growth (Supplementary Fig. 12) but showed reduced virulence (halo blight) as compared to the wild-type strain at 4 and 7 days post-inoculation (dpi) (Fig. 7e). During the infection process in the tissue, water-soaked spots are produced on leaves, which starts with a chlorotic process characterized by the development of yellowing areas. Next, damages quickly expand to necrosis, which produces the formation of darkened lesions, causing damage to the plant tissue. Image analysis of infected leaves was performed to determine the percentage of necrotic and chlorotic lesions per infiltrated area (Fig. 7f and Supplementary Fig. 18). Necrotic lesions of the tissue induced by wild-type Pph 1448A covered approximately 15.57 ± 2.61% and 68.36 ± 13.66% of total infiltrated area at 4 and 7 dpi, respectively. In contrast, leaves inoculated with Pph ΩNRPS exhibited significantly fewer necrotic lesions than those caused by the wild-type strain, accounting for approximately 4.06 ± 2.07% and 19.55 ± 6.53% of the

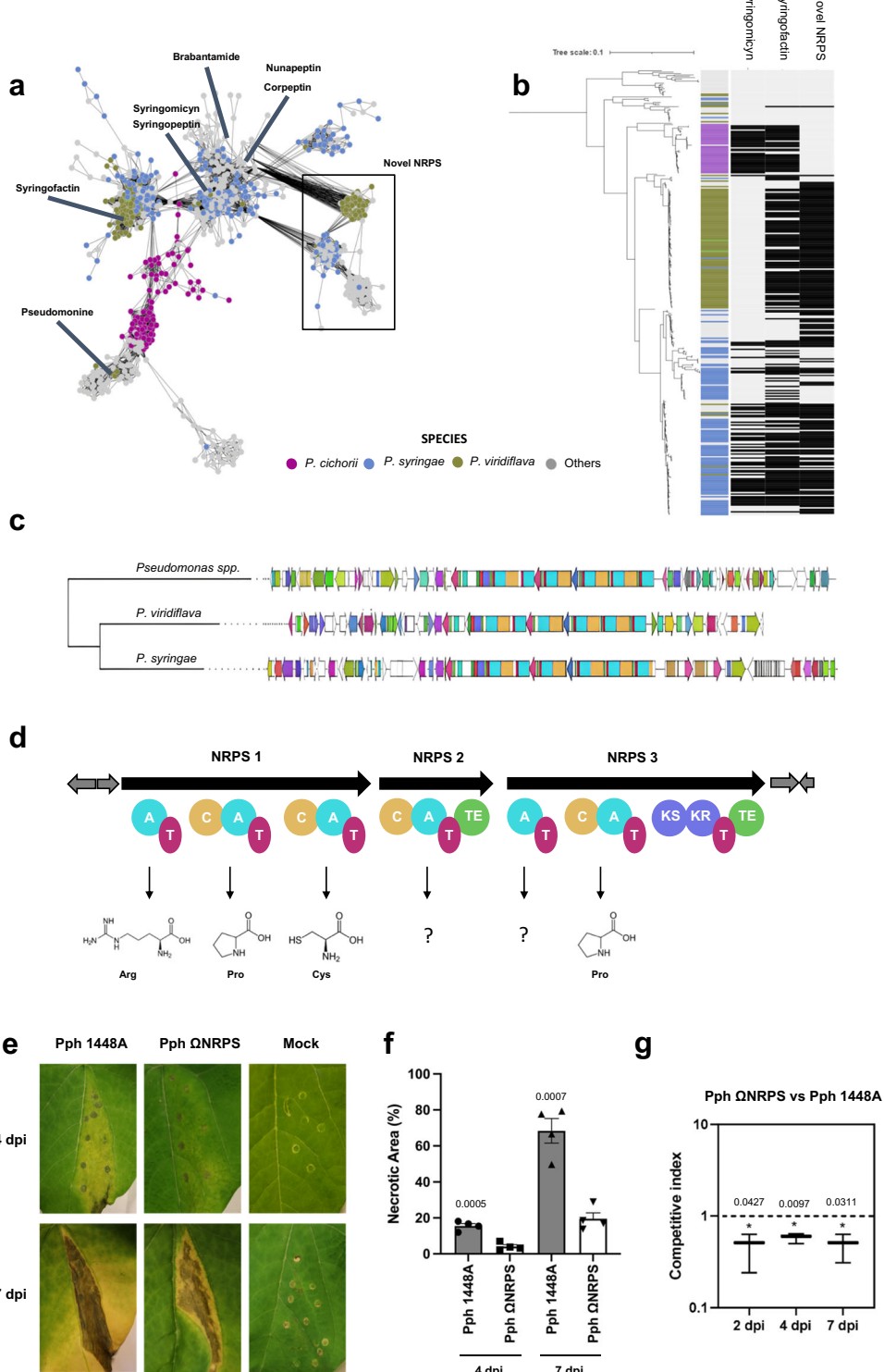

infiltrated area at 4 and 7 dpi, respectively (Fig. 7f). At 7 dpi, however, chlorosis of the tissue induced by Pph ΩNRPS mutant was significantly higher compared to the wild-type strain (31.17 ± 5.16% vs 19.58 ± 7.8%) (Supplementary Fig. 18). These results suggest a delay in symptom development upon leaf inoculation with Pph ΩNRPS. To further test the implications of this novel NRPS-PKS in virulence of Pph 1448A, competition assays between the wild-type strain and Pph ΩNRPS mutant were performed (Fig. 7g). The competitive index (CI) of the mutant relative to the wild-type strain showed that the wild-type strain was able to outcompete the mutant strain *in planta*, revealing that the

wild-type strain had a competitive advantage over the mutant strain, as evidenced by the calculated CI values of 0.5. (Fig. 7g). Altogether these results show the involvement of this NRPS-PKS in the fine-tuned process of infection of Pph 1448A and that it is required for competitive growth and survival on bean leaves. We postulate that this newly identified NRPS-PKS gene cluster, which shows high conservation among various plant pathogenic *Pseudomonas*, codes for the production of specific compound(s) that either enhance pathogenicity or protect the pathogen against plant defense metabolites. To test this hypothesis, large-scale isolation and purification, structural

**Fig. 7 | A systematic analysis of the distribution and virulence role of a novel Non-Ribosomal Peptide Synthase (NRPS-PKS) from *Pseudomonas* plant pathogens. a** Subset Sequence similarity network of the NRPSs present in *Pseudomonas* constructed with BiG-SCAPE[42] using a threshold of 0.3. Nodes are representing the BGCs, each BGC is colored based on the *Pseudomonas* specie where the BGC was detected. The novel NRPS-PKS is present in three BGC clusters, two of them significantly assigned to the phytopathogenic bacteria *P. syringae* and *P. viridiflava* and one associated with environmental species such as *P. fluorescens* (highlighted with a black square). Well-studied NRPSs from the MIBiG database[83] involved in *Pseudomonas* plant pathogenesis such as syringofactin[80], syringopeptin[111], syringomycin[79], brabantamide[112], nunapeptin[113], corpeptin[114] and pseudomonine[115] are highlighted in the network. **b** Phylogenetic analysis of the plant pathogen *Pseudomonas* spp. (n = 332), harboring this NRPS family, based on concatenated alignment (11001 positions) 77 ubiquitously conserved proteins identified with PhyloPhlAn 3.0[102] and visualized/annotated using iTOL[106]. The phylogenetic tree is coupled with a colored strip showing the species and a heatmap indicating the distribution of syringomycin (syr), syringofactin (syf), and the novel NRPS. Light gray indicates the absence, whereas black indicates presence. This NRPS family is highly conserved in the phytopathogenic bacteria *P. syringae*, *P. viridiflava* having the same BGC product prediction by antiSMASH[41]. **c** CORASON[42] phylogenetic reconstruction of the novel NRPS using a random representative of

each of the three different clusters associated with this NRPS detected in the network shown in panel (**a**). Colors indicate the domains predicted for each gene. **d** Shown below the NRPS and PKS genes are the module and domain organizations of the encoded proteins. The domains are labeled as follows: C condensation, A adenylation, T thiolation, TE thioesterase, KS ketosynthase, and KR ketoreductase. Predicted substrates of the NRPS and PKS modules in novel NRPS-PKS are arginine (Arg), proline (Pro), cysteine (Cys), 2 unknown amino acids, and, again, cysteine. **e** Symptoms induced by *P. syringae* pv. phaseolica 1448A (Pph 1448A) and Pph NRPS mutant in bean leaves (*Phaseolus vulgaris* cv. Canadian Wonder) at 4 dpi and 7 dpi after inoculation with suspensions of $10^6$ CFU mL$^{-1}$. **f** Quantification of necrotic areas as a percentage of necrotic lesion per infiltrated area in the inoculated leaves shown in panel (**e**). Error bars correspond to the standard deviation of five different leaves from two independent experiments. The mean serves as the measure to define the center of the error bars. Asterisks indicate significantly differences between wild-type and mutant strains using Student's *t*-test two-sided ($P \le 0.05$). **g** Competitive index (CI) assay in leaves of beans cv. Canadian Wonder after inoculation of mixed suspensions of wild-type strain and NRPS mutant strain at $10^4$ CFU mL$^{-1}$. Error bars correspond to the standard deviation of three biological replicates. The mean serves as the measure to define the center of the error bars. Asterisks indicate indices that deviate significantly from unity using Student's *t*-test two-sided ($P < 0.05$).

identification, extensive bioassays, and mode-of-action experiments will be required.

In summary, the comprehensive computational workflow presented here as bacLIFE enabled large-scale comparative genomics, prediction of lifestyles, and genes associated with the predicted lifestyle. Additionally, the interactive and intuitive user interface we developed enables comprehensive investigation and visualization of these advanced outputs. In this study, we have shown the potential of bacLIFE to predict bacterial lifestyle using machine learning algorithms trained with genetic signature distributions. We also demonstrated the accuracy of bacLIFE in the prediction of known and unknown LAGs, several of which were validated experimentally as true LAGs. Using *B. plantarii* DSM 9509 as a model, site-directed mutagenesis and plant assays revealed 5 selected LAGs involved in virulence. Importantly, bacLIFE was more efficient in LAG prediction in *B. plantarii* when these were present in regions enriched in LAGs. bacLIFE also led to the discovery of a novel NRPS-PKS gene cluster in plant pathogenic *Pseudomonas* species. Subsequent mutagenesis of this gene cluster in *P. syringae* pv. phaseolicola 1448A showed a reduced virulence in bean, reinforcing the potential of bacLIFE as a valuable tool for the discovery of new virulence-associated genes. While the current bacLIFE pipeline is tailored for prokaryotic genomes, the framework can be extended to eukaryotic organisms, in particular fungi and yeasts. bacLIFE will not only advance our understanding of the functions and evolution of genes associated with a specific lifestyle, but also provide a valuable resource for risk assessment studies of specific microorganisms or synthetic microbial communities targeted for agronomic or environmental applications.

## Methods

We developed a bioinformatic pipeline for comparative genomics that provides an easy and interactive way of exploring genomes and their lifestyles. This pipeline consists of three main different modules (Fig. 1): the first one is a Snakemake-based[37] workflow that processes, annotates and generates the data in the correct format; the second is a lifestyle prediction module that allows to test the predictability of the metadata and uses the trained machine learning models for prediction in genomes with poor metadata; the third is a User Interface Application module meant for interactive exploration and analysis of the data obtained in the Snakemake module. The first module is a Python-based system which allows to organize and execute different steps or rules that must be run in a linux environment, preferably a server. All the software needed is called by the Snakefile file and automatically

executed in the terminal. More information about the different pipeline steps and a user guideline can be found in the pipeline documentation file (https://github.com/Carrion-lab/bacLIFE). Although bacLIFE is designed for big datasets, it can be used for small datasets with few genomes if there is a very strong association between genes and lifestyles; of course, as in all such statistical analyses, the more genomic observations are available in the input data, the more power the analysis will have. Crucial for the success rate is to have an approximate minimum of 10 genomes per lifestyle in the comparative analysis to have reliable statistics. bacLIFE offers versatility, accommodating various taxonomic levels, including genera, families, and orders within its comparative analysis framework. The choice of taxonomic level depends on the research objectives. In our study, we focused on the genus level to investigate bacterial lifestyles effectively, particularly within groups like *Pseudomonas* and *Burkholderia*. This level strikes a balance between detail and inclusivity, enabling us to capture meaningful lifestyle variations while managing dataset size. The gene clustering in bacLIFE is responsive to the characteristics of the input dataset. When employing a dataset with a higher taxonomic rank, resulting in increased sequence divergence, the gene clustering tends to become more general. Consequently, this may lead to a loss of resolution in lower taxonomic levels within the dataset.

### Genomes and data retrieval

Two case studies were selected to test the accuracy and limitations of bacLIFE, each of them with genome datasets corresponding to different bacterial genera: *Burkholderia*/*Paraburkholderia* and *Pseudomonas* (Supplementary Datasets 1). All RefSeq assemblies were downloaded from the NCBI using custom scripts present in the GitHub documentation. To reduce redundancy in our datasets, we clustered genomes at 0.99 Mash-estimated[85] Average Nucleotide Identity (ANI) distances and then randomly picked one genome as the representative for the analysis. As a part of our study, we have introduced an optional feature in bacLIFE that involves incorporating code to enable users to download and cluster genomes based on an ANI value that is specified by the user. This process involved utilizing Structured Query Language (SQL) queries sourced from the National Center for Biotechnology Information (NCBI) Reference Sequence database[86]. This feature of bacLIFE empowers users to perform genome redundancy reduction similar to the approach detailed in our study.

After removing redundant genomes, we used CheckM[87] quality analysis to only retain high-quality genomes (>90% completeness and <5% contamination) (Supplementary Fig. 19). Given the high

completeness levels in our dataset, averaging at 99.64%, the likelihood of genes being genuinely absent due to technical issues is minimal. According to our data, we can estimate that, on average, there are roughly 31 missing genes per *Burkholderia/Paraburkholderia* genome (out of 6995 genes/genome) and approximately 24 missing genes per *Pseudomonas* genome (out of 5330 genes/genome). We also assessed the quality of the genomes with regard to genome size, GC%, number of predicted genes, number of contigs, and N50 (Supplementary Fig. 20). Note that this quality analysis is not part of the bacLIFE framework.

For each dataset, metadata information was collected as an initial prediction of bacterial lifestyle. This was accomplished by conducting a literature review where each species was investigated, paying special attention to reports of infection and information about the isolated source of bacteria from those species.

## Gene prediction and clustering

Gene prediction was performed using Prokka 1.14.6[88]. The predicted protein sequence files from Prokka are extracted and combined in a single fasta file. For the sequence clustering step, the file containing all of the amino-acid sequences from the genes is utilized as input. Gene clustering is performed using a combination of linclust[39] and the Markov clustering algorithm[38] (Supplementary Fig. 1). Linclust is a k-mer- and alignment-based algorithm from the MMseqs2 software suite[39]. We used this algorithm to cluster all sequences that are >90% identical with a minimum coverage of 80% (parameters --min-seq-id 0.9 --cov-mode 0 -c 0.8). This first clustering step helps us reduce redundancy and reduce the number of highly similar genes in the Markov clustering, which is computationally expensive. The representatives of the linclust clusters were aligned to each other with a blast all vs all using diamond[89] with an e-value cutoff of $1e^{-5}$. The output of diamond was formatted into a network-style file where the first two columns contain the nodes or genes and the third column represents the diamond bitscore.

To optimize the inflation value parameter for our objective, we run the MCL clustering in two small datasets of 8 *Paraburkholderia/Burkholderia* and 10 *Pseudomonas* respectively (Supplementary Datasets 9) with an inflation value of 1.1, 1.5, 2.0, 2.5, 3.0, 3.5 and 4.0. Following a similar validation as in the OrthoMCL paper[90], we analyzed the consistency of the MCL clusters with respect to enzyme commission (EC) numbers of the ENZYME database[91]. To find the optimal inflation value for our dataset, we looked at the proportion of gene clusters with at least two members with an EC number with all its members being the same EC. This tells us about the consistency of the gene clusters. The proportion of EC numbers that are in more than one gene cluster indicates to us the granularity of the groups. We evaluated the consistency and granularity of our method in comparison with OrthoFinder[40] in default settings to benchmark our method and provide guidance for hyperparameter adjustment. Plots along the inflation value increase (Supplementary Fig. 21) show an increase in the consistency and granularity. We decided to use an inflation value of 3.0, which gives us similar consistency as OrthoFinder using default settings and lower granularity in both *Burkholderia/Paraburkholderia* and *Pseudomonas* datasets. The longest sequence of the cluster is chosen as the representative of the cluster for annotation purposes. The inflation parameter for protein clustering in bacLIFE was tuned and benchmarked for our specific datasets. However, it is important to note that the ideal choice of the inflation parameter may vary depending on the user's dataset, research question, and phylogenetic scope. Currently, bacLIFE does not incorporate an automated mechanism for users to fine-tune the inflation parameter specifically for their datasets within its framework and it defaults to a value of 3.

The non-redundant genomes were used as input for the bacLIFE clustering module, which predicted a total of 5,968,939 (6995 genes/bacteria) and 10,928,452 (5330 genes/bacteria) gene sequences and grouped into 157,696 and 191,300 gene clusters in *Burkholderia/Paraburkholderia* and *Pseudomonas*, respectively. Noteworthily, 49% and 54% of the gene clusters were present in only one strain (unique gene clusters) of *Burkholderia/Paraburkholderia* and *Pseudomonas*, respectively. It is worth noting that the evolutionary origins of these unique genes can differ, potentially arising through various mechanisms including horizontal gene transfer facilitated by viruses, as well as conjugation and mutations/rearrangements within bacteria that result in the emergence of de novo genes[92]. There is also a possibility that these unique genes lacked sufficient sequence similarity with other genes to be part of a gene cluster with multiple sequences. We established the core-genome by identifying genes present in over 90% of the genomes. Our analysis yielded 2512 core gene clusters for *Burkholderia/Paraburkholderia* and 1951 for *Pseudomonas*. These findings align with prior pan-genome[46], where 1645 soft-core genes, present in 95% of *Pseudomonas* genomes, were identified. The core-genome incorporates important genes in maintaining essential cellular functions that are under strong, purifying genetic selection.

## Gene functional annotation

Annotation of the representative sequence of each cluster is performed using different databases and tools, such as DBCAN[92], PFAM[93], KEGG[94], and COG[45]. DBCAN[92] and PFAM[93] are based on hidden Markov models (HMM). Hmmsearch from hmmer 3.1[95] annotation is used for this purpose. KEGG[94] and COG[45] annotations are obtained using the sequence alignment-based tool eggNOG-mapper 2.1.2[95] against the eggNOG 5.0 database[96]. Prokka description of the representative sequence of each cluster is extracted from the Prokka annotations using a custom Python script. An absence/presence matrix of the gene clusters in the input genomes is created using a custom R script. The absence/presence matrix is combined with the annotations and descriptions of each gene cluster acquired using custom R scripts. This produces one of the Snakemake module's main outputs.

## Biosynthetic Gene Cluster prediction and clustering

Biosynthetic Gene Clusters (BGCs) are predicted using the tool antiSMASH 5.1.2[41]. antiSMASH is executed using the.gbk files generated by Prokka annotation as input and the following arguments: --cb-general --cb-knownclusters and --cb-subclusters. BGCs are grouped in Gene Cluster Families (GCFs) using BiG-SCAPE 1.1.5[42]. Using a set of custom scripts, an absence/presence matrix of the individual GCFs at 0.70 BiG-SCAPE distance in the input genomes is generated. BGC networks were created using cytoscape 3.8.2[97]. A total of 12,474 BGCs (~14 BGCs/bacteria) and 19,087 BGCs (~9 BGCs/bacteria) were predicted for *Burkholderia/Paraburkholderia* and *Pseudomonas*, respectively. BiG-SCAPE clustered the BGCs and reduced the complexity of the analysis down to a total of 856 and 1279 GCFs (Gene Cluster Families) in *Burkholderia/Paraburkholderia* and *Pseudomonas*, respectively.

## Lifestyle prediction

Lifestyle prediction (Supplementary Fig. 2) accuracy was tested and evaluated with ROC plots and areas under the curve (AUC) with the predictions of random forest and nearest neighbor as baseline. We built multiple binary classifiers for each lifestyle assigned based on metadata from literature (Supplementary Datasets 10). Because of the high number of variables in the absence/presence matrices, a chi2 feature selection with scikit-learn 1.0.2[98] is performed before the prediction evaluation of each lifestyle. The top 1000 important features are selected and used in a 5-fold cross-validation scheme (70% training, 30% test) with random forest (n_estimator = 50) and nearest neighbor classifiers (n_neighbors = 1 and metric = 'jaccard') using scikit-learn 1.0.2[98]. AUC values are calculated as the mean of the 5 cross-validated tests AUCs for each lifestyle and genera. Furthermore, we utilized a leave one species out validation approach where we trained the random forest model without including any *B. vietnamiensis, P.*

*fluorescens*, and *P. cichorii* genomes in the *Burkholderia/Para-burkholderia* and *Pseudomonas* datasets, respectively. Subsequently, we applied this trained model to predict the lifestyle of the omitted genomes and examined the probability confidence values associated with their true lifestyles.

After accuracy testing, the random forest algorithm was used to predict the lifestyle of bacteria labeled with an unknown lifestyle. A lifestyle is selected if the random forest adjudication has a probability confidence value of >0.8. This means that the fraction of trees in a forest that voted for a certain class is bigger than 80%. This allows us to increase the lifestyle information with high confidence in the predictions. Confidence values for all genomes can be found in Supplementary Datasets 2. In certain instances, such as with *Pseudomonas fluorescens* GCF.902497735.1 and *Pseudomonas fluorescens* GCF.902497865.1, initially classified as environmental based on species, the classifier expressed confidence scores of 0.668 for them being environmental and 0.368 for them being plant pathogens. Upon examining these genomes using bacLIFE PCoA, it becomes evident that they deviate significantly from other *P. fluorescens* spp. genomes and are closely grouped with *P. syringae* spp. These genomes, marked by uncertain prediction values, underscore the challenges of relying solely on species-level annotation and prediction when a genome stands as an outlier within its designated species.

As a complementary approach to the random forest lifestyle predictor, other unsupervised data-based approaches, such as HDBSCAN[99] (methods included in the app), were employed for statistics purposes. The unsupervised methods incorporated into the bacLIFE app prove valuable for grouping bacteria with similar gene compositions. This is achieved through clustering based on the PCoA, which is constructed using the gene absence/presence table.

## Explorative and statistical analysis

A principal coordinate analysis (PCoA) based on the Sørensen–Dice dissimilarity is performed in order to explore the absence/presence matrices generated by the bacLIFE clustering module. Sørensen–Dice places double weight on the intersection in the numerator, subtly emphasizing commonalities (which are of greater interest to us) between sets over differences as the more commonly used Jaccard distance does. R base function cmdscale() for PCoA was used. In order to visualize the PCoA results, the first two principal coordinates/components are plotted along with a density plot using ggplot2[100] and ggMarginal function from the ggExtra 0.10.0[101] package. We define and explore the core-genome looking for gene clusters that occur in >90% of the samples at the complete dataset and lifestyle level.

We define a predicted Lifestyle-Associated Gene (pLAG) as a gene that is frequently present in strains annotated with one lifestyle and frequently absent in the others. To obtain pLAGs we used the threshold of 70% of the genomes belonging to a lifestyle/group and more than a 2 log2 fold change between the lifestyle/group mean and the remaining bacteria mean presence. Log2fold change between lifestyle/group relative gene cluster presence is calculated using the foldchange() and foldchange2logratio() functions of the R package gtools 3.8.2[100]. Absence/presence matrices are used as input for the base R fisher.test() function to obtain the *p*-values of each gene cluster or GCF individually. P-adjusted values are calculated using the base R function p.adjust() with the Benjamini-Hochberg method. Our approach relies on the presence or absence patterns of genes, without considering the number of gene copies. It is important to note that certain genes may be linked to a particular lifestyle due to the number of copies they possess, and these associations may go unnoticed in our study. It is also worth noting that different bacterial clades may adapt to similar niches or lifestyles in distinct ways, involving the acquisition of different genes. These unique adaptations, while crucial for their respective ecological niches, may not be detected by bacLIFE's default settings, which rely on a predefined threshold for gene cluster

inclusion. However, bacLIFE's analytical module offers a level of flexibility that empowers users to explore these nuances. Users can customize the percentage inclusion parameter to capture genes that might be exclusive to particular clades or subsets of strains within a lifestyle group. This feature allows for targeted investigations into the genetic variations associated with specific adaptations.

We used the threshold of >70% group presence and >2 log2fold change to define pLAGs and Fisher test for ranking purposes. Fisher test doesn't take into account the phylogenetic bias in the analysis so it is not a reliable method to define pLAGs if multiple clones of the same genome are present. The genome redundancy reduction at 0.99 ANI partially takes care of this by removing highly similar genomes. Other tools such as phyloglm[52] or TreeWAS[53] are designed to address the issue of phylogenetic structure and its potential impact on gene presence/absence patterns in bacterial genomes. We conducted a comparison with results obtained using TreeWAS and phyloglm (with a significance threshold of *P* < 0.05) (Supplementary Fig. 22). Phyloglm, being more restrictive, identifies a lower number of plant pathogen LAGs, specifically 31 and 71 plant pathogen pLAGs in *Burkholderia* and *Pseudomonas*, respectively. In the case of *Burkholderia*, all 31 phyloglm-identified pLAGs also align with the results from TreeWAS and bacLIFE. However, in *Pseudomonas*, only 15 out of the 71 phyloglm-predicted LAGs are detected by all three methods, with 44 pLAGs not overlapping with either TreeWAS or bacLIFE. On the other hand, TreeWAS appears to be less restrictive, yielding a larger number of pLAGs compared to phyloglm and bacLIFE. However, it is worth noting that the overlap between bacLIFE (using the threshold of >70% group presence and >2 log2fold change) and TreeWAS is substantial, with 100% agreement in the *Burkholderia* dataset and 97% in *Pseudomonas*. This suggests that, while there are differences between existing methods, the threshold we used in bacLIFE for these specific datasets aligns closely with the outcomes from TreeWAS, albeit being slightly more restrictive. Additionally we extracted the importance of the features from the random forest classifiers, this way we can know which gene clusters are important to predict each lifestyle.

We compared important features (Gini importance > 0) selected by random forest for classifying plant pathogen genomes in *Burkholderia* and *Pseudomonas* datasets, along with pLAGs (Supplementary Fig. 22). In the *Pseudomonas* dataset, there was a substantial 95% overlap between bacLIFE-selected pLAGs (criteria: >70% presence, >2 logfold2change) and random forest's important features. This suggests strong agreement between the two methods. In contrast, the *Burkholderia/Paraburkholderia* dataset showed only a 47% overlap, likely due to random forest's handling of redundant features, which is more pronounced in datasets with limited plant pathogen genomes, like *Burkholderia*. In such cases, many gene clusters may exhibit similar distribution patterns in comparisons between plant pathogens and non-plant pathogens. Consequently, the model may consider only one of these similar features as important, while the others are omitted.

## Phylogenetic analysis

Species phylogenies under maximum likelihood were inferred with the software PhyloPhlAn 3.0.2[102] using MAFFT[103] and RAxML[104] for the alignment and maximum likelihood phylogeny inference respectively. The markers used for the multisequence alignment are the collection of proteins from AMPHORA2[105]. In the *Burkholderia/Paraburkholderia* dataset, the phylogeny was constructed using a concatenated alignment of 54 ubiquitously conserved proteins, comprising 2146 amino-acid positions. In the case of *Pseudomonas*, the phylogeny was inferred from a concatenated alignment that encompasses 11001 positions and includes 77 ubiquitously conserved proteins. Obtained trees are annotated with iTOL[106] in order to generate figures. Annotations of the presence of different genes studied in the members of the tree are generated using BLASTP with a minimum subject cover of 70% and a minimum sequence similarity of 40%.

## bacLIFE shiny app

An interactive Shiny app 1.7.4[107] was created in order to offer users the capability to perform a similar analysis as we did in this work. Users can employ Principal Coordinate Analysis (PCoA) and Permanova tests to discern group dissimilarities within the dataset. Additionally, bacLIFE app seamlessly incorporates the antiSMASH user interface, allowing for in-depth exploration of each bacterium's secondary metabolism. Exploration of pan genomes, core genomes, and unique genes is also supported within the app. bacLIFE's app primary feature is its capacity to identify genes associated with specific groups or lifestyles. It generates an interactive table featuring Fisher test p-values and the relative abundance of each gene cluster in each group. Users have the option to select a bacterium and generate a similar table, focusing solely on the gene clusters present in that bacterium, along with their genomic positions. This feature aids users in identifying consecutively conserved genes within a group or lifestyle. Additional functionalities of the app encompass the ability to conduct BLAST searches against the dataset for amino-acid sequences, visualize the distribution of bacLIFE gene clusters, and utilize Mash for genome mapping to the bacLIFE dataset based on Average Nucleotide Identity (ANI). All plots are created with ggplot2 3.4.0 and transformed to an interactive html widget with the function ggplotly() from the package plotly 4.9.3[108].

## Bacterial strains and growth conditions

Bacterial strains were grown at 28 °C (*Pseudomonas* strains), 30 °C (*Burkholderia* strains) or 37 °C (*Escherichia coli* strains) in lysogenic culture medium (LB)[109]. The bacterial strains used in this study are indicated in Supplementary Datasets 11. When necessary, culture media were supplemented with the corresponding antibiotic (Supplementary Datasets 11). Growth curve measurements were performed using a microplate reader (GENios, TECAN, Austria) and 96-well clear bottom microplates (Sarstedt, Germany) in LB medium supplemented with the corresponding antibiotic when necessary (Supplementary Datasets 11).

## Construction of bacterial strains and plasmids

For the construction of *Burkholderia* insertion mutant strains, the target gene of *B. plantarii* DSM 9509 was disrupted by insertion mutagenesis using the recombinant suicide vector pSHAFT2 (Supplementary Datasets 11). The target gene was amplified by PCR using Q5 DNA polymerase (New England Biolabs, Hitchin, UK) and the corresponding primers (Supplementary Datasets 11). PCR fragments and pSHAFT2 were digested using the restriction enzymes KpnI and EcoRI and ligated with T4 DNA ligase (NEB, Ipswich, MA). *Escherichia coli* CC118 λpir competent cells were transformed with the constructions and grown in LB medium containing chloramphenicol (10 µg mL$^{-1}$). The correct ligation was confirmed by PCR using GoTaq DNA polymerase (Promega, Madison, EEUU), pSHAFTFor2, and pSHAFTseqrev4 primers (Supplementary Datasets 11). The recombinant plasmid was introduced into *B. plantarii* DSM 9509 by electroporation. Overnight cultures of *B. plantarii* DSM 9509 were diluted to OD$_{600}$ = 0.1 and grown at 30 °C, 150 rpm for 2 h. Cells were then centrifuged at 9.400 rcf for 1 min at room temperature. The pellet was resuspended in 300 mM sucrose at room temperature and centrifuged again. This procedure was repeated twice. The cell pellet was suspended in 30 µl of 300mM sucrose and 500 ng of the plasmid was added. Electroporation was performed using ice-cold cuvettes (1 mm) and a Bio-Rad Gene Pulser II Electroporation System (1.3 kV, 10 µF, 600 Ohms). Then, 1 mL LB medium was added after electroporation. The cells were incubated at 30 °C for 3 h with shaking at 150 rpm and then plated on LB plates containing chloramphenicol (200 µg mL$^{-1}$). The correct insertion of the suicide vector was verified by PCR using pSHAFTFor2 and an upstream sequence of the target gene as primers (Supplementary Datasets 11).

For the construction of *Pseudomonas* insertional mutants the same procedure described above was followed. Target gene was amplified by PCR using GoTaq DNA polymerase and the corresponding primers (Supplementary Datasets 11). The fragment was A/T cloned into pCR2.1 and the correct ligation was confirmed by PCR. The recombinant plasmid was transformed by electroporation into the *P. syringae* pv. phaseolicola 1448A wild-type strain and plated in LB plates supplemented with Km (10 µg mL$^{-1}$). The correct insertion of the suicide vector was verified by PCR using M13-F and an upstream sequence of the target gene as primers (Supplementary Datasets 11).

## Plant virulence assays

Virulence assay of *Burkholderia* mutants were performed in rice seedlings (*Oryza sativa* L. cv. "Baldo"). The seeds were sterilized in 75% ethanol for 5 min followed by a 40% commercial hypochlorite solution for 30 min. Bacterial suspensions were prepared from an overnight culture and diluted to OD$_{600}$ = 0.5. For in vitro analysis, sterilized seeds were pre-germinated in the bacterial suspensions for 48 h at 30 °C. Pre-germinated seeds were transferred to Petri dishes containing sterile water and grown for 10 days at 28 °C, 80% humidity in a 16 h–8 h light-darkness cycle. After 10 days, pictures were taken and length was measured using ImageJ software. For *in planta* analysis, sterilized seeds were pre-germinated in MgSO$_4$ for 48 h at 30 °C and grown in a potting mixture of one part of commercial potting soil (Lensli Substrates, The Netherlands) and two parts of sand (Het Noorden BV, The Netherlands) at 28 °C, 80% humidity and in a 16 h light and 8 h darkness cycle. To test the virulence of *B. plantarii* and the mutants, 15 mL of a bacterial suspension adjusted to an OD$_{600nm}$ of 0.5 ($1 \times 10^8$ CFU gr soil$^{-1}$) were inoculated in the soil before sowing pregerminated seeds. Disease severity was visually scored on a scale of symptoms, moderate symptoms or no symptoms at 7, 14, and 21 days post-inoculation. Virulence assays of *P. syringae* pv. phaseolicola mutant was performed in bean plants (*Phaseolus vulgaris* cv. Canadian wonder). Eight-days-old bean plants, grown at 24 °C with a photoperiod cycle of 16 and 8 h, light and darkness, respectively. The bacterial suspensions for inoculation were adjusted to $10^4$ CFU mL$^{-1}$ or $10^6$ CFU mL$^{-1}$ and were infiltrated into bean unifoliate leaves using a 2-mL syringe without needle. After 4 and 7 days, pictures were taken. Quantification of necrotic and chlorotic lesions was performed using the image analysis software ImageJ. Data were represented as the percentage of necrotic and chlorotic lesions per infiltrated area. Competition assays were performed by mixing equal volumes cultures of Pph 1448A wild-type and Pph ΩNRPS mutant with an OD$_{600nm}$ = 0.5. Plants were inoculated with $5 \times 10^4$ CFU mL$^{-1}$ of the mixed bacterial suspension described above and bacteria were recovered from the infected leaves using a 12-mm diameter cork borer. Three discs per plant were homogenized by mechanical disruption into 1 mL of 10 mM MgCl$_2$ and then counted by plating serial dilutions on LB with and without Km. CI was determined by dividing the CFU ratio of Pph ΩNRPS mutant to Pph 1448A wild-type within the output sample by the ratio of mutant to wild-type within the input[110].

## Reporting summary

Further information on research design is available in the Nature Portfolio Reporting Summary linked to this article.

# Data availability

All genomes used in this study are publicly available at the RefSeq database and accessed through the NCBI Assembly (https://www.ncbi.nlm.nih.gov/assembly). A list of all genomes accession numbers of this study are available at Supplementary Datasets 1. Protein sequences of genes associated with host-pathogen interactions were downloaded from the PHI-base database (http://www.phi-base.org/).

## Code availability

All the codes for bacLIFE are stored in GitHub https://github.com/Carrion-lab/bacLIFE under the https://doi.org/10.5281/zenodo.10630333.

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

## Acknowledgements

The project was financially supported, in part, by the Spanish "Ministerio de Ciencia, Innovación y Universidades" project RYC2020-029240-I, by the "Ayuda G.1. a la Actividad Investigadora de beneficiarios de los programas Ramon y Cajal y Beatriz Galindo del II Plan Propio de la Universidad de Malaga" project 15 and by internal funding from Instituto de Hortofruticultura Subtropical y Mediterránea "La Mayora" (IHSM-UMA-CSIC) and the Institute of Biology Leiden. A.P. was supported by Margarita Salas Grant and co-funded by European Union-Next Generation EU and Ministerio de Universidades (Spain). The contribution of J.M.R. was supported by the Microp Gravitation project funded by NWO (grant number 024.004.14). The Contribution of F.M.C. was supported by PROYEXCEL21_00012, Junta de Andalucia. C.R. was supported by PID2020-115177RB-C21 from the MCIN/AEI/ERDF. This publication was supported by Proyecto QUAL21 012 IHSM, Consejería de Universidad, Investigación e Innovación, Junta de Andalucía. We thank people from the Carrión laboratory for their valuable feedback during group meetings. All analyses were performed in the computing cluster of the Netherlands Institute of Ecology (NIOO-KNAW).

## Author contributions

G.G.E. and V.J.C. conceived the idea. The study was designed by G.G.E., J.M.R, M.H.M. and V.J.C. The lab work was conducted by G.G.E., A.P., K.B. and V.J.C. Contributions to data analysis came from G.G.E, A.P., K.B., L.M.A.G., M.H.M., J.H.P and V.J.C. The manuscript was drafted by G.G.E, A.P., M.H.M., J.M.R. and V.J.C. Contribution with feedback in specific areas H.P.S., D.C., C.R., F.M.C. Lifestyle annotation was performed by G.G.E, A.P., K.B., L.M.A.G., H.P.S., D.C., C.R., F.M.C. and V.J.C. All authors contributed to the revision and agreed upon the final draft.

## Competing interests

The authors declare no competing interests.
