## [Peer Review File · Nature Communications]

bacLIFE: a user-friendly computational workflow for genome analysis and prediction of lifestyle-associated genes in bacteriaReviewer #1 (Remarks to the Author):

Summary:

The study presented a novel computational workflow named "microLife," designed for genome annotation, large-scale comparative genomics, taxonomic delineation, and identification of lifestyle-associated genes (LAGs). The workflow comprised three distinct modules: clustering, lifestyle prediction, and analytical modules.

In the clustering module, gene and Biosynthetic Gene Cluster (BGC) level prediction and clustering were performed. The lifestyle prediction module utilized metadata from the literature to assign lifestyles to bacteria with unknown lifestyles. Subsequently, the analytical module performed further statistical analysis to identify LAGs.

To evaluate the effectiveness of microLife, the authors conducted tests on two datasets pertaining to Burkholderia/Paraburkholderia and Pseudomonas genera. Focusing on the phytopathogenic lifestyle of both taxa, they functionally validated the LAGs identified by microLife. Notably, *B. plantarii* DSM 9509 exhibited enrichment of plant pathogen LAGs in virulence factors compared to all genes when subjected to BLAST analysis against the Pathogen Host Interactions database. The authors further selected 13 LAGs associated with the phytopathogenic lifestyle in Burkholderia, using *B. plantarii* DSM 9509 as a model for experimental validation. Through virulence assays conducted in rice, they successfully validated the role of five new LAGs identified by microLife in the virulence of Burkholderia.

Additionally, microLife facilitated the discovery of a novel NRPS-PKS gene cluster in plant pathogenic *Pseudomonas* species. To validate this finding, the authors performed mutagenesis of the identified gene cluster in *P. syringae* pv. *phaseolicola* 1448A, revealing a reduced virulence in bean.

Overall, the study demonstrated the efficacy of the microLife workflow in efficiently annotating genomes, conducting comparative genomics, and identifying LAGs, thereby providing valuable insights into the phytopathogenic lifestyles of Burkholderia and Pseudomonas

I can tell that a lot of good work went into this paper. Here were some of my overall impressions:

1. Nice shiny app! Much better than the average database/site in the bioinformatic world
2. Very good experiments, controls, and interpretation of results. Very good potential for development of antibacterial treatments for plants.
3. Very well organized supplemental tables, which is not a given.
4. Great benchmarking of the MCL hyperparameters. Detailed and sensical.

Overall, I think it's a good tool, and it would be interesting to see how other groups use it, and if they indeed find success identifying LAGs as well. It is worth publishing certainly but I do have some major concerns here, which I hope can mostly be answered with some clarification about methodology, and with some further explanation in the text. I do think that there are many points and questions I raise here that would fit in a discussion section (rather than a results+discussion combination section).

Major comments:

1. Line 202: Is *P. viridiflava* misclassified taxonomically? Unlikely given your high AUC values, no? If we are basing it on your classifier, isn't it more likely that substrains *P. viridiflava* have different lifestyle? Maybe be more clear on who is misclassifying here exactly? Do you mean the database where you got the taxonomy? The way it is worded a bit confusing

Anyways, looking at the bigger picture, I would overall downplay the taxonomic classification aspect of microLife. Taxonomy has a field full of arguments over the best method for classifying microbes, and many quite developed and sophisticated programs (e.g. <https://www.ncbi.nlm.nih.gov/pmc/articles/PMC6360808/>) that goes far beyond gene presence and absence. I think the paper is strong enough anyway without the taxonomy aspect, and it only adds a bit of confusion.

2. The random forest classifier uses the top 1000 important features based on the fisher exact test

(which I think in the code actually says top 5000, if I understood correctly? Please clarify). So it's pre-filtered for those genes that already show some variance between the groups. However the materials and methods are unclear. Is it the same 1000 features for both the lifestyle and the genus predictions? Because in this analysis the two are highly correlated it seems, so it's not exactly clear. It's important to make this distinction.

3. While reading the paper I was wondering what are the most important features in the random forest models? This will help understand everything better. You can find out and highlight those. (https://scikit-learn.org/stable/auto_examples/ensemble/plot_forest_importances.html). Again and again I asked myself the question: Why even do the Fisher test again independently if you have the fisher-based features? And you have them ranked further by the model! I think you could have used the trained model along with presence and the 70% presence in genomes etc. I think ignoring these features is a mistake, and it's interesting to at least look and see if the LAGs from the short list are in there.

4. The statistical analysis to define a LAG as a gene that is frequently present in strains annotated with one lifestyle, is based on Fisher's exact test (Lines 579-581). This test looks solely at the enriched and depleted functions for each lifestyle. The phylogenetic lineage is not taken into account in this analysis, thus ignoring cases in which gene presence/absence patterns are correlated with taxonomy, and the taxonomy is correlated with a lifestyle. In those cases, the gene may be a "passenger" in the genome. Tools like TreeWAS, Pyseer, Scoary, and others try to take this issue into account by correcting for phylogenetic structure. I think that looking for genes present in 70% of lifestyle X partially takes care of this idea, but if 100 genomes of lifestyle X are pretty much the same (very short branch lengths), then what is 70% presence but inflation of the counting?

The results regarding the secretion systems, and of course the experiments vindicate the methodology, but I think it's worth mentioning a bit more about how the "basic statistics" mentioned on line 250 help deal with this issue. Otherwise, it seems that you predicted lifestyle based on taxonomy, and then do enrichment analysis based on these lifestyles, which could then identify mostly taxonomic related genes that separated the groups in the first place. Gene profiles predict taxonomy AND can predict lifestyle. Again, the experiments look good, and perhaps more explanation about the features could help (are core genes filtered out? Etc. How are the features different in each model? If they are different this will help). But, this is another reason I would maybe drop the taxonomic analysis from this paper, so you don't have to address how hard it is to distinguish between (1) correlations of gene profiles ~ taxonomy, with (2) correlations of gene profiles ~ lifestyle.

5. On a related note, Figure 3 shows a massive correlation between genus and lifestyle, and looks "cleaner" than Extended figure 8. But actually Extended figure 8 is kind of more convincing, because it shows the messy reality of the fact that core genome and lifestyle are not necessarily correlated. Figure 3 for example has a strip of highly related plant pathogen genomes, with similar cog profiles. Are these not just sort of clonal expansions? It actually vindicates your methodology more that you find genes correlated with lifestyle that are NOT correlated with phylogeny. Anyway, Figure 3 is very busy, and really impressive that you have that many "dimensions" like colors, sizes, graphs etc. all in one figure. But I wonder what the added value is. It's hard to really tell anything about it. There are no Y axes for the values of the COG and # CDS barplots. Also, how can the COG bar plots represent COG abundances if the values seem to be negative sometimes?

6. It would be interesting to know if there are LAGs which are shared between Burkholderia and Pseudomonas.

7. Extended figure 7: You look at all COG categories except category Y, nuclear structure, for reasons that prokaryotes have no nucleus (although it's in the figure legend, but that is a minor point; another minor point, Kruskal is misspelled in the legend). However you keep category A and category Z, which (at least originally) are not present in prokaryotes either. Please confirm that this is indeed the case and then remove them. This may explain the very low numbers you are getting as opposed to the other categories. Also, you are getting massively significant differences (e.g. 9.7×10^{-134} for category Z in Pseudomonas) in these possibly meaningless categories, which

brings up questions about the meaning of these statistics as a whole... Please think about this and confirm that you believe the other results.

8. Lines 480-483: In regard to the assignment of lifestyles to bacterial genomes, the authors have based their classifications on existing literature for each species. However, it is important to consider the possibility that certain strains within the same species might exhibit different lifestyles. This is a key point which needs to be addressed. How exactly did you define the lifestyles? Do you have heterogeneity of lifestyle in one 99% ANI group?

9. Given the 90% completeness cutoff, and 99% ANI, you might have a gene cluster in 98% of genes, and missing in 2% for technical reasons. This might produce a fisher enrichment where there is none. How do you deal with this? Do you remove those gene clusters are all "1", i.e. genes present in all genomes? What about a gene cluster with 95% "1" labels? Extended figure 20 makes me more relaxed, but it would be good to include a graph of completeness as well so we know how many times a gene might be "missing" when it's not really missing.

10. Line 188-189. Lifestyle switching / flexibility is the charitable interpretation of mixed clusters. Although I think it is true that biology is too complicated to really put into simple lifestyles, it is worth looking deeper into here. Just to be sure, perhaps looking into the third principle coordinate could potentially separate between these mixed clusters?

Minor comments:

1. Line 34: As *a* proof of concept,
2. Line 37: add ",respectively" at the end.
3. I understand that you want this to be a general tool, but mentioning LAGs in the abstract and then not giving some examples (pathogen vs. non-pathogen or something) is a bit too unspecific. I understand you don't want to mention plant pathogenicity too deeply since it's a general tool but some hint as to which kinds of lifestyles microLife can analyze would be good.

4. I recommend that you add the github or site to the abstract

5. In Line 93, you mention "...tool like microLife", and then only in line 99 you introduce the term "...designated microLife"

6. Lines 182-188: you mention many species that are in the figure, but when looking at the figure, there is no way to tell which dots refer to these known species. Maybe provide the coordinates in the legend at the least.

7. Lines 185-189: The environmental Pseudomonas are not distributed in a large cluster as the author claim. They are distributed in multiple clusters (Figure 2b)

8. Figure 2 - mark the species names on the tree. The reader has to guess what the authors describe in the associated text.

9. Figure 2b is referred to in the text first, and then figure 2a.

10. Line 289: the -2 does not need to be superscript, and the E should be capital. Lines 293 and 296 also have strange formulations of the pvalue. Please confirm it's meant to be $0.1 * 10^{-6}$

11. Extended data 10: LAG 11 label is somewhat unreadable

12. Extended data 12: Why is the overall pipeline shown again? It's the third instance I think and you can just show the experimental design part. It's already in Fig 1, Ext Fig 3 (I understand why), but here it seems unnecessary. If it's meant to be different, I did not notice

13. Maybe mention the stages/characteristics expected process of color loss and leaf dying in one sentence for the non-plant science reader. microLife is meant to be for wider use so it would be good to define the jargon.

14. Line 268: I think it's important to mention that this idea is one hypothesis, and the other hypothesis is that the highly related species with the highly related lifestyles may simply have a shared genetic history or environment with a specific phage. In other words, these phage genes could be a "passenger" gene with no function. Without more details shown, it is hard to know. Are these lysozymes encoded next to the enriched LAGs? Maybe then you can come up with a hypothesis about their origin.

15. The logic of going from COG category U to T3SS is a bit weird to me. It seems only vaguely connected in the flow of the story of the paper. You have the secretion systems as a result in the LAG section anyway... I would structure the paper such that the LAG analysis is done first (paragraph starting at 249), then put figure 4. The connection between COG and T3SS is a bit weak, and honestly all the COG stuff can be removed. It raised more questions for me than answers. Your main result is the identification of LAGs, including secretion systems.

16. Regarding the comparison between genomes from different lifestyles, the authors have employed a method based on the presence and absence of specific functions. However, it is worth noting, at least in the Discussion section, that this approach does not consider the copy number variations of these functions within each genome. As copy number variations can significantly influence the functional potential and gene expression of organisms, their omission in the analysis might introduce certain limitations to the accuracy and comprehensiveness of the results (209-212). Furthermore, SNPs and other changes at a higher resolution may account for differences that cannot be seen with gene presence absence profiles. Comment on this and explain why your methodology is still advantageous.

17. There is no acknowledgement section. I guess there are some funding sources the authors would like to acknowledge.

Good luck

Reviewer #2 (Remarks to the Author):

The manuscript "From genomes to lifestyles: discovery and functional characterization of lifestyle" deals with the description of a new tool named MicroLife, which aims to relate microorganisms and genes/functions with lifestyles based on a classifier trained with trusted data. The authors validate the results through the exploration of *Paraburkholderia* and *Pseudomonas* genomes and the discovery of several genes related with plant pathogenicity. Both the bioinformatic tool and the specific findings on the putative pathogenic role of those few genes described to be related with that trait are of great interest. However, I consider that the metadata used to validate the tool is too biased to be useful for such purposes, which threaten the reliability of the results.

The initial classification of species into lifestyles is too biased for diverse reasons: (I) The number of species selected from literature is too narrow. (II) The authors did not classify properly the genomes into species through reliable methods (i.e.: ANIb, dDDH), and then, misclassified genomes such as those of *P. viridiflava* ('pathogens'), that were indeed within the *P. fluorescens* clade ('environmental'), were incorrectly used for the training of the random forest model. (III) The categorization of strains into lifestyles just based on taxonomy is also too biased. Some commensal strains can gain pathogenic or beneficial genes and change their host-performance^{1,2}. Also, some strains from the same species may have different lifestyles. It may be more correct to delineate the lifestyles based on the isolation source or based on validated traits for each strain, rather than based on the lifestyle of the most common one for the species. In any case, using the literature metadata of species, but not having taxonomically classified each genome reliably, is too risky. Also, the microhabitat can be biasing the life-style based analyses; for instance, it is not discussed whether the epiphytic vs rhizospheric (soil or root) habitat could affect the analyses rather than the effect of each strain within the host. Probably, the distribution of genomes in the PCoA can also be distinguished based on this type of microhabitats, not only based on the host

performance, since there may be more genes/functions needed to use the differential resources among microhabitats rather than the number of genes required to establish a beneficial or pathogenic outcomes. Hence, as an example, would it be possible that the genes found to be related with pathogenicity may not be related with this, but with the survival/fitness of the strain within the plant leaves?

Another important issue is the effect of phylogeny in the discovery of trait-associated genes. For instance, as shown in Extended Data Fig. 11, Burkholderia lifestyles are highly clade-specific. Hence, the discovery of lifestyle-associated genes might be biased to find clade-specific genes rather than those related with their lifestyle. In the case of Pseudomonas, for which their lifestyles are less associated with phylogeny, this bias could be reduced by using strain isolation data rather than extrapolating species metadata to all the strains named as that strain. In any case, in both cases, authors might use statistical methods that reduce this bias, such as PhyloGLM.

1. Li, E., de Jonge, R., Liu, C., Jiang, H., Friman, V. P., Pieterse, C. M., ... & Jousset, A. (2021). Rapid evolution of bacterial mutualism in the plant rhizosphere. *Nature Communications*, 12(1), 3829.

2. Drew, G. C., Stevens, E. J., & King, K. C. (2021). Microbial evolution and transitions along the parasite–mutualist continuum. *Nature Reviews Microbiology*, 19(10), 623–638.

Beyond those major concerns, I have some other comments that may be of interest:

- L 73-74. To date, the Pseudomonas genus encompass 316 valid species, not 144. I suggest updating reference 27 (year 2017) for some more recent work that revise the taxonomic status of this genus, such as some of these:

o <https://doi.org/10.3390/biology10080782>

o <https://doi.org/10.1099/ijs.0.064634-0>

o <https://doi.org/10.1016/j.syapm.2021.126289>

- L. 144 or 149: It could be useful to state how many genomes/strains belongs to Burkholderia and how many to Paraburkholderia

- L. 200-204. Please, confirm the taxonomy of these genomes through valid methods such as ANIb or dDDH values shared with type strains,

- Discussion (i.e. L. 218-226). I recommend providing more discussion about the genes found to be associated with some lifestyle. I.e.: detail some genes that not had never been related with such lifestyle, comparing the results with more works beyond that of Levy et al. (2018), etc. Here I provide some references that may be useful:

o Melnyk, R. A., Hossain, S. S., & Haney, C. H. (2019). Convergent gain and loss of genomic islands drive lifestyle changes in plant-associated Pseudomonas. *The ISME journal*, 13(6), 1575-1588.

o Saati-Santamaría, Z., Baroncelli, R., Rivas, R., & García-Fraile, P. (2022). Comparative genomics of the genus Pseudomonas reveals host-and environment-specific evolution. *Microbiology spectrum*, 10(6), e02370-22.

o Chewapreecha, C., Mather, A. E., Harris, S. R., Hunt, M., Holden, M. T., Chaichana, C., ... & Peacock, S. J. (2019). Genetic variation associated with infection and the environment in the accidental pathogen Burkholderia pseudomallei. *Communications Biology*, 2(1), 428.

- L. 251 and 253. It would be useful to add supplementary tables with the gene names or even with the gene/protein sequences of these LAGs.

- L 274-275. AHLs act also as beneficial molecules for the root colonization of plant-beneficial bacteria. Please, provide some discussion on why the LAGs related with AHLs are associated with pathogenicity but not with commensalism or mutualism.

- L. 288-297. I suggest to not only compare 1 genome/genus against the PHI-base to validate these results, but against more genomes.

- Fig. 6A. "...showing the distribution of the 15 selected LAGs. "13" instead of "15" ?

- Table 1. Please, provide the sequences (nt or aa) for these genes as a supplementary table.

- L. 361-370. You introduced the importance of NRPS systems in the Pseudomonas genus, but you are then focused on an hybrid NRPS-PKS biosynthetic gene cluster. Please, provide equal discussion on the importance of PKSs or even these hybrid clusters in Pseudomonas, if possible. Indeed, NRPS-PKS BGCs are not so common within this genus

(<https://doi.org/10.1099/mgen.0.000758>). The following article includes some data that may be useful for the discussion:

o Gross, H., & Loper, J. E. (2009). Genomics of secondary metabolite production by Pseudomonas spp. *Natural product reports*, 26(11), 1408-1446.

- Fig. 7f and Extended Data Fig. 19. What is the number of replicates (n) used for these

experiments? I recommend including these values within the figures or captions.

- L. 410. "7" instead of "seven"? Just to use the same writing style along the manuscript
- Extended data Fig. 20. It seems that there are several genomes too fragmented (500-2000 contigs), which threatens the GWAS analyses (some genes could be fragmented or absent). I recommend removing these genomes from the analysis.
- L. 485. Please, remove the detail of the citation ("Seemann, 2014") and just leave the numerical cite.
- L. 490. What is the rationale of using 90% of similarity for clustering sequences? Apart from this, usually, protein sequences rather than gene sequences are used for such clustering. As told in the lines 487-488, you used "gene" clustering. Please, just be sure that you aim to mention "gene" instead of "protein".
- L. 591-592. Please, detail the algorithm used (i.e.: Maximum Likelihood).
- L. 595. Why these thresholds? These are not the same than those used to create gene clusters.
- Results and discussion. Authors are providing a new pipeline to discover trait-associated genes, among other features. It would be useful for the readership to compare MicroLife with other workflows or tools (i.e.: Scoary, PhyloGLM, etc.) or at least discuss the pros and cons among them.

Reviewer #3 (Remarks to the Author):

Guerrero-Egido and co-authors introduce the bio-informatic pipeline microLife that aims to find genes associated with phenotypic traits within microbial taxa. MicroLife aids in downloading microbial genomes, reconstructs the pan-genome via clustering of homologous genes, and identifies genes and biosynthetic gene clusters that are associated with a user-defined phenotype using basic statistics. The authors call these genes 'Lifestyle Associated Genes (LAGs)'. In addition, a binary classifier is built to predict phenotype based on the pan-genome. MicroLife includes a point-and-click R Shiny web application that allows for graphical exploration of its results. In the manuscript, two microbial genera are analysed to identify plant-pathogenic-associated genes as an example use-case for microLife.

While none of the described methods are novel, the authors are upfront about this, and I do believe that there is a need for easy exploration and visualisation of the pan-genome aimed at researchers with very little bioinformatic experience. However, the validity of a pipeline like this depends on the soundness of the underlying methods and a comprehensive exploration of what it can and cannot do. I identify a number of major concerns with the bioinformatic methods and the way the manuscript is presenting these methods.

1. The basic promise of microLife is that genes that are associated with a certain phenotype can be identified with a Fisher exact test of the presence/absence profile in the pan-genome, and that this leads to a genetic understanding of the phenotype. However, genes may also be shared because of clade-specific common descent, and have nothing to do with the phenotype per se. The (Para)Burkholderia group is a good example, because the phenotype ('lifestyle') just follows the species tree (Fig. 3). Predictions based on genome content are trivial in this case, as shown by the high predictability of the classifier. Any gene that pops up is associated with the clade, but only a handful may be responsible for the phenotype, and microLife has no way of discerning them. The Pseudomonas case is more interesting because lifestyle and phylogeny are decoupled (Extended Data Fig. 8), but the classifier is doing worse, and I expect it to be bad at the 'difficult' cases exactly where phylogeny and phenotype are decoupled. An exploration of this dependency between phylogeny and lifestyle (potentially by contrasting the Burkholderia to the Pseudomonas case) is missing.

2. In relation to this, the manuscript does not discuss the pan-genome and how core and accessory genes may be involved in adaptation (and associated extensive literature), but this is

what it studies.

3. Similar pipelines exist, and are not cited or discussed. Scoary (10.1186/s13059-016-1108-8) comes to mind, whose underlying pan-genome prediction pipeline (based on Roary (10.1093/bioinformatics/btv421)) is arguably more advanced because it attempts orthology prediction as opposed to only clustering of homologous sequences, and has more statistical tests for gene associations to phenotype (including phylogeny correction, see previous comment). How do microLife's results of (Para)Burkholderia/Pseudomonas compare to Scoary, and what does it add that Scoary does not?

4. Again, given the strong correlation between phylogeny and lifestyle, I am not surprised that the classifiers are relatively good in predicting lifestyle based on genome content. It would be relevant to discuss where the classifiers go wrong, for example by plotting the misclassifications on the phylogenetic trees and PCoA. The classifier scores should be added to Supplementary Table 2.

5. MicroLife is aimed at researchers with little bioinformatic experience, however most of the analyses for the Burkholderia and Pseudomonas cases are as far as I can tell not output from microLife but require again bioinformatics expertise. For example, are the phylogenetic trees with iTOL annotations, Extended Data Fig. 7, and the BCG network, coming out of microLife? It seems that microLife was only used to identify the genes of interest (and aggregation of functional categories), which given that that this method is not novel or groundbreaking, reduces the significance of the tool for the findings presented.

6. The manuscript would benefit from a rewrite with a clear description of what the microLife R Shiny module can do, and how that relates to these findings.

7. It is unclear how the 13 + 1 LAGs that were selected for mutagenesis were chosen, out of 786 potential LAGs in total. Are they randomly selected, or is there cherry picking going on. If so, how? How is microLife contributing to this selection process?

8. A good example of coupled lifestyle and phylogeny (see first comment) may be the 8 genes whose mutagenesis resulted in impaired growth. These genes may not be related to lifestyle but instead to living.

I have put my next concerns in more or less chronological order of the manuscript:

9. The title says "discovery and functional characterization of LAGs", however I do not agree that the functional characterization is the take-home-message, and it largely oversells the pipeline as if it is a silver bullet to genotype-phenotype association. I would suggest focusing on the user-friendliness of aggregating multiple commonly-used tools in a point-and-click interface, which is its main benefit.

10. In the introduction the authors talk about beneficial or detrimental effects to the host, however they focus in subsequent analyses only on pathogens. Would microLife also be appropriate in identifying genes associated with beneficial effects, why/why not? I can see how a pathogenic lifestyle is an absolute best-case scenario for association studies.

11. The introduction would benefit from a discussion of microbial niches and phenotype, given that microLife can be potentially used for any type of association. Currently the introduction reads as if 'making another organism sick' is the niche of a microbe, but microbial niches are of course much more diverse.

12. What do you need for a successful microLife run? How many genomes, with how many phenotypic annotations?

13. Line 150 "Based on literature": how was this data gathered, which literature, etc? The methods section only mentions a literature review. Please add an extra table with relevant literature.

14. Are species names matched to literature? Some of the genomes in the databases are

taxonomically misclassified (as the authors themselves find out further on) so this approach is tricky and needs a discussion. Also, sometimes phenotypes are strain-specific instead of species-specific.

15. Does microLife only work for genera, or also for families and orders, and why?

16. Why not use the decision trees to identify genes that are important for classification, i.e. via the GINI importance, as opposed to / in combination with the Fisher exact test?

17. The protein clustering involves a pretty extensive benchmark of the inflation parameter of MCL on only a couple of genomes. Is a user expected to use the inflation parameter based on this benchmark, or to do this benchmark themselves on their own subset of genomes?

18. Why do you develop of novel method for protein clustering? Why not use one of the existing orthology prediction tools (potentially with tuned parameter values)?

19. Line 79 "to conclusively discriminate": further on you contradict this statement by saying that "this approach enables classification" (line 83).

20. Line 99 "To test the accuracy": how do your examples test the accuracy of the tool?

21. Line 109 "role of 13 yet unknown" \diamond "role of 5 yet unknown".

22. Line 127 "is known as the most accurate algorithm to cluster proteins by function [ref. 39]": very old reference for this statement.

23. Line 154 "reported in the databases": is this a database problem or a literature problem? As far as I understood the authors gathered information from literature.

24. Line 165 "perfect classification" this may be a good place to discuss the correlation between phylogeny (defining most of the pan-genome) and lifestyle (defining only part of the pan-genome).

25. Line 170 "B. vietnamensis" and line 174 "P. cichorii": these species is very easy to predict given that they fall phylogenetically very clearly within their respective lifestyle-type clade. Are there phylogenetic clades / species that have multiple different lifestyles (likely for *Pseudomonas*), and how does the classifier perform on them if it is not trained with them?

26. Line 172 "Confidence level": the amount of agreeing decision trees is not the same as confidence level.

27. Line 177 "we were able to accurately predict the (...) lifestyles": you do not know if these predictions are accurate.

28. In relation to earlier comments, do you predict the lifestyle or actually the phylogeny?

29. Lines 183-...: The location of the mentioned species is not clear on the PCoA, I suggest to add them.

30. Lines 188-189 "These findings provide additional evidence of the adaptive ability of *Pseudomonas* species to shift between lifestyles depending on the environmental conditions". Rather it shows that with very similar genome content pseudomonads can do very different things. One of the premises of microLife is that only genes that are shared between most of these strains are LAGs, but the different clades of pseudomonads may do the 'same' niche adaptation in different ways with different genes, and these are missed by microLife (depending on the % inclusion parameter). It would be good to address this in the introduction.

31. Line 196-204, microLife as taxonomic classifier: binary classification of a species is not taxonomic classification. It's unclear why you would want to do this? The gold standard for taxonomic classification is phylogenetic placement in a tree, which the authors do. The authors

could use the taxonomic classification accuracy of the classifier to discuss the high correlation between phylogeny and gene content and what it means for a user looking for lifestyle associated genes.

32. Lines 226-247 "To further test microLife's gene cluster distribution accuracy (...) aligning with findings reported earlier in the literature": it's unclear what exactly the authors are testing here. The gene distribution is a feature of the gene caller (in this case Prokka) and not of microLife at all. The results are as expected and do not have anything to do with microLife.

33. Line 233 "exclusively": the genes are not exclusive to *P. syringae*.

34. Line 250 "basic statistics": what do you mean?

35. Given that these are binary classifiers / comparisons, 1 for each lifestyle type, could there potentially be overlap between the classified lifestyle / identified LAGs?

36. Lines 259-260 "underscoring the need for studying this largely unexplored genomic information": genes in the accessory genome being often unknown is well-documented, so this statement is pretty flat. Maybe the authors could discuss here how most of these genes are part of the accessory genome instead of the core.

37. Line 267 "genomic regions enriched in LAGs": in a lot of places the authors talk about the genomic context of these genes. The way I understand the methods, if they talk about genes they talk about the MCL clusters, which likely contain multiple genes from different organisms, so do not necessarily have genomic context (i.e. which genes are close, their direction, etc.). Of course if you look into where these genes are present in single genomes you may talk about genomic context, and the context may differ per genome. Is the genomic context of these genes only found in specific genomes or a summary of all genomes / how did the authors decide on this? If so, is it a feature of microLife to be able to do this?

38. Line 271-282 "ToxC and ToxD": I could not find these genes in Extended Data Table 4.

39. Line 276 "consecutive genes": see earlier comment about genomic localisation, how did the authors define this / in which genome / are they colocalised in an X number of genomes? If finding genomic contexts of MCL clusters is a feature of microLife, it is nice because it takes away a potentially complicated bioinformatic analysis and deserves a bit more attention.

40. Lines 288-289 "Our results showed that in *B. plantarii* DSM 9509, 210 out of 786 LAGs had a significant hit to PHI-BASE." Why only in this single genome? Please give a summary of PHI-BASE hits in all genomes. Idem for *Pseudomonas* that is discussed later.

41. Also, it's unclear how Extended Data Table 4 shows this. Doesn't the table contain the MCL clusters instead of the genes from a single genome?

42. Could the authors comment on why not all genes with PHI-base hits are found back in the LAGs? For example for *Pseudomonas* only 118 out of 377 PHI-base hits are found in the LAGs.

43. Line 304 "consecutive": in which genome?

44. Line 305 "were positioned in the same transcriptional direction". Is the transcriptional direction within a genome an output in the R Shiny app (see earlier comments about genomic context)? Or do you need more bioinformatic skills for that? Again I think the sole benefit of microLife is that it makes common analyses easy to do for non-bioinformaticians, so if this genomic-context analysis is not part of microLife it's unfair to give that as an example of the benefits of microLife.

45. Line 308 "Burkholderia genera": strains or genomes?

46. Line 311-312 "5 out of 13 generated mutants were not affected in growth as compared to the wild type strain" What does this say about the other 8 genes and about microLife's performance of

identifying 'real' LAGs?

47. Line 329 "exclusively": is this true? Exclusive means 'only there, nowhere else'.

48. Line 347-351 "Remarkably, microLife was more efficient in the identification of genes involved in virulence when LAGs were present in LAG-enriched regions". This is not a feature of microLife but rather of the way HGT works, operons, etc. There is no reason microLife should be efficient in identifying clustered genes and it should not be implied. Second, whether this is a biologically relevant observation depends on how random the selection of the mutagenesis LAGs was. If the authors want to make a point about LAGs being more often clustered, they should analyse all LAGs on all genomes.

49. Line 362 "majority": the majority means 'more than half'.

50. Line 367 "we observe a trend": add statistics.

51. When looking at the biosynthetic gene clusters, why is only 15% group presence required for LAG definition, vs 70% for the MCL clusters?

52. It may be good to discuss why there are more GCF LAGs found in Burkholderia vs Pseudomonas. I think it relates to the phylogeny versus lifestyle concern raised before.

53. Line 381 "discovery of a highly conserved NRPS-PKS". What do you mean with highly conserved?

54. Line 385: "combined with a BLAST search". It's unclear why the authors do a BLAST search here. The presence of the clusters should come from the clustering module.

55. The NRPS-PKS is said to have a similar distribution to syringofactin and syringomicyn, but I really do not see that in Fig. 7b.

56. The methods section is not clear on what's part of a standard microLife pipeline and what's added custom for this paper. For example, the CheckM and genome statistics, and the type of figures that are generated in the R Shiny module, do they include phylogenetic trees and iTOL annotation files, etc. It's a relevant distinction, because the main benefit of this tool would be the user-friendliness and not the biological results that are presented in this manuscript—which could have been generated without the tool.

57. Line 486 "uniq fasta file": do you mean a "fasta file with unique sequences"?

58. Line 518-522 "It is worth noting that the origins of these unique genes can differ...". Or alternatively the clustering is not complete.

59. Also note that there is a reference left in line 522.

60. Lines 523-533 "gene functional annotation". I think annotating the MCL clusters with different functional databases is nice and something users will appreciate. It can be mentioned in the main text as well.

61. It's unclear in the methods which databases are used, because it says 'such as DBCAN and PFAM', which suggest there are more. Please mention all.

62. Adding InterProScan would be a comprehensive next one.

63. Lines 567-569: please explain how unsupervised approaches are useful for clade/species-specific genes.

64. "Dice dissimilarity" Do you mean the Sørensen–Dice coefficient? Why was this measure chosen?

65. Throughout the manuscript, whenever there is a phylogeny it should be clear how the tree is generated, how it is rooted, whether it is a maximum likelihood tree, etc. Just referring to PhyloPhlAn in the methods is not good enough. Also add support values.
66. The method section is also unclear about marker gene selection and alignment.
67. Fig. 3: What's on the y-axis of the rings?
68. Fig. 3: Why is the Z (cytoskeleton) GO term omitted? It is part of the top significantly associated COG category in Supplementary Table 3.
69. Fig. 3: please put the GO term letters in the figure behind the terms instead of in the caption.
70. Fig. 3: Why are the CDS in this figure?
71. Supplementary Table 3: Can you put the GO terms in the table as well, not just the letters?
72. Supplementary Table 3: Sort the Pseudomonas case according to P value.
73. Fig. 4: How did you define the *P. syringae* complex?
74. Fig. 5: Why not show the distribution for all genes?
75. Fig. 5: How do you define 'function known'?
76. Fig. 6: the caption says (n=845). I only count 17 branches. The caption also says "of the 15 selected LAGs", but it should be 13.
77. Fig. 7a: it says that the BGC are coloured based on lifestyle but it's based on species.
78. Line 1187: It's unclear what's meant by "significantly".
79. Fig 7c: How are the 3 representative BCGs selected?
80. Table 1: What do you mean with "Annotation"? COG annotations?
81. Extended Data Fig. 4: It's unclear which lifestyle are the left-middle-right panels.
82. Extended Data Fig. 4: Why is the mean of the ROC curves not 1 for the Burkholderia case?
83. Extended Data Fig. 7: Please put names of the GO terms in the figure, not just in the caption.
84. Extended Data Fig.7: Change order of lifestyles between panel a and b so that they are consistent.
85. Extended Data Fig. 8: Fig. 3: Why is the Z (cytoskeleton) GO term omitted? It is part of the top significantly associated COG category in Supplementary Table 3.
86. Extended Data Fig. 10: This is a nice figure, is it coming directly from microLife?
87. Extended Data Fig. 15: How is the clustering of the heatmap done? Why not cluster according to the phylogeny like the other plots?
88. Extended Data Fig. 18: How did you define the "top 10 functional type categories"?
89. I would advice to remove hollow excitement generating words like "remarkably", "interestingly", "last but not least", "notably", "enormous [pharmaceutical potential]", and "the framework can be extended to eukaryotic organisms, in particular fungi and yeast". 😊

90. Maybe I missed it, but I could not find the Random Forest results in the R Shiny app.

Reviewer #1 (Remarks to the Author):

Summary:

The study presented a novel computational workflow named "microLife," designed for genome annotation, large-scale comparative genomics, taxonomic delineation, and identification of lifestyle-associated genes (LAGs). The workflow comprised three distinct modules: clustering, lifestyle prediction, and analytical modules.

In the clustering module, gene and Biosynthetic Gene Cluster (BGC) level prediction and clustering were performed. The lifestyle prediction module utilized metadata from the literature to assign lifestyles to bacteria with unknown lifestyles. Subsequently, the analytical module performed further statistical analysis to identify LAGs.

To evaluate the effectiveness of microLife, the authors conducted tests on two datasets pertaining to Burkholderia/Paraburkholderia and Pseudomonas genera. Focusing on the phytopathogenic lifestyle of both taxa, they functionally validated the LAGs identified by microLife. Notably, *B. plantarii* DSM 9509 exhibited enrichment of plant pathogen LAGs in virulence factors compared to all genes when subjected to BLAST analysis against the Pathogen Host Interactions database.

The authors further selected 13 LAGs associated with the phytopathogenic lifestyle in Burkholderia, using *B. plantarii* DSM 9509 as a model for experimental validation. Through virulence assays conducted in rice, they successfully validated the role of five new LAGs identified by microLife in the virulence of Burkholderia.

Additionally, microLife facilitated the discovery of a novel NRPS-PKS gene cluster in plant pathogenic *Pseudomonas* species. To validate this finding, the authors performed mutagenesis of the identified gene cluster in *P. syringae* pv. *phaseolicola* 1448A, revealing a reduced virulence in bean.

Overall, the study demonstrated the efficacy of the microLife workflow in efficiently annotating genomes, conducting comparative genomics, and identifying LAGs, thereby providing valuable insights into the phytopathogenic lifestyles of Burkholderia and Pseudomonas

I can tell that a lot of good work went into this paper. Here were some of my overall impressions:

1. Nice shiny app! Much better than the average database/site in the bioinformatic world
2. Very good experiments, controls, and interpretation of results. Very good potential for development of antibacterial treatments for plants.
3. Very well organized supplemental tables, which is not a given.
4. Great benchmarking of the MCL hyperparameters. Detailed and sensical.

Overall, I think it's a good tool, and it would be interesting to see how other groups use it, and if they indeed find success identifying LAGs as well. It is worth publishing certainly but I do have some major concerns here, which I hope can mostly be answered with some clarification about methodology, and with some further explanation in the text. I do think that there are many points and questions I raise here that would fit in a discussion section (rather than a results+discussion combination section).

Major comments:

1. Line 202: Is *P. viridiflava* misclassified taxonomically? Unlikely given your high AUC values, no? If we are basing it on your classifier, isn't it more likely that substrains *P. viridiflava* have different lifestyle? Maybe be more clear on who is misclassifying here exactly? Do you mean the database where you got the taxonomy? The way it is worded a bit confusing

Anyways, looking at the bigger picture, I would overall downplay the taxonomic classification aspect of microLife. Taxonomy has a field full of arguments over the best method for classifying microbes, and many quite developed and sophisticated programs (e.g. <https://www.ncbi.nlm.nih.gov/pmc/articles/PMC6360808/>) that goes far beyond gene presence and absence. I think the paper is strong enough anyway without the taxonomy aspect, and it only adds a bit of confusion.

Based on our analyses, we indeed observed potential misclassification of *P. viridiflava* strains in the NCBI database, especially for multiple strains originating from the same studies. This had a notable impact on the performance of our classifier, lifestyle metadata was collected from literature at the species level. Wrong species annotations such as these *P. viridiflava* lead to mislabeling of input data in the training as well as test sets. After careful reflection and taking into account the broader implications of our study, we agree with the reviewer's suggestion to remove emphasis from the taxonomy aspect of microLife. Taxonomic classification is indeed a complex field with ongoing debates and evolving methodologies, often extending beyond gene presence/absence, as highlighted in the reference. As a result of these considerations and the valuable feedback, we decided to remove or substantially minimize the taxonomy classification section from our study.

2. The random forest classifier uses the top 1000 important features based on the fisher exact test (which I think in the code actually says top 5000, if I understood correctly? Please clarify). So it's pre-filtered for those genes that already show some variance between the groups. However the materials and methods are unclear. Is it the same 1000 features for both the lifestyle and the genus predictions? Because in this analysis the two are highly correlated it seems, so it's not exactly clear. It's important to make this distinction.

We employed a feature selection process to identify the top 1000 most important gene clusters in each of our datasets, and we have corrected the code on GitHub to reflect this value (the original code applied for the analyses described in the paper also used 1000). For clarity, it is important to note that we constructed separate binary classifiers for predicting each lifestyle category (e.g., plant pathogens, opportunistic animal pathogens, environmental) as well as for species classification. This way we optimized the performance of random forest classifiers by reducing the number of variables while retaining critical information. The features selected are different in every lifestyle classifier, overlapping with the LAGs selected for each lifestyle (see next question for more details). Features or gene clusters that are absent in one lifestyle are also selected within the 1000 features to feed random forests, making the overlap limited.

The reviewer also inquired about whether the same features used for lifestyle prediction are employed in species prediction, potentially due to correlations with phylogeny. The answer varies depending on the species. For instance, in the case of *Pseudomonas syringae*, which encompasses the majority of plant pathogen genomes within the *Pseudomonas* dataset, the features selected for classification indeed are very similar to those used for plant pathogen prediction. However, the scenario differs when considering other species, such as *Pseudomonas putida*, which represents a

small portion of the environmental lifestyle group. In this context, the classifier for predicting *P. putida* selects features or gene clusters that are specific to, or exclusive of, this particular species and are not necessarily applicable to other environmental species. To avoid this complexity/confusion in taxonomic classification by MicroLife (see reply to earlier comment), we have now removed this section from the revised manuscript.

3. While reading the paper I was wondering what are the most important features in the random forest models? This will help understand everything better. You can find out and highlight those. (https://scikit-learn.org/stable/auto_examples/ensemble/plot_forest_importances.html). Again and again I asked myself the question: Why even do the Fisher test again independently if you have the fisher-based features? And you have them ranked further by the model! I think you could have used the trained model along with presence and the 70% presence in genomes etc. I think ignoring these features is a mistake, and it's interesting to at least look and see if the LAGs from the short list are in there.

Thank you for highlighting this point. The second Fisher test (for the LAG detection) is carried out on the predicted lifestyles (on all genomes) instead of on the manually annotated lifestyles (on a subset of genomes), thus increasing the number of samples that can be taken along in the analysis

To shed more light on the features of the model, we have calculated feature importance values and compared them with the selected LAGs in our study. We examined all the features with an importance value greater than 0 that were utilized by the Random Forest algorithm to classify plant pathogen genomes in both the *Burkholderia* and *Pseudomonas* datasets and its overlap with the LAGs chosen (Rebuttal Figure 1). In the *Pseudomonas* dataset, we observed a reassuring overlap of 95% between the LAGs selected using the Fisher test (with criteria of >70% presence and >2 logfold2change) and the features considered important by the Random Forest model. This high degree of concordance suggests that the information used by the Random Forest model largely coincides with the importance of the LAGs identified through traditional statistical methods. In contrast, the *Burkholderia/Paraburkholderia* dataset exhibited a different pattern. Here, only 47% of the LAGs selected overlapped with features identified as important by the Random Forest model. This discrepancy can be attributed to how Random Forest deals with redundant features. When multiple features contain essentially the same information, Random Forest is likely to select only one of them for some of the trees during feature randomization. This behavior is more pronounced in datasets with a limited number of plant pathogen genomes, such as *Burkholderia*. In such cases, many gene clusters may exhibit similar distribution patterns in comparisons between plant pathogens and non-plant pathogens. Consequently, the model may consider only one of these similar features as "important," while the others are omitted.

The examination of the intersection between LAGs and Random Forest features has been incorporated into the Materials & Methods section of the manuscript. (L673-683)

Rebuttal figure 1

It is worth noting that the lifestyle prediction module from microLife currently operates as a separate code due to compatibility issues with the app software. However, we intend to integrate this module into the Shiny app in future microLife versions. This integration will allow users to employ the random forest algorithm and extract important variables interactively in a similar/parallel way as the Fisher test.

4. The statistical analysis to define a LAG as a gene that is frequently present in strains annotated with one lifestyle, is based on Fisher's exact test (Lines 579-581). This test looks solely at the enriched and depleted functions for each lifestyle. The phylogenetic lineage is not taken into account in this analysis, thus ignoring cases in which gene presence/absence patterns are correlated with taxonomy, and the taxonomy is correlated with a lifestyle. In those cases, the gene may be a "passenger" in the genome. Tools like TreeWAS, Pyseer, Scoary, and others try to take this issue into account by correcting for phylogenetic structure. I think that looking for genes present in 70% of lifestyle X partially takes care of this idea, but if 100 genomes of lifestyle X are pretty much the same (very short branch lengths), then what is 70% presence but inflation of the counting?

The results regarding the secretion systems, and of course the experiments vindicate the methodology, but I think it's worth mentioning a bit more about how the "basic statistics" mentioned on line 250 help deal with this issue. Otherwise, it seems that you predicted lifestyle based on taxonomy, and then do enrichment analysis based on these lifestyles, which could then identify mostly taxonomic related genes that separated the groups in the first place. Gene profiles predict taxonomy AND can predict lifestyle. Again, the experiments look good, and perhaps more explanation about the features could help (are core genes filtered out? Etc. How are the features different in each model? If they are different this will help). But, this is another reason I would maybe drop the taxonomic analysis from this paper, so you don't have to address how hard it is to distinguish between (1) correlations of gene profiles ~ taxonomy, with (2) correlations of gene profiles ~ lifestyle.

Thank you for this comment; also the other reviewers requested that we provide insightful comments regarding the statistical analysis employed for defining LAGs in our microLife tool. We perform Fisher tests and group abundance comparisons (>70% group presence and >2 log₂fold change) based on user-defined groups. The threshold of >70% group presence and >2 log₂fold change, which we use to define LAGs, is derived from our focus on 'broad' lifestyles and the identification of lifestyle associated genes, irrespective of their taxonomic distribution. Fisher's exact test answers this question by providing a measure of the significance of the association between the presence/absence of a gene and a lifestyle, without any additional layers of complexity. Acknowledging the potential impact of taxonomy-related biases, the process of reducing genome redundancy via Average Nucleotide Identity (ANI) aids in mitigating this bias in Fisher tests. This reduction helps minimize the presence of highly similar genomes, consequently decreasing potential inflation of results within the Fisher test.

The reviewer rightly points out that tools like phyloglm or TreeWAS are designed to address the issue of phylogenetic structure and its potential impact on gene presence/absence patterns in bacterial genomes; they also introduce their own assumptions and complexities. We have taken this into consideration and conducted a comparison with results obtained using TreeWAS and phyloglm (with a significance threshold of $\alpha = 0.05$). Our findings from this comparative analysis revealed some interesting insights (Rebuttal Figure 2). For instance, phyloglm, being more restrictive, identifies a lower number of plant pathogen LAGs, specifically 31 and 71 plant pathogen LAGs in *Burkholderia* and *Pseudomonas*, respectively, leaving out all the LAGs that were experimentally validated in *B. plantarii*. In the case of *Burkholderia*, all 31 phyloglm-identified LAGs align with the results from TreeWAS and microLife. However, in *Pseudomonas*, only 15 out of the 71 phyloglm-identified LAGs are detected by all three methods, with 44 LAGs not overlapping with either TreeWAS or microLife. TreeWAS appears to be less restrictive, yielding a larger number of LAGs compared to phyloglm and microLife. However, it's worth noting that the overlap between microLife (using the threshold of >70% group presence and >2 log₂fold change) and TreeWAS is substantial, with 100% agreement in the *Burkholderia* dataset and 97% in *Pseudomonas*. This suggests that, while there are differences between existing methods, the threshold we used in microLife for these specific datasets aligns very well with the outcomes from TreeWAS, albeit being slightly more restrictive. The comparison of phyloglm, treeWAS, and microLife has been integrated into the Materials & Methods section of the manuscript (L655-672), and a new figure, Extended Data Figure 22, has been added to illustrate this comparison.

Rebuttal figure 2

Addressing the potential influence of phylogeny on LAG selection, we examined the distribution and correlation of LAGs chosen for mutagenesis and experimental validation. We found that three out of five LAGs identified as reducing virulence in rice (LAGs 14, 22, and 23) are entirely correlated with the phylogeny and were present in all plant pathogen *Burkholderia* genomes in the dataset. This observation raises an important point - excluding genes highly correlated with phylogeny may result in missing important candidates. However, it may also increase the number of "passenger genes" identified as LAGs. To distinguish between phylogenetic noise and true LAGs, we implemented a comparative approach by examining the genomic context of LAGs. Specifically, we categorized LAGs into those situated in genomic regions with other identified LAGs and those located in genomic areas without additional LAGs nearby. Interestingly, the LAGs within regions of other LAGs (LAGs 14, 16, and 22) consistently exhibited reduced virulence in rice. In contrast, only two out of ten "lonely" LAGs (LAGs 11 and 23) displayed reduced virulence in plants. Exploring genomic regions enriched in LAGs as a potential filtering approach offers promise in further refining LAG selection and identifying genuine lifestyle associations.

5. On a related note, Figure 3 shows a massive correlation between genus and lifestyle, and looks "cleaner" than Extended figure 8. But actually Extended figure 8 is kind of more convincing, because it shows the messy reality of the fact that core genome and lifestyle are not necessarily correlated. Figure 3 for example has a strip of highly related plant pathogen genomes, with similar cog profiles. Are these not just sort of clonal expansions? It actually vindicates your methodology more that you find genes correlated with lifestyle that are NOT correlated with phylogeny. Anyway, Figure 3 is very busy, and really impressive that you have that many "dimensions" like colors, sizes, graphs etc. all in one figure. But I wonder what the added value is. It's hard to really tell anything about it. There are no Y axes for the values of the COG and # CDS barplots. Also, how can the COG bar plots represent COG abundances if the values seem to be negative sometimes?

To mitigate the possibility of clonal expansions in our analysis, we took steps to ensure genome diversity. We employed Mash and clustered genomes at a 0.99 Average Nucleotide Identity (ANI) threshold. While it's true that in our *Burkholderia* dataset, phylogeny and lifestyle exhibit a high degree of correlation compared to the *Pseudomonas* dataset, it's important to note that the group of plant

pathogens within *Burkholderia* encompasses species such as *B. gladioli*, *glumae*, and *plantarii*. Although the branch lengths in the phylogenetic tree may appear relatively short, these three species possess distinct phenotypic and genetic characteristics. Therefore, it would not be accurate to classify them as clonal expansions. We've made revisions in Figure 3 and Extended figure 6 to enhance its clarity and focus. Specifically, we've streamlined the figure to reduce visual complexity. We now present only the top four significant COG (Clusters of Orthologous Groups) categories as barplots. Regarding the negative values in the COG bar plots, this is indeed a point worth explaining. The bar plots can display negative values in certain cases because the zero point on the plot represents the mean percentage of that COG category in the entire dataset. Negative values indicate a proportion below the genus average, while positive values signify a proportion above the genus average. This representation allows us to visualize deviations from the average COG abundance within each genus. More description about the barplots were added in figure legends for clarification.

6. It would be interesting to know if there are LAGs which are shared between *Burkholderia* and *Pseudomonas*.

We compared plant pathogenic LAGs in *Burkholderia* and *Pseudomonas*. Using BLAST, we searched for bidirectional best hits between plant pathogenic LAGs of *Pseudomonas* and *Burkholderia*, finding 13 shared LAGs out of 786 in *Pseudomonas* and 377 in *Burkholderia*. This suggests a minimal 1-3% overlap between plant pathogen LAGs of these two genera. We included a supplementary table (Extended Data Table 6) with information about the 13 overlapping LAGs and discussed these results in the manuscript L311-320).

7. Extended figure 7: You look at all COG categories except category Y, nuclear structure, for reasons that prokaryotes have no nucleus (although it's in the figure legend, but that is a minor point; another minor point, Kruskal is misspelled in the legend). However you keep category A and category Z, which (at least originally) are not present in prokaryotes either. Please confirm that this is indeed the case and then remove them. This may explain the very low numbers you are getting as opposed to the other categories. Also, you are getting massively significant differences (e.g. 9.7×10^{-134} for category Z in *Pseudomonas*) in these possibly meaningless categories, which brings up questions about the meaning of these statistics as a whole... Please think about this and confirm that you believe the other results.

Thanks for your comment, you are right; categories A and Z, which are typically associated with eukaryotes, should not have been included in the analysis. We appreciate your observation, and we have now removed these categories from Extended Data Figure 7. This adjustment ensures that the analysis focuses solely on categories relevant to prokaryotes.

8. Lines 480-483: In regard to the assignment of lifestyles to bacterial genomes, the authors have based their classifications on existing literature for each species. However, it is important to consider the possibility that certain strains within the same species might exhibit different lifestyles. This is a key point which needs to be addressed. How exactly did you define the lifestyles? Do you have heterogeneity of lifestyle in one 99% ANI group?

In our study, we opted to categorize bacterial genomes into three broad lifestyle groups: plant pathogens, opportunistic animal pathogens, and environmental. This choice allowed us to effectively assess the capabilities of microLife, in identifying genes associated with these 'broad' lifestyle categories at the genus level. Furthermore, we conducted experimental validation to ensure the reliability of our findings. The flexibility of microLife is a key feature. It empowers users to integrate their own metadata and perform customized comparisons tailored to their specific research questions, including strain-specific lifestyle metadata of high specificity if desired. However, as a proof-of-principle, our focus here was on establishing a broad overview of bacterial lifestyles within these three general categories. This approach enabled us to gain valuable insights into bacterial lifestyles and in particular the plant pathogenic lifestyle.

Regarding the potential heterogeneity of lifestyles within 99% Average Nucleotide Identity (ANI) groups, we did observe some examples where different species were clustered together at 99% ANI. For example, in Extended Data Table 1, we identified cases like *Pseudomonas viridiflava* GCF.900589445.1, which was clustered with *Pseudomonas fluorescens* GCF.902497905.1 at the 99% ANI level. Such discrepancies can be attributed to misclassifications within the NCBI database, which can affect the homogeneity of 99% ANI clusters and our lifestyle annotation. To assess this, we quantified the proportion of 99% ANI clusters that contained more than one species, excluding unassigned species labeled as "sp." In *Pseudomonas*, a significant proportion of clusters (92.4%) contained only a single species, suggesting homogeneity. In contrast, *Burkholderia* exhibited higher homogeneity, with 97.9% of clusters containing only one species.

9. Given the 90% completeness cutoff, and 99% ANI, you might have a gene cluster in 98% of genes, and missing in 2% for technical reasons. This might produce a fisher enrichment where there is none. How do you deal with this? Do you remove those gene clusters are all "1", i.e. genes present in all genomes? What about a gene cluster with 95% "1" labels? Extended figure 20 makes me more relaxed, but it would be good to include a graph of completeness as well so we know how many times a gene might be "missing" when it's not really missing.

To address potential technical limitations, we implemented rigorous quality control criteria. Our selected genomes had to meet a >90% completeness threshold and contain less than 5% contamination, as determined by CheckM. We have incorporated a scatter plot in a new figure (Extended Data Figure 19), depicting contamination versus completeness. This plot underscores the high completeness levels in our dataset, with an average of 99.64%. Consequently, the potential for genes to be genuinely missing due to technical reasons is low. Based on this data, we estimate that, on average, there are approximately 31 missing genes per *Burkholderia/Paraburkholderia* genome (out of 6995 genes/genome) and around 24 missing genes per *Pseudomonas* genome (out of 5330 genes/genome). The expanded analysis of genome completeness has been incorporated into the Materials & Methods section of the manuscript (L525-533), and a new figure, Extended Data Figure 19, has been included to visualize these findings.

10. Line 188-189. Lifestyle switching / flexibility is the charitable interpretation of mixed clusters. Although I think it is true that biology is too complicated to really put into simple lifestyles, it is worth looking deeper into here. Just to be sure, perhaps looking into the third principle coordinate could potentially separate between these mixed clusters?

Our analysis included Principal Coordinate Analysis (PCoA) to gain insights into the dataset's complexity and the potential separation of mixed clusters. In the PCoA of *Pseudomonas* (Figure 2), which utilizes the entire dataset, we identified a cluster of bacteria primarily categorized as environmental, but also comprising some plant pathogens and opportunistic/animal pathogens. This cluster is depicted in the bottom right corner of Figure 2b. Subsequently, to investigate further, we conducted a dedicated PCoA focusing exclusively on this group (as illustrated in Rebuttal Figure 3). In this refined PCoA, we observed a clearer differentiation among the various subgroups within this mixed cluster. While genomes annotated as environmental still displayed some degree of dispersion, we discerned a distinct separation between opportunistic animal pathogens (positioned on the left side of the plot) and plant pathogens, which clustered at different points on the right side of the plot. This suggests that even within mixed clusters, there are discernible patterns and differences that warrant consideration.

Rebuttal figure 3

Minor comments:

1. Line 34: As *a* proof of concept,

This was done as suggested.

2. Line 37: add “,respectively” at the end.

This was done as suggested.

3. I understand that you want this to be a general tool, but mentioning LAGs in the abstract and then not giving some examples (pathogen vs. non-pathogen or something) is a bit too unspecific. I understand you don't want to mention plant pathogenicity too deeply since it's a general tool but some hint as to which kinds of lifestyles microLife can analyze would be good.

We categorized lifestyles in our dataset according to pathogenic interactions with other organisms. However, it's essential to note that users should determine the appropriate lifestyle classes based on their specific research questions. We included this clarification in the revised introduction (L108-112)

4. I recommend that you add the github or site to the abstract

This was done as suggested.

5. In Line 93, you mention "...tool like microLife", and then only in line 99 you introduce the term "...designated microLife"

microLife name now is introduced in L103 of the revised manuscript

6. Lines 182-188: you mention many species that are in the figure, but when looking at the figure, there is no way to tell which dots refer to these known species. Maybe provide the coordinates in the legend at the least.

Good point. We added a new figure (Extended Data Figure 5) with PCoA plots for both genus colored by species to complement what is indicated in the text

7. Lines 185-189: The environmental Pseudomonas are not distributed in a large cluster as the author claim. They are distributed in multiple clusters (Figure 2b)

Corrected as suggested. (L202-205)

8. Figure 2 - mark the species names on the tree. The reader has to guess what the authors describe in the associated text.

We added a new figure (Extended Data Figure 5) with PCoA plots for both genus colored by species to complement what is indicated in the text

9. Figure 2b is referred to in the text first, and then figure 2a.

Corrected as suggested.

10. Line 289: the -2 does not need to be superscript, and the E should be capital. Lines 293 and 296 also have strange formulations of the pvalue. Please confirm it's meant to be $0.1 * 10^{-6}$

Corrected as suggested.

11. Extended data 10: LAG 11 label is somewhat unreadable

The mistake was corrected in Extended Data Figure 9.

12. Extended data 12: Why is the overall pipeline shown again? It's the third instance I think and you can just show the experimental design part. It's already in Fig 1, Ext Fig 3 (I understand why), but here it seems unnecessary. If it's meant to be different, I did not notice

The bioinformatic pipeline was removed in what is now Extended Data Figure 11.

13. Maybe mention the stages/characteristics expected process of color loss and leaf dying in one sentence for the non-plant science reader. microLife is meant to be for wider use so it would be good to define the jargon.

Discussion of expected process of color loss and leaf dying is added in the main text (L450-455)

14. Line 268: I think it's important to mention that this idea is one hypothesis, and the other hypothesis is that the highly related species with the highly related lifestyles may simply have a shared genetic history or environment with a specific phage. In other words, these phage genes could be a "passenger" gene with no function. Without more details shown, it is hard to know. Are these lysozymes encoded next to the enriched LAGs? Maybe then you can come up with a hypothesis about their origin.

These genomic regions enriched in LAGs associated with type II, III and VI secretion system in *Burkholderia* were in some of them, for example rows 692-711 of Extended Data Table 4 is a genomic region with a gene annotated as Type IV secretion system together with viral origin genes. Our

hypothesis comes from observing these genes next to each other. To clarify, we have now specified this in the main text (L291-292) of the revised manuscript

15. The logic of going from COG category U to T3SS is a bit weird to me. It seems only vaguely connected in the flow of the story of the paper. You have the secretion systems as a result in the LAG section anyway... I would structure the paper such that the LAG analysis is done first (paragraph starting at 249), then put figure 4. The connection between COG and T3SS is a bit weak, and honestly all the COG stuff can be removed. It raised more questions for me than answers. Your main result is the identification of LAGs, including secretion systems.

We acknowledge that the correlation between the COG categories and the Type III Secretion System (T3SS) may appear somewhat detached or less immediately pertinent in the context of the primary findings concerning the LAGs and secretion systems. Nevertheless, the inclusion of COG categories serves to furnish users with a holistic insight into the functional landscape encompassing the identified LAGs. While we recognize the limitations in interpreting individual genes within these COG categories, our intent is to present a broader functional categorization that aligns with our objective to deliver a comprehensive overview of the genetic landscape.

16. Regarding the comparison between genomes from different lifestyles, the authors have employed a method based on the presence and absence of specific functions. However, it is worth noting, at least in the Discussion section, that this approach does not consider the copy number variations of these functions within each genome. As copy number variations can significantly influence the functional potential and gene expression of organisms, their omission in the analysis might introduce certain limitations to the accuracy and comprehensiveness of the results (209-212). Furthermore, SNPs and other changes at a higher resolution may account for differences that cannot be seen with gene presence absence profiles. Comment on this and explain why your methodology is still advantageous.

Very good point. Indeed copy number and SNPs or frameshifts may also influence gene expression and the associated lifestyle. Around 83% and 89% of gene clusters contain only one copy in *Burkholderia* and *Pseudomonas* respectively. Given the well-established significance of gene acquisition and deletion in microbial evolution, we initially chose to concentrate on presence/absence patterns. In the future, the ability to assess differences in gene copy numbers could unveil latent associations that cannot be identified solely through presence/absence data. This valid point of discussion is added in the Materials & Methods section of the revised manuscript (L647-650)

17. There is no acknowledgement section. I guess there are some funding sources the authors would like to acknowledge.

Added as suggested.

Reviewer #2 (Remarks to the Author):

The manuscript "From genomes to lifestyles: discovery and functional characterization of lifestyle" deals with the description of a new tool named MicroLife, which aims to relate microorganisms and genes/functions with lifestyles based on a classifier trained with trusted data. The authors validate the results through the exploration of *Paraburkholderia* and *Pseudomonas* genomes and the discovery of several genes related with plant pathogenicity. Both the bioinformatic tool and the specific findings on the putative pathogenic role of those few genes described to be related with that trait are of great interest. However, I consider that the metadata used to validate the tool is too biased to be useful for such purposes, which threat the reliability of the results.

The initial classification of species into lifestyles is too biased for diverse reasons: (I) The number of species selected from literature is too narrow. (II) The authors did not classify properly the genomes into species through reliable methods (i.e.: ANIb, dDDH), and then, misclassified genomes such as those of *P. viridiflava* ('pathogens'), that were indeed within the *P. fluorescens* clade ('environmental'), were incorrectly used for the training of the random forest model. (III) The categorization of strains into lifestyles just based on taxonomy is also too biased. Some commensal strains can gain pathogenic or beneficial genes and change their host-performance^{1,2}. Also, some strains from the same species may have different lifestyles. It may be more correct to delineate the lifestyles based on the isolation source or based on validated traits for each strain, rather than based on the lifestyle of the most common one for the species. In any case, using the literature metadata of species, but not having taxonomically classified each genome reliably, is too risky. Also, the microhabitat can be biasing the life-style based analyses; for instance, it is not discussed whether the epiphytic vs rhizospheric (soil or root) habitat could affect the analyses rather than the effect of each strain within the host. Probably, the distribution of genomes in the PCoA can also be distinguished based on this type of microhabitats, not only based on the host performance, since there may be more genes/functions needed to use the differential resources among microhabitats rather than the number of genes required to establish a beneficial or pathogenic outcomes. Hence, as an example, would it be possible that the genes found to be related with pathogenicity may not be related with this, but with the survival/fitness of the strain within the plant leaves?

Another important issue is the effect of phylogeny in the discovery of trait-associated genes. For instance, as shown in Extended Data Fig. 11, Burkholderia lifestyles are highly clade-specific. Hence, the discovery of lifestyle-associated genes might be biased to find clade-specific genes rather than those related with their lifestyle. In the case of *Pseudomonas*, for which their lifestyles are less associated with phylogeny, this bias could be reduced by using strain isolation data rather than extrapolating species metadata to all the strains named as that strain. In any case, in both cases, authors might use statistical methods that reduce this bias, such as PhyloGLM.

1. Li, E., de Jonge, R., Liu, C., Jiang, H., Friman, V. P., Pieterse, C. M., ... & Jousset, A. (2021). Rapid evolution of bacterial mutualism in the plant rhizosphere. *Nature Communications*, 12(1), 3829.
2. Drew, G. C., Stevens, E. J., & King, K. C. (2021). Microbial evolution and transitions along the parasite–mutualist continuum. *Nature Reviews Microbiology*, 19(10), 623-638.

We acknowledge the reviewer's concerns about the initial species selection and taxonomy classification. Our choice to categorize bacterial genomes into broad lifestyle groups (plant pathogens, opportunistic animal pathogens, and environmental) was made to demonstrate the capabilities of our tool, microLife, at a high level. As you rightly pointed out, microLife is designed with flexibility in mind, allowing users to integrate their own metadata for customized analyses. For example, can we identify genes involved in beneficial activities of bacteria to help plants cope with (a)biotic stresses?"

We understand your point that this approach may have limitations due to the diversity of lifestyles within species. Considering isolation sources and validated traits would provide more accurate insights, obtaining this information for a very large number of isolates/strains is currently challenging. In this context, in an ongoing follow-up study, microLife was recently employed with more specific metadata tailored to investigate host-specificity within the syringae complex. Genomes were classified as either woody or herbaceous based on their infection capacity, enabling the identification of gene regions associated with woody infectious lifestyles, among them is the WHOP region, which had been previously characterized and experimentally validated (Caballo-Ponce et al., 2017). Furthermore,

while we recognize that a more specific metadata approach can yield deeper insights, it's important to strike a balance. Keeping our initial classification simple, as mentioned, allows us to convey the primary objective of microLife clearly to potential users. As we refine the tool and incorporate more detailed metadata in future iterations, we anticipate that it will offer a broader range of answers. Regarding the issue with misclassified "P. viridiflava," we completely agree with your observation. In fact, that paragraph and all related content about species classification have been removed from this version of the manuscript (see also our reply to a related comment by reviewer 1). Regarding the correlation between phylogeny and lifestyle, we kindly refer to the response provided to Question 4 by Reviewer 1 for more detailed information.

Beyond those major concerns, I have some other comments that may be of interest:

- L 73-74. To date, the *Pseudomonas* genus encompass 316 valid species, not 144. I suggest updating reference 27 (year 2017) for some more recent work that revise the taxonomic status of this genus, such as some of these:

- o <https://doi.org/10.3390/biology10080782>
- o <https://doi.org/10.1099/ijs.0.064634-0>
- o <https://doi.org/10.1016/j.syapm.2021.126289>

Added as suggested. Thanks for the references (L78)

- L. 144 or 149: It could be useful to state how many genomes/strains belongs to *Burkholderia* and how many to *Paraburkholderia*

The number of *Burkholderia* and *Paraburkholderia* genomes was added as suggested. (L156-157)

- L. 200-204. Please, confirm the taxonomy of these genomes through valid methods such as ANIb or dDDH values shared with type strains,

The taxonomic classification of the genomes extracted from the NCBI was confirmed using gtdbtk. This tool employs Average Nucleotide Identity (ANI) values for categorize strains and providing approximate taxonomy or species assignments. In general, the overlap between the species assignment between NCBI and gtdbtk in *Paraburkholderia/Burkholderia* and *Pseudomonas* (only applied to genomes with a species assigned) was approximately 75% (1485/1967). As discussed and pointed out by the reviewers it is essential to note that taxonomic classification is a field that change constantly and specially within the *Pseudomonas* genus the frequent renaming of species contributes to some degree of discrepancy between the NCBI species assignment and the gtdbtk assignment. However, both gtdbtk and comparable methodologies undergo regular updates using NCBI assemblies. Additionally, the NCBI database has implemented an (ANI)-based quality control process to validate genomes from type strains, enhancing related sequence records similar to gtdbtk. Incorporating a taxonomy classification method like gtdbtk into the microLife framework will help prevent misclassifications when assigning lifestyles based on species in large datasets. We contemplate incorporating this into future iterations of the tool.

- Chaumeil, P. A., Mussig, A. J., Hugenholtz, P., & Parks, D. H. (2020). GTDB-Tk: a toolkit to classify genomes with the Genome Taxonomy Database.
- Kannan, S., Sharma, S., Ciuffo, S., Clark, K., Turner, S., Kitts, P. A., ... & Kimchi, A. (2023). Collection and curation of prokaryotic genome assemblies from type strains at NCBI. *International Journal of Systematic and Evolutionary Microbiology*, 73(1).

- Discussion (i.e. L. 218-226). I recommend providing more discussion about the genes found to be associated with some lifestyle. I.e.: detail some genes that not had never been related with such lifestyle, comparing the results with more works beyond that of Levy et al. (2018), etc. Here I provide some references that may be useful:

o Melnyk, R. A., Hossain, S. S., & Haney, C. H. (2019). Convergent gain and loss of genomic islands drive lifestyle changes in plant-associated *Pseudomonas*. *The ISME journal*, 13(6), 1575-1588.

o Saati-Santamaría, Z., Baroncelli, R., Rivas, R., & García-Fraile, P. (2022). Comparative genomics of the genus *Pseudomonas* reveals host-and environment-specific evolution. *Microbiology spectrum*, 10(6), e02370-22.

o Chewapreecha, C., Mather, A. E., Harris, S. R., Hunt, M., Holden, M. T., Chaichana, C., ... & Peacock, S. J. (2019). Genetic variation associated with infection and the environment in the accidental pathogen *Burkholderia pseudomallei*. *Communications Biology*, 2(1), 428.

Inorganic ion transport and metabolism (P) has been reported as more abundant in *Pseudomonas* associated with humans, according to Saati-Santamaria et al. 2022. This trait is also observed to be more prevalent in opportunistic animal-associated bacteria within both the *Pseudomonas* and *Burkholderia* genera. This additional discussion has been added thanks to the references given (L235-239)

- L. 251 and 253. It would be useful to add supplementary tables with the gene names or even with the gene/protein sequences of these LAGs.

We added an Extended Data Table 11 with gene names and gene sequences.

- L 274-275. AHLs act also as beneficial molecules for the root colonization of plant-beneficial bacteria. Please, provide some discussion on why the LAGs related with AHLs are associated with pathogenicity but not with commensalism or mutualism.

We provided more discussion of the potential role of AHLs in the revised manuscript (L297-301)

- L. 288-297. I suggest to not only compare 1 genome/genus against the PHI-base to validate these results, but against more genomes.

We revised our analysis strategy concerning the PHI-base. In this updated approach, we select the representative sequence from each gene cluster and subject it to a BLAST search against the PHI-base (evalue < 1e-2). Subsequently, we compare the relative prevalence of significant hits between phytopathogenic LAGs to the entire pan genome of the genus. (L322-335)

- Fig. 6A. "...showing the distribution of the 15 selected LAGs. "13" instead of "15" ?

The mistake was corrected in Fig. 6A.

- Table 1. Please, provide the sequences (nt or aa) for these genes as a supplementary table.

We added a new Extended Data Table 11 with gene names and sequences.

- L. 361-370. You introduced the importance of NRPS systems in the *Pseudomonas* genus, but you are then focused on an hybrid NRPS-PKS biosynthetic gene cluster. Please, provide equal discussion on the importance of PKSs or even these hybrid clusters in *Pseudomonas*, if possible. Indeed, NRPS-PKS BGCs are not so common within this genus (<https://doi.org/10.1099/mgen.0.000758>). The following article includes some data that may be useful for the discussion:

o Gross, H., & Loper, J. E. (2009). Genomics of secondary metabolite production by *Pseudomonas* spp. *Natural product reports*, 26(11), 1408-1446.

Thank you for providing the guidance and reference. As revealed in Extended Data Figure 16b, NRPS-PKS biosynthetic gene clusters (BGCs) are indeed relatively scarce within *Pseudomonas*. We have

incorporated a discussion on this topic, including insights from well-studied NRPS-PKS hybrids, based on your provided reference. (L443-446)

- Fig. 7f and Extended Data Fig. 19. What is the number of replicates (n) used for these experiments? I recommend including these values within the figures or captions.

We added to the captions of Figure 7f and Extended Data Figure 18 that the values are the combined outcomes of two independent experiments.

- L. 410. "7" instead of "seven"? Just to use the same writing style along the manuscript

Corrected in the revised manuscript.

- Extended data Fig. 20. It seems that there are several genomes too fragmented (500-2000 contigs), which threatens the GWAS analyses (some genes could be fragmented or absent). I recommend removing these genomes from the analysis.

Correct that there are genomes with a high number of contigs, but for our analyses we followed the threshold of >90% completeness and < 5% contamination, values given by checkM (L525-533)

- L. 485. Please, remove the detail of the citation ("Seemann, 2014") and just leave the numerical cite.

Corrected in the revised manuscript.

- L. 490. What is the rationale of using 90% of similarity for clustering sequences? Apart from this, usually, protein sequences rather than gene sequences are used for such clustering. As told in the lines 487-488, you used "gene" clustering. Please, just be sure that you aim to mention "gene" instead of "protein".

The microLife clustering module performs two clustering methods in a hierarchical way. First method consists of clustering the sequences at 90% sequence similarity. The second method consists of BLASTing the representative of each 90% sequence similarity cluster "all vs all" followed by Markov clustering to generate the final clusters. The reviewer is right here, clustering of sequences is with proteins so we corrected this in the text (L539).

- L. 591-592. Please, detail the algorithm used (i.e.: Maximum Likelihood).

Added in the Material & Methods section of the revised manuscript (L686)

- L. 595. Why these thresholds? These are not the same than those used to create gene clusters.

Genes were clustered based on network graph theory with Markov Clustering which is based on sequence similarity but doesn't use a threshold to make the clusters. The within-gene-cluster sequence similarities are different for each cluster. When blasting a gene to our genomes to confirm the distributions of Markov Clustering, we use a threshold of 70% coverage to avoid hits that are caused by proteins that share one domain or small parts of the sequence and a threshold of 40% sequence similarity to allow flexibility in the sequence as Markov clustering does.

- Results and discussion. Authors are providing a new pipeline to discover trait-associated genes, among other features. It would be useful for the readership to compare MicroLife with other workflows or tools (i.e.: Scoary, PhyloGLM, etc.) or at least discuss the pros and cons among them.

Very good point. For this, we kindly refer to the response provided to Question 4 by Reviewer 1 for more detailed information.

Reviewer #3 (Remarks to the Author):

Guerrero-Egido and co-authors introduce the bio-informatic pipeline microLife that aims to find genes associated with phenotypic traits within microbial taxa. MicroLife aids in downloading microbial genomes, reconstructs the pan-genome via clustering of homologous genes, and identifies genes and biosynthetic gene clusters that are associated with a user-defined phenotype using basic statistics.

The authors call these genes 'Lifestyle Associated Genes (LAGs)'. In addition, a binary classifier is built to predict phenotype based on the pan-genome. MicroLife includes a point-and-click R Shiny web application that allows for graphical exploration of its results. In the manuscript, two microbial genera are analysed to identify plant-pathogenic-associated genes as an example use-case for microLife.

While none of the described methods are novel, the authors are upfront about this, and I do believe that there is a need for easy exploration and visualisation of the pan-genome aimed at researchers with very little bioinformatic experience. However, the validity of a pipeline like this depends on the soundness of the underlying methods and a comprehensive exploration of what it can and cannot do. I identify a number of major concerns with the bioinformatic methods and the way the manuscript is presenting these methods.

1. The basic promise of microLife is that genes that are associated with a certain phenotype can be identified with a Fisher exact test of the presence/absence profile in the pan-genome, and that this leads to a genetic understanding of the phenotype. However, genes may also be shared because of clade-specific common descent, and have nothing to do with the phenotype per se. The (Para)Burkholderia group is a good example, because the phenotype ('lifestyle') just follows the species tree (Fig. 3). Predictions based on genome content are trivial in this case, as shown by the high predictability of the classifier. Any gene that pops up is associated with the clade, but only a handful may be responsible for the phenotype, and microLife has no way of discerning them. The *Pseudomonas* case is more interesting because lifestyle and phylogeny are decoupled (Extended Data Fig. 8), but the classifier is doing worse, and I expect it to be bad at the 'difficult' cases exactly where phylogeny and phenotype are decoupled. An exploration of this dependency between phylogeny and lifestyle (potentially by contrasting the Burkholderia to the *Pseudomonas* case) is missing.

We appreciate the reviewer's insights into the challenges of linking genes to phenotypes, especially when phylogeny and phenotype are closely aligned, as seen in the (Para)Burkholderia group. In such cases, the risk of introducing phylogenetic noise is higher, making it difficult to pinpoint true Lifestyle-Associated Genes (LAGs). To address this, we've adopted an approach that focuses on genomic context, identifying adjacent gene regions (genomic context) enriched with LAGs. This strategy has considerably improved the efficiency of selecting genes for experimental validation, leading to observed reductions in virulence upon gene mutation. We aim to isolate genes consistently associated with lifestyles within genomic regions, reducing the likelihood of including genes solely driven by phylogenetic factors. We kindly refer to the response provided to Question 4 by Reviewer 1 for more detailed information. The valid point of the dependency between phylogeny and lifestyle (potentially by contrasting the Burkholderia to the *Pseudomonas* case) is addressed in the discussion of the revised manuscript (L211-219).

2. In relation to this, the manuscript does not discuss the pan-genome and how core and accessory genes may be involved in adaptation (and associated extensive literature), but this is what it studies. We included more results and discussion of the pan genome, core genome and unique genes in the Materials & Methods section of the revised manuscript. (L567-581).

3. Similar pipelines exist, and are not cited or discussed. Scoary (10.1186/s13059-016-1108-8) comes to mind, whose underlying pan-genome prediction pipeline (based on Roary (10.1093/bioinformatics/btv421)) is arguably more advanced because it attempts orthology prediction as opposed to only clustering of homologous sequences, and has more statistical tests for gene associations to phenotype (including phylogeny correction, see previous comment). How do

microLife's results of (Para)Burkholderia/Pseudomonas compare to Scoary, and what does it add that Scoary does not?

For further details on alternative methods and the comparison of their outputs (pros/cons) to that of microLife, we kindly refer to the response provided in Question 4 by Reviewer 1. We didn't include Scoary in the alternative methods because of the complexity in adapting our gene cluster absence/presence tables to the Scoary input format.

4. Again, given the strong correlation between phylogeny and lifestyle, I am not surprised that the classifiers are relatively good in predicting lifestyle based on genome content. It would be relevant to discuss where the classifiers go wrong, for example by plotting the misclassifications on the phylogenetic trees and PCoA. The classifier scores should be added to Supplementary Table 2.

We retained the lifestyle prediction while excluding the species prediction analysis. We added the classifier scores in the Extended Data Table 2. These scores represent the proportion of trees that vote for each lifestyle category. Note that a binary classifier is built for each lifestyle category.

5. MicroLife is aimed at researches with little bioinformatic experience, however most of the analyses for the Burkholderia and Pseudomonas cases are as far as I can tell not output from microLife but require again bioinformatics expertise. For example, are the phylogenetic trees with iTOL annotations, Extended Data Fig. 7, and the BCG network, coming out of microLife? It seems that microLife was only used to identify the genes of interest (and aggregation of functional categories), which given that that this method is not novel or groundbreaking, reduces the significance of the tool for the findings presented.

The users do not need to have extensive bioinformatic skills such as coding to run microLife but they do need to have skills in using tools for the representation of the obtained results. In our case, iTOL serves as a crucial tool for visualizing and annotating phylogenetic trees, which plays a pivotal role in validating our methodology. Furthermore, BGC networks are constructed using Cytoscape, with the input being generated by BiG-SCAPE within the microLife framework, establishing a consistent connection between these two tools. Additionally, we have integrated the antiSMASH User Interface (UI) within the microLife app. To provide a comprehensive understanding of the app's capabilities, we have expanded upon their descriptions in the methods section. (L699-714)

6. The manuscript would benefit from a rewrite with a clear description of what the microLife R Shiny module can do, and how that relates to these findings.

A more detailed explanation of the microLife R Shiny module is included in the revised manuscript (L699-714).

7. It is unclear how the 13 + 1 LAGs that were selected for mutagenesis were chosen, out of 786 potential LAGs in total. Are they randomly selected, or is there cherry picking going on. If so, how? How is microLife contributing to this selection process?

microLife Shiny app allows users to generate a table of the detected LAGs in a specific genome of the dataset. This table includes the genome position of the LAGs in that genome, this allows us to find regions enriched in LAGs. We gave priority to these regions but we wanted to also test the role of LAGs that were in nonLAG-enriched genomic regions. With this in mind, we randomly selected LAGs in genomic regions enriched in other LAGs, and selected 'lonely' LAGs for comparison.

8. A good example of coupled lifestyle and phylogeny (see first comment) may be the 8 genes whose mutagenesis resulted in impaired growth. These genes may not be related to lifestyle but instead to living.

We investigated the distribution and phylogenetic correlation of LAGs selected for mutagenesis. Notably, three of the five LAGs identified for their role in reducing virulence in rice (LAGs 14, 23, and 22 in Figure 6a) exhibited complete alignment with the phylogenetic tree and were present in all *Burkholderia* genomes associated with plant pathogens in our dataset. On the other hand, we observed that three LAGs (LAGs 1, 8, and 18 in Figure 6a), with delayed growth compared to the wild type, may be linked to essential housekeeping functions and display a lesser degree of phylogenetic correlation. For more detail information please refer to Question 4 from reviewer 1

I have put my next concerns in more or less chronological order of the manuscript:

9. The title says “discovery and functional characterization of LAGs”, however I do not agree that the functional characterization is the take-home-message, and it largely oversells the pipeline as if it is a silver bullet to genotype-phenotype association. I would suggest focusing on the user-friendliness of aggregating multiple commonly-used tools in a point-and-click interface, which is its main benefit.

We appreciate the insightful feedback provided by the reviewer regarding the emphasis on functional characterization in the title. We acknowledge the importance of clarifying the key takeaway message and agree that the user-friendliness of the microLife pipeline is the ability to aggregate multiple commonly-used tools in a point-and-click interface. We propose a modification of the title to better align with the suggestions: "microLife: a user-friendly interface for genome mining and discovery of lifestyle-associated genes in bacteria". This adjustment aims to accurately reflect the primary benefit of microLife, which lies in its accessible approach to genomic analysis rather than solely in the functional characterization of lifestyle-associated genes.

10. In the introduction the authors talk about beneficial or detrimental effects to the host, however they focus in subsequent analyses only on pathogens. Would microLife also be appropriate in identifying genes associated with beneficial effects, why/why not? I can see how a pathogenic lifestyle is an absolute best-case scenario for association studies.

The primary reason for our focus on plant pathogens in the initial validation was practicality. Validation experiments can be resource-intensive and time-consuming. Therefore, it was more feasible for us to perform these experiments with plant pathogenic bacteria. However, we would like to emphasize that microLife's capabilities extend beyond pathogenic lifestyles. It is designed to be adaptable to diverse research questions and can be applied to different types of metadata or "lifestyles." As already mentioned in this rebuttal our groups are already utilizing microLife to answer other questions.

11. The introduction would benefit from a discussion of microbial niches and phenotype, given that microLife can be potentially used for any type of association. Currently the introduction reads as if ‘making another organisms sick’ is the niche of a microbe, but microbial niches are of course much more diverse.

We clarify now in the introduction that the distinct functions of each bacterium within its specialized niche contribute to the unique phenotype of a bacterium and that interactions with other organisms are important to define the niche function of a bacterium (L48-54)

12. What do you need for a successful microLife run? How many genomes, with how many phenotypic annotations?

Although microLife is designed for big datasets, it can be used for small datasets with few genomes if there is a very strong association between genes and lifestyles; of course, as in all such statistical analyses, the more genomic observations in the input data, the more power the analysis will have. In the example app here: http://178.128.251.24:3838/microLife_linux/ we have a small microLife performed with 20 bacteria, 10 environmental and 10 plant pathogens as an example for the reviewers. Crucial for the success rate is a very well defined metadata sheet of the microorganisms under study.

13. Line 150 “Based on literature”: how was this data gathered, which literature, etc? The methods section only mentions a literature review. Please add an extra table with relevant literature.

The metadata was compiled from available literature at the species level. Given the vast number of genomes involved, a manual annotation of each genome was deemed impractical. We acknowledged that this approach may introduce some errors or inconsistencies. However, it is emphasized that such discrepancies are expected to be isolated cases, and, on the whole, the annotation is considered accurate and reliable.

14. Are species names matched to literature? Some of the genomes in the databases are taxonomically misclassified (as the authors themselves find out further on) so this approach is tricky and needs a discussion. Also, sometimes phenotypes are strain-specific instead of species-specific.

Species names were obtained from the RefSeq database, with only a few genomes found to be taxonomically misclassified. To delve deeper into this issue, please refer to the response to Question 8 by reviewer 1, where we assess the consistency of species annotations within the 99% Average Nucleotide Identity (ANI) bacterial clusters used as input for microLife.

15. Does microLife only work for genera, or also for families and orders, and why?

microLife offers versatility, accommodating various taxonomic levels, including genera, families, and orders within its comparative analysis framework. The choice of taxonomic level depends on the research objectives. In our study, we focused on the genus level to investigate bacterial lifestyles effectively, particularly within groups like *Pseudomonas* and *Burkholderia*. This level strikes a balance between detail and inclusivity, enabling us to capture meaningful lifestyle variations while managing dataset size. It's important to note that microLife's capabilities extend beyond the genus level. Researchers can easily adapt microLife for comparative genomics at higher taxonomic ranks, such as families or orders, by providing genomes from the desired taxonomic group as input. This flexibility allows for tailored analyses to match specific research questions and goals.

16. Why not use the decision trees to identify genes that are important for classification, i.e. via the GINI importance, as opposed to / in combination with the Fisher exact test?

Kindly refer to the response provided to Question 3 by Reviewer 1 for more detailed information.

17. The protein clustering involves a pretty extensive benchmark of the inflation parameter of MCL on only a couple of genomes. Is a user expected to use the inflation parameter based on this benchmark, or to do this benchmark themselves on their own subset of genomes?

The inflation parameter for protein clustering in microLife was indeed tuned and benchmarked for our specific datasets. However, it's important to note that the ideal choice of the inflation parameter may vary depending on the user's dataset, research question, and phylogenetic scope. Currently, microLife

does not incorporate an automated mechanism for users to fine-tune the inflation parameter specifically for their datasets within its framework. We are considering the inclusion of an option for users to adjust and optimize the inflation parameter according to their specific data and research requirements. This addition would empower users to fine-tune the clustering process for their datasets, potentially improving the accuracy and relevance of the results.

18. Why do you develop of novel method for protein clustering? Why not use one of the existing orthology prediction tools (potentially with tuned parameter values)?

This decision was driven by several key considerations, which we believe enhance the utility of microLife for specific applications, particularly when dealing with large datasets. One of the primary motivations behind creating microLife clustering was to provide a tool capable of efficiently handling large genomic datasets. While existing orthology prediction tools, such as OrthoFinder, are undoubtedly powerful and widely used, they operate on a pairwise comparison approach. In OrthoFinder, genes from each bacterium are aligned with genes from all other bacteria in a pairwise manner. While this approach is effective for smaller datasets, it leads to exponentially increasing computational times as the dataset size grows. The fundamental innovation of microLife protein clustering lies in its ability to circumvent this pairwise comparison bottleneck. Instead of conducting pairwise comparisons for each gene in each bacterium, microLife merges all the protein files from the dataset into one comprehensive file. It then uses efficient algorithms like DIAMOND for self-alignment. This approach significantly reduces the computational burden, especially when dealing with large genomic datasets. To assess the performance of microLife's protein clustering approach, we conducted a rigorous comparison with OrthoFinder (Extended Data Figure 21), which is considered one of the most advanced tools for protein clustering currently available.

19. Line 79 “to conclusively discriminate”: further on you contradict this statement by saying that “this approach enables classification” (line 83).

Corrected as suggested

20. Line 99 “To test the accuracy”: how do your examples test the accuracy of the tool?

We changed “To test the accuracy” for “To test the potential”

21. Line 109 “role of 13 yet unknown” à “role of 5 yet unknown”.

Corrected as suggested.

22. Line 127 “is known as the most accurate algorithm to cluster proteins by function [ref. 39]”: very old reference for this statement.

Thank you for highlighting this reference. We have updated the reference as suggested.

23. Line 154 “reported in the databases”: is this a database problem or a literature problem? As far as I understood the authors gathered information from literature.

The annotation of lifestyles at the species level in our study was primarily based on information extracted from the scientific literature. The species name is extracted from the NCBI database where we faced situations where the genomes we downloaded did not have species-level assignments readily available in the NCBI. This absence of species-level information limited our ability to assign lifestyles to these genomes.

24. Line 165 “perfect classification” this may be a good place to discuss the correlation between phylogeny (defining most of the pan-genome) and lifestyle (defining only part of the pan-genome).

We added discussion of the correlation between phylogeny and lifestyle at the end of the lifestyle prediction section (L211-219)

25. Line 170 “*B. vietnamensis*” and line 174 “*P. cichorii*”: these species is very easy to predict given that they fall phylogenetically very clearly within their respective lifestyle-type clade. Are there phylogenetic clades / species that have multiple different lifestyles (likely for *Pseudomonas*), and how does the classifier perform on them if it is not trained with them?

The classifier's performance can indeed vary when applied to phylogenetic clades or species with multiple different lifestyles, which is more common, especially among *Pseudomonas* species. For instance, let's consider the case of "*P. fluorescens*." This species represents a challenge as some of previously named *fluorescens* genomes have been renamed as other species while others kept the *fluorescens* species label and they are closely related to other species with various lifestyles, including plant pathogens and animal pathogens. In such scenarios, the classifier's predictions may be less confident compared to species like "*P. cichorii*" or "*B. vietnamensis*," which have more distinct lifestyle-type phylogenetic placements. In our analysis, we found that Random Forest performed with a high degree of accuracy and confidence for species like "*B. vietnamensis*" and "*P. cichorii*," achieving 100% and 86% confident classifications. However, for species like "*P. fluorescens*", the classifier confidently classified 74%. This observation aligns with the inherent challenge of classifying species with less clear-cut phylogenetic boundaries.

26. Line 172 “Confidence level”: the amount of agreeing decision trees is not the same as confidence level.

Thank you for your comment. It's important to clarify that these values represent the count of trees that support a particular class, rather than confidence values. We have made the necessary adjustments in the text to prevent any potential confusion. (L621-623)

27. Line 177 “we were able to accurately predict the (...) lifestyles”: you do not know if these predictions are accurate.

Corrected as suggested (L187-188)

28. In relation to earlier comments, do you predict the lifestyle or actually the phylogeny?

The data in both datasets show a clear connection between lifestyle and phylogeny at the genus level. This means we can use gene presence and absence patterns to make predictions about both lifestyle and phylogeny. However, we chose not to include an analysis of taxonomic classification in the manuscript to avoid getting into complex methodological discussions.

29. Lines 183-...: The location of the mentioned species is not clear on the PCoA, I suggest to add them. We included a PCoA of the *Burkholderia/Paraburkholderia* and *Pseudomonas* dataset colored by the most common species in Extended Data Figure 5.

30. Lines 188-189 “These findings provide additional evidence of the adaptive ability of *Pseudomonas* species to shift between lifestyles depending on the environmental conditions”. Rather it shows that with very similar genome content *pseudomonads* can do very different things. One of the premises of *microLife* is that only genes that are shared between most of these strains are LAGs, but the different clades of *pseudomonads* may do the ‘same’ niche adaptation in different ways with different genes, and these are missed by *microLife* (depending on the % inclusion parameter). It would be good to address this in the introduction.

The reviewer makes a valid observation regarding the adaptive ability of *Pseudomonas* species and the potential nuances in niche adaptation that may not be fully captured by *microLife*'s methodology. Indeed, our study has revealed that *Pseudomonads* with very similar genome content can exhibit

diverse behaviors and lifestyle adaptations. One of the foundational premises of microLife is its focus on Lifestyle-Associated Gene Clusters (LAGs) shared among a majority of strains within a lifestyle group. However, as the reviewer correctly points out, different clades of Pseudomonads may adapt to similar niches in distinct ways, involving the acquisition of different genes. These unique adaptations, while crucial for their respective ecological niches, may not be detected by microLife's default settings, which rely on a predefined threshold for gene cluster inclusion. However, microLife's analytical module offers a level of flexibility that empowers users to explore these nuances. Users can customize the percentage inclusion parameter to capture genes that might be exclusive to particular clades or subsets of strains within a lifestyle group. This feature allows for targeted investigations into the genetic variations associated with specific adaptations.

31. Line 196-204, microLife as taxonomic classifier: binary classification of a species is not taxonomic classification. It's unclear why you would want to do this? The gold standard for taxonomic classification is phylogenetic placement in a tree, which the authors do. The authors could use the taxonomic classification accuracy of the classifier to discuss the high correlation between phylogeny and gene content and what it means for a user looking for lifestyle associated genes.

We chose not to include an analysis of taxonomic classification in the manuscript to avoid getting into complex methodological discussions. Please refer to Question 1 from reviewer 1.

32. Lines 226-247 "To further test microLife's gene cluster distribution accuracy (...) aligning with findings reported earlier in the literature": it's unclear what exactly the authors are testing here. The gene distribution is a feature of the gene caller (in this case Prokka) and not of microLife at all. The results are as expected and do not have anything to do with microLife.

Prokka is responsible for gene prediction within individual genomes but doesn't inherently provide information about gene similarity or identity. To address this, microLife incorporates a clustering module that compares genes across genomes and groups together those with similar functions. Subsequently, microLife generates an absence/presence matrix, indicating the presence or absence of these gene clusters within each genome. In this specific section, our approach involves mapping Type III secretion system genes using BLAST to determine the relevant gene cluster. We then investigate and compare the distribution of this gene cluster, aligning our findings with existing literature.

33. Line 233 "exclusively": the genes are not exclusive to *P. syringae*.

Corrected as suggested (L247-249)

34. Line 250 "basic statistics": what do you mean?

More description was added in the text as suggested (L267-268)

35. Given that these are binary classifiers / comparisons, 1 for each lifestyle type, could there potentially be overlap between the classified lifestyle / identified LAGs?

Kindly refer to the response provided to Question 3 by Reviewer 1 for more detailed information.

36. Lines 259-260 "underscoring the need for studying this largely unexplored genomic information": genes in the accessory genome being often unknown is well-documented, so this statement is pretty flat. Maybe the authors could discuss here how most of these genes are part of the accessory genome instead of the core.

This was rewritten as suggested (L278-283)

37. Line 267 “genomic regions enriched in LAGs”: in a lot of places the authors talk about the genomic context of these genes. The way I understand the methods, if they talk about genes they talk about the MCL clusters, which likely contain multiple genes from different organisms, so do not necessarily have genomic context (i.e. which genes are close, their direction, etc.). Of course if you look into where these genes are present in single genomes you may talk about genomic context, and the context may differ per genome. Is the genomic context of these genes only found in specific genomes or a summary of all genomes / how did the authors decide on this? If so, is it a feature of microLife to be able to do this?

The reviewer raises an important point regarding the genomic context of genes discussed in our study. It's true that in many instances, when we refer to genes, we are actually referring to clusters produced by the MCL algorithm. These clusters can indeed contain genes from different organisms within the same lifestyle group and may not inherently possess a specific genomic context, such as neighboring genes, their direction, or organization. To address this, we ensured that when discussing the genomic context of genes or the arrangement of Lifestyle-Associated Gene Clusters (LAGs), we considered their presence in multiple representative genomes (at least two genomes from the same lifestyle), rather than relying solely on a single genome's context. This approach allows us to make more robust observations about the recurring patterns of LAG presence and arrangement within a particular lifestyle category.

Within the microLife analytical module, we provide the means for users to explore the genomic context of these genes in a broader context. After performing statistical analyses, users have the option to select a specific bacterium from the dataset. This feature allows them to examine the arrangement of genes within that chosen bacterium, taking into account the filtering thresholds applied (>70% group presence and >2 log₂fold change in this study).

38. Line 271-282 “ToxC and ToxD”: I could not find these genes in Extended Data Table 4.

ToxC and ToxD gene names have been included in Extended Data Table 4 as suggested.

39. Line 276 “consecutive genes”: see earlier comment about genomic localisation, how did the authors define this / in which genome / are they colocalised in an X number of genomes? If finding genomic contexts of MCL clusters is a feature of microLife, it is nice because it takes away a potentially complicated bioinformatic analysis and deserves a bit more attention.

As previously addressed in response to question 37, the microLife analytical module provides the capability for users to select a specific bacterium from the dataset. It then displays the LAGs that are present in that particular bacterium, along with the respective order of appearance in the genome for each instance of a LAG. This functionality enables users to identify consecutive LAGs by simply sorting the results table based on their genomic positions. To confirm these consecutive regions we looked into at least two bacteria with the same lifestyle that are not phylogenetically close.

40. Lines 288-289 “Our results showed that in *B. plantarii* DSM 9509, 210 out of 786 LAGs had a significant hit to PHI-BASE.” Why only in this single genome? Please give a summary of PHI-BASE hits in all genomes. Idem for *Pseudomonas* that is discussed later.

We revised our analysis strategy concerning the PHI-base. In this updated approach, we select the representative sequence from each gene cluster and subject it to a BLAST search against the PHI-base (evalue < 1e-2). Subsequently, we compare the relative prevalence of significant hits between phytopathogenic LAGs to the entire pan genome of the genus. (L322-335)

41. Also, it's unclear how Extended Data Table 4 shows this. Doesn't the table contain the MCL clusters instead of the genes from a single genome?

Extended Data Table 4 contains the MCL clusters and the genomic position in *B. gladioli* GSRB05. Note that the first 17 rows contain "0" in the column "Genome position *B. gladioli* GSRB05" which means that these MCL clusters considered LAGs are not present in this specific genome. We ordered the LAGs by position in this genome as an example to show the genomic regions identified and highlighted in the table. *B. gladioli* GSRB05 was used because it contains most of the LAGs of plant pathogenic *Burkholderia*. Same applies to Extended Data Table 5 and *P. syringae* 7969.

42. Could the authors comment on why not all genes with PHI-base hits are found back in the LAGs? For example for *Pseudomonas* only 118 out of 377 PHI-base hits are found in the LAGs.

In this revised approach, we identified only 253 out of 4,788 significant hits to the PHI-base within phytopathogenic LAGs in *Burkholderia*, and 142 out of 4,693 within *Pseudomonas*. Notably, while we have a higher proportion of hits within our LAGs, the majority of these hits were observed in the pangenome. This phenomenon can be attributed to the filtering criteria we applied, specifically the requirement of >70% group presence and >2 log₂fold change to define a LAG. It's worth noting that genes excluded by this filter may still play roles in interactions with the host. These genes could include those present in only a few bacteria and are pathovar-specific, as well as genes found in most bacteria but are involved in host interactions only in certain strains. Consequently, these genes may not satisfy the rigorous LAG criteria, highlighting a limitation of microLife, which primarily operates based on gene distributions. However, these genes can still hold relevance in the context of host-pathogen interactions, as the reviewer indicated.

43. Line 304 "consecutive": in which genome?

The reference genome in the analysis was *Burkholderia plantarii* ATCC 43733. We have added this information to the reviewed manuscript. (L343)

44. Line 305 "were positioned in the same transcriptional direction". Is the transcriptional direction within a genome an output in the R Shiny app (see earlier comments about genomic context)? Or do you need more bioinformatic skills for that? Again I think the sole benefit of microLife is that it makes common analyses easy to do for non-bioinformaticians, so if this genomic-context analysis is not part of microLife it's unfair to give that as an example of the benefits of microLife.

While microLife primarily is designed to simplify routine genomic analyses for non-bioinformaticians, such as gene prediction, functional annotation and comparative genomics. Therefore, it may not encompass intricate genomic context analyses, such as the determination of the transcriptional direction of genes within a genome. Nevertheless, it is worth noting that the tool's developers are committed to enhancing the user experience and integrating more advanced features in future iterations.

45. Line 308 "*Burkholderia* genera": strains or genomes?

Corrected as suggested (L346)

46. Line 311-312 "5 out of 13 generated mutants were not affected in growth as compared to the wild type strain" What does this say about the other 8 genes and about microLife's performance of identifying 'real' LAGs?

microLife is a tool designed for the identification of putative genes associated with a specific bacterial lifestyle. In our study, 5 out of 13 selected genes exhibited growth patterns similar to the wild-type strain, and subsequent plant experiments confirmed their involvement in the plant pathogenic lifestyle. Regarding the remaining 8 LAGs, it was evident that mutations within these genes had noticeable effects on bacterial growth, suggesting their potential relevance to the bacterial lifecycle. We decided to not subject these mutants to plant experiments, as their reduction in virulence may be correlated with reduced bacterial growth. Consequently, we refrained from definitively categorizing these genes as LAG within the context of plant pathogenicity. Nevertheless, it is reasonable to infer that these 8 LAGs are intricately involved in fundamental processes necessary for proper bacterial development and may have implications for plant virulence. Further investigations are required to fully elucidate the precise roles and significance of these genes in the context of bacterial lifestyle and host interactions. But, one aspect that remains unquestionable is the value of microLife as an effective tool for the identification of candidate LAGs.

47. Line 329 “exclusively”: is this true? Exclusive means ‘only there, nowhere else’.

Extended Data Figure 10 shows the distribution of the LAGs where mutagenesis is performed. There are few hits in non plant pathogen bacteria so we removed “exclusively” (L366-367)

48. Line 347-351 “Remarkably, microLife was more efficient in the identification of genes involved in virulence when LAGs were present in LAG-enriched regions”. This is not a feature of microLife but rather of the way HGT works, operons, etc. There is no reason microLife should be efficient in identifying clustered genes and it should not be implied. Second, whether this is a biologically relevant observation depends on how random the selection of the mutagenesis LAGs was. If the authors want to make a point about LAGs being more often clustered, they should analyse all LAGs on all genomes. We harnessed the workings of horizontal gene transfer (HGT) and operons to introduce an extra filter for pinpointing lifestyle-associated regions. While we can't definitively assert that LAGs frequently cluster together, our experimental validation underscores the enhanced potential for discovering crucial lifestyle-associated regions when LAGs reside within enriched regions, potentially representing operons or HGT groups, rather than isolated occurrences.

49. Line 362 “majority”: the majority means ‘more than half’.

Corrected as suggested (L401)

50. Line 367 “we observe a trend”: add statistics.

We conducted a t-test comparing the quantity of NRPSs in plant pathogen bacteria to that in opportunistic animal pathogens and environmental samples for *Burkholderia/Paraburkholderia* and *Pseudomonas*. In all instances, the comparisons yielded significant results at a significance level of $\alpha = 0.01$, indicating that plant pathogens possess a notably higher number of NRPSs compared to opportunistic animal pathogens and environmental bacteria. (L406-407)

51. When looking at the biosynthetic gene clusters, why is only 15% group presence required for LAG definition, vs 70% for the MCL clusters?

BGC clusters or GCFs showed a way more sparse distribution than gene clusters making most of the GCF to be present only in a few genomes. We decided to use a threshold of 15% based on the distribution pattern observed.

52. It may be good to discuss why there are more GCF LAGs found in *Burkholderia* vs *Pseudomonas*. I think it relates to the phylogeny versus lifestyle concern raised before.

Good point. The number of LAGs in *Burkholderia* is larger potentially because of the correlation with phylogeny, This discussion is added into the main text (L273-275)

53. Line 381 “discovery of a highly conserved NRPS-PKS”. What do you mean with highly conserved?
We change it for “plant pathogen associated NRPS-PKS” (L422)

54. Line 385: “combined with a BLAST search”. It’s unclear why the authors do a BLAST search here. The presence of the clusters should come from the clustering module.

We use BLAST for confirmation of the gene cluster distribution we observed with microLife

55. The NRPS-PKS is said to have a similar distribution to syringofactin and syringomicyn, but I really do not see that in Fig. 7b.

Change to similar distribution in *P. syringae* species and not the whole syringae complex

56. The methods section is not clear on what’s part of a standard microLife pipeline and what’s added custom for this paper. For example, the CheckM and genome statistics, and the type of figures that are generated in the R Shiny module, do they include phylogenetic trees and iTOL annotation files, etc. It’s a relevant distinction, because the main benefit of this tool would be the user-friendliness and not the biological results that are presented in this manuscript—which could have been generated without the tool.

We indicated in Material & Methods when a type of analysis (phylogenetic analysis and quality check) is not performed by microLife.

57. Line 486 “uniq fasta file”: do you mean a “fasta file with unique sequences”?

Prokka predicts one amino acid fasta file for each genome. we combined all these files in one large fasta files that is used as input for the clustering. We rewrote it in the revised manuscript to clarify (L540)

58. Line 518-522 “It is worth noting that the origins of these unique genes can differ...”. Or alternatively the clustering is not complete.

There's a possibility that these unique genes lacked sufficient sequence similarity with other genes to be part of a gene cluster with multiple sequences. Nevertheless, in the text, we discuss the potential origin of these sequences under the assumption that they are indeed unique.

59. Also note that there is a reference left in line 522.

Corrected in revised manuscript.

60. Lines 523-533 “gene functional annotation”. I think annotating the MCL clusters with different functional databases is nice and something users will appreciate. It can be mentioned in the main text as well.

While gene functional annotation serves a general utility for users, it is not the primary focus of the pipeline. Consequently, we opted to retain this aspect within the methodology section.

61. It's unclear in the methods which databases are used, because it says 'such as DBCAN and PFAM', which suggest there are more. Please mention all.

As recommended, we have provided clarification in the methods section. (L585)

62. Adding InterProScan would be a comprehensive next one.

Indeed. Future iterations of the tool could incorporate new databases for functional annotation.

63. Lines 567-569: please explain how unsupervised approaches are useful for clade/species-specific genes.

The unsupervised methods incorporated into the microLife app prove valuable for grouping bacteria with similar gene compositions. This is achieved through clustering based on the PCoA, which is constructed using the gene absence/presence table. This explanation was included in the revised manuscript (L647-650)

64. "Dice dissimilarity" Do you mean the Sørensen–Dice coefficient? Why was this measure chosen? We opted for the Dice dissimilarity metric because it is suitable for binary data and closely resembles the widely-used Jaccard distance. Like the Jaccard index, the Sørensen–Dice similarity coefficient also gauges the ratio of the intersection to the total elements in both sets. However, there are important distinctions. In Sørensen–Dice, the denominator consists of the sum of set cardinalities, unlike the Jaccard index, which uses the union of the sets. Additionally, Sørensen–Dice places double weight on the intersection in the numerator, subtly emphasizing commonalities (which are of greater interest to us) between sets over differences.

See this post for more information: <https://towardsdatascience.com/similarity-measures-and-graph-adjacency-with-sets-a33d16e527e1>

65. Throughout the manuscript, whenever there is a phylogeny it should be clear how the tree is generated, how it is rooted, whether it is a maximum likelihood tree, etc. Just referring to PhyloPhlAn in the methods is not good enough. Also add support values.

More information about the details of the phylogenetic analysis has been included in the Material and Methods section (L685-696)

66. The method section is also unclear about marker gene selection and alignment.

Please refer to the answer to the previous question.

67. Fig. 3: What's on the y-axis of the rings?

The y-axis illustrates the variation from the dataset's average relative abundance for each COG category. Negative values indicate that the COG category has a lower relative proportion compared to the dataset's average. This clarification has been provided in the figure legend.

68. Fig. 3: Why is the Z (cytoskeleton) GO term omitted? It is part of the top significantly associated COG category in Supplementary Table 3.

Please see the answer and question 7 from reviewer 1. COG categories A and Z were removed from the analysis because they are mainly based in Eukaryota

69. Fig. 3: please put the GO term letters in the figure behind the terms instead of in the caption.

GO terms letters were added in the revised Figure 3.

70. Fig. 3: Why are the CDS in this figure?

CDS were removed from Figure 3.

71. Supplementary Table 3: Can you put the GO terms in the table as well, not just the letters?

GO terms were added to supplementary table 3.

72. Supplementary Table 3: Sort the Pseudomonas case according to P value.

Corrected in the revised supplementary table 3.

73. Fig. 4: How did you define the *P. syringae* complex?

The syringae complex contains strains from species such as *P. syringae*, *P. savastanoi*, *P. viridiflava* and *P. cichorii*. This complex is known for their plant pathogenic lifestyle but also contains environmental strains.

- Gutiérrez-Barranquero, J. A., Cazorla, F. M., & De Vicente, A. (2019). *Pseudomonas syringae* pv. *syringae* associated with mango trees, a particular pathogen within the “hodgepodge” of the *Pseudomonas syringae* complex. *Frontiers in Plant Science*, 10, 570.

74. Fig. 5: Why not show the distribution for all genes?

Unique genes and pangenome were added to Figure 5.

75. Fig. 5: How do you define ‘function known’?

We define it as gene clusters with a COG annotation given by the eggNOG-mapper tool.

76. Fig. 6: the caption says (n=845). I only count 17 branches. The caption also says “of the 15 selected LAGs”, but it should be 13.

Corrected as suggested

77. Fig. 7a: it says that the BGC are coloured based on lifestyle but it’s based on species.

Corrected in the revised manuscript.

78. Line 1187: It’s unclear what’s meant by “significantly”.

The term “Significantly assigned” was changed to “highly associated”

79. Fig 7c: How are the 3 representative BCGs selected?

With microLife we detected two GCFs to be significantly associated with plant pathogen bacteria. BGC networks from BiG-SCAPE were loaded into Cytoscape, and upon examination of these two GCFs in the network we found another GCF family associated with them in the network that presented the

same predicted BGC product and structure but in non pathogenic bacteria such as *P. fluorescens* and others spp. One representative bacteria from each GCF was chosen as representative and used to generate the phylogenetic tree in Figure 7c.

80. Table 1: What do you mean with “Annotation”? COG annotations?

The term “Annotation” was changed to “NCBI Annotation”.

81. Extended Data Fig. 4: It’s unclear which lifestyle are the left-middle-right panels.

Corrected in the revised manuscript.

82. Extended Data Fig. 4: Why is the mean of the ROC curves not 1 for the Burkholderia case?

The package scikit-learn was used to calculate the AUC values. This package doesn’t round to 1 when calculating the mean AUC values (expand or change)

83. Extended Data Fig. 7: Please put names of the GO terms in the figure, not just in the caption.

GO terms were added to what is now Extended Data Figure 6 in the revised manuscript.

84. Extended Data Fig.7: Change order of lifestyles between panel a and b so that they are consistent.

This was corrected as suggested

85. Extended Data Fig. 8: Fig. 3: Why is the Z (cytoskeleton) GO term omitted? It is part of the top significantly associated COG category in Supplementary Table 3.

Please see the answer and question 7 from reviewer 1. COG categories A and Z were removed from the analysis because they are mainly based in Eukaryota

86. Extended Data Fig. 10: This is a nice figure, is it coming directly from microLife?

While the microLife app enables the identification of consecutive LAG regions using positional information, it relies solely on this data. Further investigation into these regions, including transcription direction, was conducted using other genome visualization tools such as IGV (include reference).

87. Extended Data Fig. 15: How is the clustering of the heatmap done? Why not cluster according to the phylogeny like the other plots?

Heatmap clustering is performed by the package heatmap.2 based on a distance matrix calculated with the input data given for the function.

88. Extended Data Fig. 18: How did you define the “top 10 functional type categories”?

The term “top 10 functional type categories” was changed to “top 10 more abundant functional type categories”

89. I would advice to remove hollow excitement generating words like “remarkably”, “interestingly”, “last but not least”, “notably”, “enormous [pharmaceutical potential]”, and “the framework can be extended to eukaryotic organisms, in particular fungi and yeast”. 😊

We appreciate the constructive feedback about the tone of the text. We have revised the manuscript to remove the subjective language.

90. Maybe I missed it, but I could not find the Random Forest results in the R Shiny app.

The lifestyle prediction module from microLife currently operates as a separate code due to compatibility issues with the app software. However, we intend to integrate this module into the Shiny app in the future. This integration will allow users to employ the random forest algorithm and extract important variables interactively in a similar/parallel way as the Fisher test.

Reviewer #2 (Remarks to the Author):

The revised version of the manuscript dealt with most of my initial concerns, or justified and explained some others in their responses. I am so grateful to read such detailed responses. Still, there is a key point that remains questionable. That is the initial categorization of lifestyles based on species definition. The authors correctly argue that the main point of this manuscript is to describe the new tool (MicroLife) and that any researcher can use their own metadata selection. However, another important message of this manuscript is the definition of LAGs. While surely most of your LAGs could be indeed real LAGs as partially demonstrated by their wet lab experiments, I am afraid of that message that suggests the overall findings of this study as true LAGs. There might be many LAGs that are just phylogenetic noise, as the authors found in the comparison with other tools such as phyloGLM. However, as I believe that this case study might not affect the performance of MicroLife, I suggest to specify these biases in both the abstract and the discussion, and to really underscore that the predicted LAGs should not be considered as true until further experimental validation. Indeed, while authors demonstrate that mutants' growth is not altered in vitro conditions, this might not be true in plant tissues. This possible event might reflect that these LAGs do not represent pathogenicity genes, but just genes that provide adaptation to that environment (leaves). Hence, it is possible that mutants colonize leaves worse than WT strains. While this case would be somehow related with the pathogen performance, the significance of the results would be really different, and might indicate that these genes could also be found in plant-associated or leave-associated strains, including beneficial, commensal or pathogenic strains.

Reviewer #3 (Remarks to the Author):

Guerrero-Egido and co-authors have revised their manuscript based on the reviewer's comments. They have notably removed the taxonomic predictions from their study and changed or added more analysis to address some of the raised issues, and changed parts of the way they present their tool and results. Their rebuttal was detailed, and I appreciate the effort they have put into it. It's also nice that the developers seem eager, although only in the future (😊), to add new features.

However, I still have concerns with the way their tool is presented. I do acknowledge that not everything can be done and we may disagree on some points. However I do think a good discussion on the LAGs is still lacking, in particular on the relation between 'lifestyle-associated genes' and 'genes involved in niche adaptation', and the related debate on phylogenetic signal in LAGs. And that this discussion is essential to an understanding of what microLife can and cannot do.

Having said that, I do believe that this does not require more analyses per se, but a better explanation in the text suffices. I noticed that the authors are very convincing in their rebuttal, but in some important cases did not change anything in the manuscript. I made my earlier remarks and questions to improve the manuscript, not to be convinced in a rebuttal text.

Below are the concerns I still have, numbered according to the questions in the earlier review (I'm sorry, the system only allows me to upload plain text so this I think is better than uploading all responses, even if you need to go back and forth between this file and the previous file). I hope they are useful in improving the manuscript!

1.

I appreciate the extra analyses the authors have done to address the influence of phylogeny on their results in the example cases of *Pseudomonas* and *Burkholderia* and explain this in their rebuttal. They have also added some extra sentences in the manuscript here and there. However I did not raise this point to be convinced that microLife does it correct in these cases, but because

only using a Fisher's exact test to identify candidate genes is a choice in microLife that influences the type of genes you identify, and that users should be aware of. The ANI reduction only solves this problem on the strain level. What kind of genes (in the context of lifestyle prediction) may you identify based on different correlations between phylogeny and lifestyle? Again, I'm not asking this to get an answer in a rebuttal, but because I think it should be addressed in the manuscript so that users understand. The added text in line 211-219 would be a good place to extend a bit more. The authors conclude that "This high correlation between lifestyle and phylogeny in both datasets poses a challenge in distinguishing between genes associated with a genuine lifestyle impact and those associated primarily with phylogeny." Please expand on this. It would show confidence to also mention other tools that did solve the phylogeny issue here. It doesn't invalidate your tool (which does many things nicely!), but being honest about what it can and cannot do really helps. I also feel that the text in the rebuttal (here and in Q4 by rev1) is more insightful than what has been added to the manuscript at different places. The case of LAGs 14, 22 and 23 being entirely correlated with phylogeny is a nice point that phylogeny correction may also miss things, and this discussion for example is super useful, not the question whether microLife is good or not (it obviously depends on the use case).

I found the remark in the rebuttal that the genomic context of the LAGs matters highly speculative but also very interesting. The authors give no biological explanation for this in the rebuttal or in the manuscript. If that is the feature of microLife that makes it better than other tools in identifying LAGs with real lifestyle-adaptation genes, they should explain a bit more. Again I envision this as an extended paragraph and insights here and there about the LAG definition (for example at line 145 as well, and line 275) and a discussion of what kind of genes microLife may identify, which in the latest manuscript version is still lacking. I think the explanation that I am missing currently is that a gene may be "lifestyle-associated" but that may not necessarily make it "involved in lifestyle adaptation", and the other way around, not all genes involved in lifestyle adaptation will be found by your definition of lifestyle-associated genes.

4.

This response does not address my concern. You have only added the classifier scores to the "unknown" strains. I was especially interested in the classification of the known cases by the three different classifiers. Can you add them and are they always correct (i.e. 1 for the known lifestyle and very low for the other 2)? This may help in explaining where the classifier likely goes wrong (e.g. a single outlier lifestyle within a clade). Again I do not want this just for the rebuttal but as a short discussion added to the manuscript.

7.

Please add the rebuttal text to the manuscript as well. Given that you specifically searched for regions that were enriched in LAGs, consider removing / rewriting text at multiple places in the manuscript that states that microLife is better at detection of LAGs that were enriched in LAGs. MicroLife allows you to do this (which is nice, I really like the genomic context angle!), but the selection is done by the user and doesn't have biological implications per se. If you think that there are biological implications in the sense that if you pick LAGs from regions that are enriched in LAGs you have a higher chance of finding a "true" lifestyle-adaptation gene, you should elaborate in the manuscript on why this may be the case (see also point 1 and point 48 in my previous review). It's ok to be speculative.

8.

Please add the short discussion about the house-keeping genes to the manuscript instead of just keeping it in the rebuttal. It goes a long way in explaining the difference between the lifestyle-associated genes and genes involved in lifestyle adaptation (see point 1).

9.

I like the new title a lot, great! 😊

11.

I appreciate that you have added an extra couple of sentences to the introduction but the first sentence + yellow highlighted text does not read nicely anymore, and contains vague language that makes it difficult to interpret what MicroLife aims to do. Please consider rewriting with only one

or two concise messages per sentence. In particular the link between function, lifestyles, role, phenotype, "niche function", "niche diversity", etc., which seem to refer to the same thing, is vague. It's important to define 'lifestyles' and how to relate to genotype so that the reader knows what Microlife aims to do. In the abstract you talk about "genes involved in niche adaptation" which I think is nice and bridges the gap between gene and phenotype. It's those genes you try to find, but they are slightly different from the "lifestyle-associated genes" that you define, which may not all be involved in niche adaptation, and genes involved in niche-adaptation may not be LAGs (see also point 1). If you can make that clear throughout the manuscript, I don't care where, I'm happy.

12.

Please add this ("what do you need for a successful microLife run") not only to the rebuttal but also to the manuscript. It's important for any user, especially the idea of scale and the final sentence ("Crucial for the success rate...").

13.

I meant that you should add a table that has a list of references from this literature survey, that shows how you got for example to the conclusion that *Burkholderia aenigmatica* AU17325 is an opportunistic animal pathogen and *Pseudomonas coleopterorum* LMG28558 is environmental. It's fine if it's on the species level, I understand that. But right now it's unclear where you base those conclusions on. Especially since the metadata is instrumental for a successful microLife run, a user should be able to see how you have done it yourself. Besides, anybody should be able to check your judgement calls.

15.

Wouldn't the gene clustering which is the basis of microlife differ depending on taxonomic rank, as sequences may be more diverged? If there would be difference between different taxonomic ranks, please add those insights to the manuscript.

17.

This rebuttal text about the manual selection of the inflation parameter is an important insight that should be added to the manuscript, so that a user knows. As far as I'm concerned the (almost) literal rebuttal text is good for this.

23.

Consider rewriting to "A large number of genomes in both datasets were not assigned to any lifestyles because they did not have species-level assignments in NCBI, which may compromise statistical power in the further steps of the microLife pipeline." Or something similar.

25.

Please add this insight not just in the rebuttal but also in the manuscript.

28.

It's a good decision to remove the taxonomic classifier. Note that there are remnants of it still in the manuscript (e.g. line 101, line 495, line 605).

30.

Please add this insight to the manuscript as well (see earlier points about the type of genes that microLife may detect).

32.

I understand what you say, but it's still unclear to me why you are doing this. Please consider at least a rewrite in the manuscript as to the reasons for this analysis. I might not be the only one that doesn't understand.

35.

I actually meant if a single genome could be annotated as both environmental and plant pathogenic for example, or that a LAG could be both environmental and plant pathogenic associated.

36.

Thanks, very nice!

37.

Please add this insight (and also those in the rebuttal of 39) to the manuscript as well. Also, please make sure that at *every* place where you talk about genomic context, "consecutive genes" etc, it's clear how you define this (for example by stating the genomes in which the genomic content is evaluated). Right now this information is added to some places, but not *all* yet.

40.

Very nice, well done!

42.

Please add this insight to the manuscript as well. It goes well with discussing the strength and weaknesses of using the LAG definition.

46.

I agree with the authors that these 8 genes may not be genes involved in pathogenicity (or at least we do not know for sure), and I think they are a good example of genes that may be identified because of the phylogenetic dependence, i.e. they are lifestyle-associated but not necessarily involved in niche adaptation. Please consider using these as an example for the suggested extra discussion about the type of genes that microLife may detect.

58.

Please add this constraint to the manuscript that some of the unique genes may be an artifact of the clustering parameters. I'd like that because the clustering is an important part of microLife.

64.

Please add the reasoning for the choice of Soerensen Dice to the manuscript. I understand that you normally don't have to defend these kind of choices but since it's part of the pipeline I do believe it's important in this case.

68.

Please add the reason for removing these categories to the manuscript.

75.

Please add to the figure caption.

79.

Please add this to the figure caption.

86.

Please add this to the figure caption.

87.

Please add this to the figure caption.

89.

While I appreciate that you may not always agree with reviewers and cannot do everything, I did not like that the authors suggest that they have done this whereas in reality they have not.

Dear reviewers,

Thank you very much for your valuable input. We have carefully addressed each of your concerns, incorporating your suggestions to refine our manuscript significantly. Your feedback has undoubtedly elevated the quality of our work.

*I would also like to bring to your attention a noteworthy change – we had to modify the name from *microLife* to *baCLIFE*. This adjustment was prompted by the editor's observation in the previous review regarding a company and a journal with the name *microLife*. Following advice from our legal department and direct communication with the company and the journal, we opted for this name change.*

*Best regards,
V́ctor Carrión*

REVIEWER COMMENTS

Reviewer #2 (Remarks to the Author):

The revised version of the manuscript dealt with most of my initial concerns, or justified and explained some others in their responses. I am so grateful to read such detailed responses. Still, there is a key point that remains questionable. That is the initial categorization of lifestyles based on species definition. The authors correctly argue that the main point of this manuscript is to describe the new tool (MicroLife) and that any researcher can use their own metadata selection. However, another important message of this manuscript is the definition of LAGs. While surely most of your LAGs could be indeed real LAGs as partially demonstrated by their wet lab experiments, I am afraid of that message that suggests the overall findings of this study as true LAGs. There might be many LAGs that are just phylogenetic noise, as the authors found in the comparison with other tools such as phyloGLM. However, as I believe that this case study might not affect the performance of MicroLife, I suggest to specify these biases in both the abstract and the discussion, and to really underscore that the predicted LAGs should not be considered as true until further experimental validation. Indeed, while authors demonstrate that mutants' growth is not altered in vitro conditions, this might not be true in plant tissues. This possible event might reflect that these LAGs do not represent pathogenicity genes, but just genes that provide adaptation to that environment (leaves). Hence, it is possible that mutants colonize leaves worse than WT strains. While this case would be somehow related with the pathogen performance, the significance of the results would be really different, and might indicate that these genes could also be found in plant-associated or leave-associated strains, including beneficial, commensal or pathogenic strains.

We appreciate the reviewer's feedback and insightful suggestions. In response to the concerns raised regarding the categorization of Lifestyle-Associated Genes (LAGs), we have implemented a distinction between predicted LAGs (pLAGs) and true LAGs, which are validated experimentally. This differentiation underscores the inherent limitations of all methods, including phyloGLM, treeWAS, and baCLIFE, in predicting true LAGs without experimental confirmation. We have explicitly addressed this point in both the abstract and the discussion to highlight the potential presence of phylogenetic noise and the necessity for further experimental validation before considering any gene as a true LAG.

Regarding the mutants with altered growth *in vitro* conditions, we acknowledge the complexity of disentangling colonization from pathogenicity. To address this challenge, we explained in detail our rationale for using *in vitro* assays to filter out mutants compromised in growth. We emphasized the interconnected nature of colonization and pathogenicity, illustrating the chicken-and-egg dilemma inherent in these processes. While we recognize the possibility of differences in colonization in plant tissues, we clarified the practical challenges of testing multiple plant species, which is beyond the scope of this study.

We have incorporated these clarifications into the manuscript to ensure transparency and prevent any potential confusion among readers (lines 421-426). We sincerely thank the reviewer for raising these important points, and we believe that these revisions enhance the overall clarity and robustness of our manuscript.

Reviewer #3 (Remarks to the Author):

Guerrero-Egido and co-authors have revised their manuscript based on the reviewer's comments. They have notably removed the taxonomic predictions from their study and changed or added more analysis to address some of the raised issues, and changed parts of the way they present their tool and results. Their rebuttal was detailed, and I appreciate the effort they have put into it. It's also nice that the developers seem eager, although only in the future (😊), to add new features.

However, I still have concerns with the way their tool is presented. I do acknowledge that not everything can be done and we may disagree on some points. However I do think a good discussion on the LAGs is still lacking, in particular on the relation between 'lifestyle-associated genes' and 'genes involved in niche adaptation', and the related debate on phylogenetic signal in LAGs. And that this discussion is essential to an understanding of what microLife can and cannot do.

Having said that, I do believe that this does not require more analyses per se, but a better explanation in the text suffices. I noticed that the authors are very convincing in their rebuttal, but in some important cases did not change anything in the manuscript. I made my earlier remarks and questions to improve the manuscript, not to be convinced in a rebuttal text.

Below are the concerns I still have, numbered according to the questions in the earlier review (I'm sorry, the system only allows me to upload plain text so this I think is better than uploading all responses, even if you need to go back and forth between this file and the previous file). I hope they are useful in improving the manuscript!

1.

I appreciate the extra analyses the authors have done to address the influence of phylogeny on their results in the example cases of *Pseudomonas* and *Burkholderia* and explain this in their rebuttal. They have also added some extra sentences in the manuscript here and there. However I did not raise this point to be convinced that microLife does it correct in these cases, but because only using a Fisher's exact test to identify candidate genes is a choice in microLife

that influences the type of genes you identify, and that users should be aware of. The ANI reduction only solves this problem on the strain level. What kind of genes (in the context of lifestyle prediction) may you identify based on different correlations between phylogeny and lifestyle? Again, I'm not asking this to get an answer in a rebuttal, but because I think it should be addressed in the manuscript so that users understand. The added text in line 211-219 would be a good place to extend a bit more. The authors conclude that "This high correlation between lifestyle and phylogeny in both datasets poses a challenge in distinguishing between genes associated with a genuine lifestyle impact and those associated primarily with phylogeny." Please expand on this. It would show confidence to also mention other tools that did solve the phylogeny issue here. It doesn't invalidate your tool (which does many things nicely!), but being honest about what it can and cannot do really helps. I also feel that the text in the rebuttal (here and in Q4 by rev1) is more insightful than what has been added to the manuscript at different places. The case of LAGs 14, 22 and 23 being entirely correlated with phylogeny is a nice point that phylogeny correction may also miss things, and this discussion for example is super useful, not the question whether microLife is good or not (it obviously depends on the use case).

I found the remark in the rebuttal that the genomic context of the LAGs matters highly speculative but also very interesting. The authors give no biological explanation for this in the rebuttal or in the manuscript. If that is the feature of microLife that makes it better than other tools in identifying LAGs with real lifestyle-adaptation genes, they should explain a bit more. Again I envision this as an extended paragraph and insights here and there about the LAG definition (for example at line 145 as well, and line 275) and a discussion of what kind of genes microLife may identify, which in the latest manuscript version is still lacking. I think the explanation that I am missing currently is that a gene may be "lifestyle-associated" but that may not necessarily make it "involved in lifestyle adaptation", and the other way around, not all genes involved in lifestyle adaptation will be found by your definition of lifestyle-associated genes.

We expanded the discussion on the challenges associated with distinguishing phylogeny from lifestyle-associated genes throughout the text. This is initially introduced in lines 217-226 following the general analysis of COG categories and the presentation of the phylogenetic tree. Further elaboration is provided in lines 283-288, introducing other tools that consider phylogeny and presenting a detailed comparison between methods, as outlined in the materials and methods section. In lines 413-421, we also address the correlation with phylogeny concerning the LAGs chosen for experimental validation. Additionally, in lines 432-434, we suggest that focusing on LAG-enriched regions may serve as a potential workaround to filter for more intriguing LAGs, although this remains speculative.

4.

This response does not address my concern. You have only added the classifier scores to the "unknown" strains. I was especially interested in the classification of the known cases by the three different classifiers. Can you add them and are they always correct (i.e. 1 for the known lifestyle and very low for the other 2)? This may help in explaining where the classifier likely goes wrong (e.g. a single outlier lifestyle within a clade). Again I do not want this just for the rebuttal but as a short discussion added to the manuscript.

The classification scores for all genomes have been incorporated into Supplementary Data Table 2. Additionally, a brief discussion has been included (L188-192), featuring two examples of *P. fluorescens* initially labelled as environmental. These instances stand out as outliers within the broader *fluorescens* category, receiving low confidence values in being environmental due to their proximity to *P. syringae* genomes. This underscores the challenges associated with annotating genome lifestyles at the species level.

7.

Please add the rebuttal text to the manuscript as well. Given that you specifically searched for regions that were enriched in LAGs, consider removing / rewriting text at multiple places in the manuscript that states that microLife is better at detection of LAGs that were enriched in LAGs. MicroLife allows you to do this (which is nice, I really like the genomic context angle!), but the selection is done by the user and doesn't have biological implications per se. If you think that there are biological implications in the sense that if you pick LAGs from regions that are enriched in LAGs you have a higher chance of finding a "true" lifestyle-adaptation gene, you should elaborate in the manuscript on why this may be the case (see also point 1 and point 48 in my previous review). It's ok to be speculative.

The rebuttal text has been incorporated into the manuscript (L301-303 and 364-373), highlighting our emphasis as bacLIFE users on giving more importance to LAG-enriched regions for identifying potential operons. However, it's worth noting that we also conducted tests specifically focusing on 'solo' LAGs.

8.

Please add the short discussion about the house-keeping genes to the manuscript instead of just keeping it in the rebuttal. It goes a long way in explaining the difference between the lifestyle-associated genes and genes involved in lifestyle adaptation (see point 1).

The rebuttal discussion on the correlation between LAGs and the phylogeny of the LAGs chosen for experimental validation has been included in the main text. (L413-421)

9.

I like the new title a lot, great! 😊

11.

I appreciate that you have added an extra couple of sentences to the introduction but the first sentence + yellow highlighted text does not read nicely anymore, and contains vague language that makes it difficult to interpret what MicroLife aims to do. Please consider rewriting with only one or two concise messages per sentence. In particular the link between function, lifestyles, role, phenotype, "niche function", "niche diversity", etc., which seem to refer to the same thing, is vague. It's important to define 'lifestyles' and how to relate to genotype so that the reader knows what MicroLife aims to do. In the abstract you talk about "genes involved in niche adaptation" which I think is nice and bridges the gap between gene and phenotype. It's those genes you try to find, but they are slightly different from the "lifestyle-associated genes" that you define, which may not all be involved in niche adaptation, and genes involved in niche-adaptation may not be LAGs (see also point 1). If you can make that clear throughout the manuscript, I don't care where, I'm happy.

We simplified certain sentences in this section for better readability. Additionally, in the introduction, we clarified that genes involved in the lifestyle indeed refer to genes associated with the specific function of the bacterium in its particular niche (L49-56).

12.

Please at this (“what do you need for a successful microLife run”) not only to the rebuttal but also to the manuscript. It’s important for any user, especially the idea of scale and the final sentence (“Crucial for the success rate...”).

It has been incorporated into the manuscript. We are highlighting the potential applicability of bacLIFE in smaller datasets. It is explicitly mentioned that for reliable statistics, an approximate minimum of 10 genomes per lifestyle is recommended.(L554-568)

13.

I meant that you should add a table that has a list of references from this literature survey, that shows how you got for example to the conclusion that Burkholderia aenigmatica AU17325 is an opportunistic animal pathogen and Pseudomonas coleopterorum LMG28558 is environmental. It’s fine if it’s on the species level, I understand that. But right now it’s unclear where you base those conclusions on. Especially since the metadata is instrumental for a successful microLife run, a user should be able to see how you have done it yourself. Besides, anybody should be able to check your judgement calls.

In the manuscript, we have incorporated Extended Data Table 10, which contains annotated species along with the references utilized for annotating their lifestyles. This provides prospective users with insight into our approach, demonstrating how we leveraged the literature to annotate the bacterial lifestyles of interest.

15.

Wouldn’t the gene clustering which is the basis of microlife differ depending on taxonomic rank, as sequences may be more diverged? If there would be difference between different taxonomic ranks, please add those insights to the manuscript.

The gene clustering in bacLIFE is responsive to the characteristics of the input dataset. When employing a dataset with a higher taxonomic rank, resulting in increased sequence divergence, the gene clustering tends to become more general. Consequently, this may lead to a loss of resolution in lower taxonomic levels within the dataset. We incorporated this information, along with additional details to provide a clearer understanding of the taxonomic scope of bacLIFE, into the manuscript. (L554-568)

17.

This rebuttal text about the manual selection of the inflation parameter is an important insight that should be added to the manuscript, so that a user knows. As far as I’m concerned the (almost) literal rebuttal text is good for this.

The manuscript now includes content from the rebuttal, clarifying that while users have the flexibility to choose the inflation value in bacLIFE, the tool does not incorporate an automatic mechanism for determining the optimal value for a specific dataset. (L624-630)

23.

Consider rewriting to “A large number of genomes in both datasets were not assigned to any lifestyles because they did not have species-level assignments in NCBI, which may compromise statistical power in the further steps of the microLife pipeline.” Or something similar.

In the manuscript, we rephrased as follows: “A significant proportion of genomes in both datasets remained unassigned to any lifestyles due to the absence of species-level designations in NCBI, which may compromise statistical power in the further steps of the bacLIFE pipeline.” (L165-168)

25.

Please add this insight not just in the rebuttal but also in the manuscript.

We included this discussion about the continuous reclassification of *P. fluorescens* genomes in the manuscript. (L188-192)

28.

It’s a good decision to remove the taxonomic classifier. Note that there are remnants of it still in the manuscript (e.g. line 101 ,line 495, line 605).

Thank you for the suggestion; we have eliminated these remains.

30.

Please add this insight to the manuscript as well (see earlier points about the type of genes that microLife may detect.

We incorporated these insights from the rebuttal into the manuscript text. (L350-358)

32.

I understand what you say, but it’s still unclear to me why you are doing this. Please consider at least a rewrite in the manuscript as to the reasons for this analysis. I might not be the only one that doesn’t understand.

The core of bacLIFE lies in its gene clustering mechanism. Although we initially validated this process by comparing it with tools such as Orthofinder, our objective was to extend the validation further. We specifically aimed to examine the distribution of genes that have been well studied in the literature. In the context of *Pseudomonas*, our focus was on genes associated with the Type III Secretion System components and effectors. The manuscript has been revised to clearly articulate our intention to validate the distribution of bacLIFE 's gene clusters by scrutinizing genes with established distributions in existing literature. (L251-255)

35.

I actually meant if a single genome could be annotated as both environmental and plant pathogenic for example, or that a LAG could be both environmental and plant pathogenic associated.

The utilization of binary classifiers may result in the prediction of a genome as both environmental and a plant pathogen. However, in such instances, the confidence values are likely to be lower. It's essential to note that, according to the criteria applied in this study (confidence values exceeding 0.8), no genome was assigned to two lifestyles simultaneously. Regarding LAGs, potential overlap can occur if the number of genomes in certain lifestyles is significantly lower compared to others. Consider a scenario with 100 animal pathogens, 10 plant pathogens, and 10 environmental genomes in the comparison. When comparing plant pathogens against the rest and environmental genomes against the rest (using the same thresholds as in this study), there might be some overlap of genes present in both plant pathogens and environmental, but not in animal pathogens.

Nevertheless, bacLIFE 's flexibility allows users to work around such challenges. To address potential overlap, users can conduct specific comparisons between individual groups. For instance, instead of comparing plant pathogens against all other groups collectively, users can perform separate comparisons like plant pathogen vs. animal pathogen and plant pathogen vs. environmental, tailoring the analysis to meet specific research needs.

36.

Thanks, very nice!

37.

Please add this insight (and also those in the rebuttal of 39) to the manuscript as well. Also, please make sure that at *every* place where you talk about genomic context, "consecutive genes" etc, it's clear how you define this (for example by stating the genomes in which the genomic content is evaluated). Right now this information is added to some places, but not *all* yet.

We added clarity in the text regarding the genomes we examined for the consecutiveness of LAGs. For the "consecutive" LAGs selected for mutagenesis, the existence of the LAG-enriched region was verified in three genomes representing distinct phytopathogenic species: *B. plantarii* ATCC 43733, *B. gladioli* ATCC 25417, and *B. glumae* AU6208. (L370-373))

40.

Very nice, well done!

42.

Please add this insight to the manuscript as well. It goes well with discussing the strength and weaknesses of using the LAG definition.

Incorporated into the manuscript as recommended. (L350-358)

46.

I agree with the authors that these 8 genes may not be genes involved in pathogenicity (or at least we do not know for sure), and I think they are a good example of genes that may be

identified because of the phylogenetic dependence, i.e. they are lifestyle-associated but not necessarily involved in niche adaptation. Please consider using these as an example for the suggested extra discussion about the type of genes that microLife may detect.

We introduced additional discussion in the manuscript as advised, addressing the LAGs selected for mutagenesis and their correlation with phylogeny. This reiterates the ongoing debate surrounding phylogeny versus lifestyle associations. Additionally, we discuss the challenges of the LAGs that exhibited reduced in vitro growth when mutated because of the interconnected nature of colonization and pathogenicity. (L413-426)

58.

Please add this constraint to the manuscript that some of the unique genes may be an artifact of the clustering parameters. I'd like that because the clustering is an important part of microLife.

We incorporated this possible explanation for the presence of unique genes into the manuscript. (L640-642)

64.

Please add the reasoning for the choice of Soerensen Dice to the manuscript. I understand that you normally don't have to defend these kind of choices but since it's part of the pipeline I do believe it's important in this case.

We incorporated the reasoning of using the Soerensen Dice distance in the manuscript. (L709-711)

68.

Please add the reason for removing these categories to the manuscript.

We incorporated the reasoning of removing these COG categories in the manuscript. (L232-234)

75.

Please add to the figure caption.

Added as suggested

79.

Please add this to the figure caption.

In Fig 7a, it was included that three clusters in the network are linked to *P. viridiflava*, *P. syringae*, and certain environmental strains like *P. fluorescens*. Subsequently, Fig 7c details that a random representative from each of these three clusters in the network was selected to generate the CORASON phylogeny.

86.

Please add this to the figure caption.

Added as suggested

87.

Please add this to the figure caption.

Added as suggested

89.

While I appreciate that you may not always agree with reviewers and cannot do everything, I did not like that the authors suggest that they have done this whereas in reality they have not. Apologies for any confusion; we have now removed the persistent exaggerated elements. In the conclusion, we retained the mention of the potential extension to eukaryotic organisms as work in this project is in its final stages of development.

Reviewer #2 (Remarks to the Author):

The authors have revised their manuscript in agreement with all my last suggestions. In my opinion, it is ready to be accepted for publication.

I hope this workflow will be highly used and cited :)

Reviewer #3 (Remarks to the Author):

Dear authors,

I thank you for answering my previous suggestions and comments, all my concerns have now been addressed. Congratulations with this excellent tool and very nice manuscript, I know it has been a lot of work. I may be a happy user of bacLIFE myself in the future.

When reading the manuscript I found some super minor textual things for the nitpicking (do with them what you want):

line 149: "allowing accurate taxonomic delineation of bacteria" -> this has been removed from the scope.

line 233: "mainly based in Eukaryote" -> Eukaryote*s*?

line 370: "direction in B. plantarri" -> *the* B. plantarri?

line 625: remove "indeed".

line 629: "its stablished" -> not sure what this means (typo?), consider rewrite.

Extended Data Fig. 3: it still says 'microLife'.

Best wishes!

REVIEWERS' COMMENTS

Reviewer #2 (Remarks to the Author):

The authors have revised their manuscript in agreement with all my last suggestions. In my opinion, it is ready to be accepted for publication.

I hope this workflow will be highly used and cited :)

Reviewer #3 (Remarks to the Author):

Dear authors,

I thank you for answering my previous suggestions and comments, all my concerns have now been addressed. Congratulations with this excellent tool and very nice manuscript, I know it has been a lot of work. I may be a happy user of bacLIFE myself in the future.

When reading the manuscript I found some super minor textual things for the nitpicking (do with them what you want):

line 149: "allowing accurate taxonomic delineation of bacteria" -> this has been removed from the scope.

Removed from the manuscript

line 233: "mainly based in Eukaryote" -> Eukaryote*s*?

Changed in the manuscript

line 370: "direction in B. plantarri" -> *the* B. plantarri?

Changed in the manuscript

line 625: remove "indeed".

Removed from the manuscript

line 629: "its stablished" -> not sure what this means (typo?), consider rewrite.

Rewritten as suggested

Extended Data Fig. 3: it still says 'microLife'.

Changed to 'bacLIFE'